# Development and application of a multi-scale modelling framework for urban high-resolution NO₂ pollution mapping

Zhaofeng Lv[1,★], Zhenyu Luo[1,★], Fanyuan Deng[1], Xiaotong Wang[1], Junchao Zhao[1], Lucheng Xu[1], Tingkun He[1],

Yingzhi Zhang[2], Huan Liu[1,*], Kebin He[1]

[1]State Key Joint Laboratory of ESPC, School of Environment, Tsinghua University, Beijing 100084, China

[2]College of Ecology and Environment, ChengDu University of Technology, Chengdu 610059, China

[★] Z. Lv. and Z. Luo. contributed equally to this work.

Corresponding Author:

*Phone and fax: 86-10-62771679; e-mail: liu_env@tsinghua.edu.cn (Huan Liu)

**Abstract.** Vehicle emissions have become a major source of air pollution in urban areas, especially for near-road environments, where the pollution characteristics are difficult to be captured by a single-scale air quality model due to the complex composition of the underlying surface. Here we developed a hybrid model CMAQ-RLINE_URBAN to quantitatively analyse the effects of vehicle emissions on urban roadside NO₂ concentrations at a high spatial resolution of 50 m × 50 m. To estimate the influence of various street canyons on the dispersion of air pollutants, a Machine Learning-based Street Canyon Flow (MLSCF) scheme was established based on Computational Fluid Dynamic and two machine learning methods. The results indicated that compared with the CMAQ model, the hybrid model improved the underestimation of NO₂ concentration at near-road sites with MB changing from -10 μg/m³ to 6.3 μg/m³. The MLSCF scheme obviously increased upwind concentrations within deep street canyons due to changes in the wind environment caused by the vortex. In summer, the relative contribution of vehicles to NO₂ concentrations in Beijing urban areas was 39% on average, similar to results from CMAQ-ISAM model, but increased significantly with the decreased distance to the road centerline, especially reaching 75% on urban freeways.

25 **Graphical abstract.**

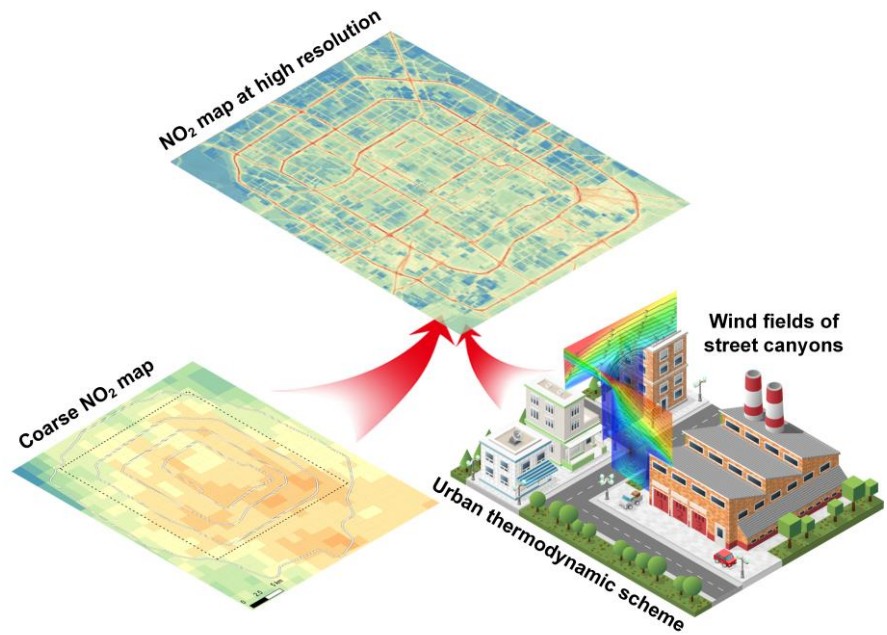

NO$_2$ map at high resolution

Coarse NO$_2$ map

Wind fields of street canyons

Urban thermodynamic scheme

## 1 Introduction

The accelerated urbanization leads to severe air pollution in China. As one of the indicators of air pollution, nitrogen dioxide ($NO_2$) causes an adverse impact on human health and promotes the generation of ozone and particulate matter (Pandey et al., 2005; Khaniabadi et al., 2017). During the last decade, benefiting from the implementations of several air pollution control strategies by the Chinese government, the air quality has improved (Jin et al., 2016; Zheng et al., 2018), and the vertical column densities of $NO_2$ displayed a decreasing trend after 2013 (Shah et al., 2020; Cui et al., 2021). However, the economic development and nitrogen oxides ($NO_x$) emissions are not decoupled in China (Luo et al., 2022a). In some megacities of China, such as Chengdu, the daily averaged $NO_2$ concentration could reach 200 $\mu g/m^3$ (Zhu et al., 2019), far exceeding the 24-h average air quality guideline of 80 $\mu g/m^3$ suggested by the Ministry of Environmental Protection of China.

The improvement of $PM_{2.5}$ in China was mainly due to the emission reduction and control measures of industrial and domestic sources (Zhang et al., 2019b), which also relieved the $NO_2$ pollution, but the reduction potential of these sources has been gradually declining. Meanwhile, as the population of vehicles is growing rapidly, vehicle emissions have become a major source of $NO_2$ pollution, especially in urban areas (Nguyen et al., 2018). Due to the low release height of vehicle emissions, combined with the negative dispersion condition caused by nearby buildings, air pollutants will be significantly accumulated near the street. According to roadside observations, within the distance of about 100-200 m near roads, the concentrations of CO, $NO_2$, ultrafine particulate matter (UFP), $PM_{2.5}$, $PM_{10}$, and other pollutants will increase with the decreased distance to the road centerline, especially for the pollution levels of $NO_2$ and UFP increasing exponentially. Therefore, the gradient of concentration around the road changes dramatically (Nayeb Yazdi et al., 2015; Hagler et al., 2012). Moreover, the dispersion of air pollutants in the near-road environment is significantly affected by geometric characteristics of the street canyon. For example, in a standard street canyon, when the external wind direction at the roof level is perpendicular to the street axis, a clockwise vortex will be generated inside, resulting in the accumulation of pollutant concentrations at the upwind grid receptors in the canyon (Oke, 1988; Manning et al., 2000). Consequently, how to quantitatively identify urban vehicle-induced air pollution around roads affected by complex underlying surface conditions has become an urgent scientific issue.

Regional-scaled air quality models, represented by Chemical Transport Models (CTMs) including Community Multi-scale Air Quality (CMAQ) model (Byun and Schere, 2006), Comprehensive Air quality Model with

extensions (CAMx), and Weather Research and Forecasting/Chemistry model (WRF-Chem) (Grell et al., 2005),
have been used extensively in assessment on the impacts of vehicle emissions on the regional atmospheric
environment, focusing on the source apportionment (Luo et al., 2022b; Vara-Vela et al., 2016; Kheirbek et al.,
2016; Lv et al., 2020) and evaluation of control measures (Zhang et al., 2020; Yu et al., 2019; Cheng et al., 2019;
Ke et al., 2017). However, the spatial resolution of CTMs is generally larger than 1 km×1 km, so the significant
impacts of vehicle emissions on near-source air quality cannot be predicted by CTMs due to the grid
homogenization on vehicle emissions.

To avoid the aforementioned disadvantages, the local-scaled numerical models based on Gaussian diffusion theory
or computational fluid dynamic (CFD) are adopted by numerous researches to study at a finer spatial resolution
(Zhang et al., 2021b; Patterson and Harley, 2019; Soulhac et al., 2012), including Research LINE-source Dispersion
Model (RLINE) (Snyder et al., 2013), Operational Street Pollution Model (OSPM), AERMOD (Cimorelli et al.,
2005), and RapidAir® (Masey et al., 2018). However, the large uncertainties in predictions from Gaussian
dispersion models come from the provided meteorological conditions and background concentrations. The natural
logarithm function is usually used to characterize the vertical profile of wind speed in both the inertial and rough
sublayers, neglecting the influence of urban complex underlying surface compositions on the wind field (Cimorelli
et al., 2005; Masey et al., 2018; Snyder et al., 2013). Nevertheless, in standard and deep street canyons, the changes
of vertical wind profile cannot be described by the logarithmic form, otherwise the actual wind speed will be greatly
overestimated (Soulhac et al., 2008). Although the OSPM has performed a large number of comparisons with field
observations in shallow or standard street canyons, the validation of model performance in deep street canyons
with a large aspect ratio was still inadequate (Kakosimos et al., 2010). Moreover, OSPM overestimated the bottom
wind speed in a deep street canyon by about 10 times compared with the predictions from CFD, resulting in greatly
underestimating pollutant concentrations (Murena et al., 2009). Comparatively speaking, CFD model can
accurately simulate the air flow and pollutant concentration in complex street canyons, but the simulation domain
of CFD model is much smaller than the urban scale, and the influence of the long-term meteorological boundary
conditions cannot be considered.

Considering the respective strengths and limitations of regional models and local models, several studies have been
carried out on coupling of air quality models applicable to different scales (Ketzel et al., 2012; Stocker et al., 2012;
Lefebvre et al., 2013; Jensen et al., 2017; Kim et al., 2018; Mallet et al., 2018; Hood et al., 2018; Benavides et al.,
2019; Kamińska, 2019; Mu et al., 2022). Although these models performed accurately in near-road simulation, the
influence of street canyons is still hard to be considered. In some hybrid models (Stocker et al., 2012; Jensen et al.,
2017; Mallet et al., 2018), OSPM was still applied to calculate concentration levels within the street, where the
application of logarithmic wind profile probably overestimated the bottom wind speed in a deep street canyon as
abovementioned. Other models simply assumed that in street canyons, wind direction followed the street direction,
and wind speed was uniform, which was not sufficient to resolve the concentration gradient within street canyons
(Kim et al., 2018; Benavides et al., 2019). Berchet et al. (2017) proposed a cost-effective method for simulating
city-scale pollution taking advantage of high-resolution accurate CFD, while the primary $NO_x$ was predicted due
to the lack of a chemical module. Therefore, it is essential to build an integrated model to predict long-term and
near-road air pollution suitable for the urban complex underlying surface environment.

The objective of the present work is to investigate the street-level $NO_2$ concentrations and quantify the contribution
of vehicle emissions considering the influence of the refined wind flow in complex urban environment. To this end,
a hybrid model CMAQ-RLINE_URBAN was developed by offline coupling the local RLINE model with the
regional CMAQ model and some localized urban thermodynamic parameter schemes. Specifically, in order to
predict the effects of urban street canyons on the diffusion of pollutants, we developed a Machine Learning-based
Street Canyon Flow (MLSCF) parameterization scheme to estimate the wind filed in a cost-effective way, which
was based on integrating two machine learning methods using big wind profile data from 1600 CFD simulations.
To evaluate the performance of CMAQ-RLINE_URBAN, simulations under several scenarios were conducted in
Beijing urban areas from August 1st to 31th of 2019, and validated through comparison with observations from
monitoring sites. Furthermore, spatial distribution characteristics of $NO_2$ concentrations in the near-road
environment were also analysed in this study.

**2 Materials and Methods**
**2.1 Hybrid model framework**
Here, we established the MLSCF scheme based on R language, and modified the code of RLINE model to add
other parameterization schemes with FORTRAN language. Finally, a multiscale air quality hybrid model was
developed to achieve a high-resolution $NO_2$ pollution mapping in urban areas. The framework of CMAQ-
RLINE_URBAN is shown in Figure 1. The hybrid model was established based on RLINE model, offline coupling
with the gridded meteorological field provided by WRF model and the pollutant background concentrations from
non-vehicle sources provided by CMAQ model with the Integrated Source Apportionment Method (ISAM),
considering the thermodynamic effects caused by the complex underlying surface compositions of the city. In our
hybrid model, a $NO_2$ pollution map with a high temporal (1 h) and spatial resolution (50 m×50 m) can finally be
obtained.

RLINE is a Gaussian line source dispersion model developed by Snyder et al. (2013) to predict pollutant
concentrations in near-road environments. In the RLINE model, the mobile source is considered as a finite line
source, from which the concentration is found by approximating the line as a series of point sources and integrating
the contributions of point sources using an efficient numerical integration scheme. The number of points needed
for convergence to the proper solution is a function of distance from the source line to the receptor, and each point
source is simulated using a Gaussian plume formulation. The RLINE model performs generally comparable results
when evaluated with other line source models for on-road traffic emissions dispersion (Snyder et al., 2013; Heist
et al., 2013; Chang et al., 2015), and has been successfully used in many studies to evaluate the impacts from traffic
emissions on air quality (Zhai et al., 2016; Valencia et al., 2018; Benavides et al., 2019; Filigrana et al., 2020;
Zhang et al., 2021a).

The simulation for local meteorological conditions in CMAQ-RLINE_URBAN included three steps: Estimation
for areas above the top of Urban Canopy Layer (UCL), inside of UCL, and inside of the street canyon. (1) In this
study, the configuration of WRF model referred to our previous study (Lv et al., 2020). The height of midpoint in
the bottom layer to the ground was set as 22.5 m, which was close to the average height of buildings near street
canyons, similar to the settings in the previous study (Benavides et al., 2019). Therefore, the meteorological field
simulated by the WRF model was used as the wind field and atmospheric stability at the top of UCL. During the
hybrid model running, the meteorological conditions over buildings near each road were obtained separately from
WRF model according to the road location. (2) Then, the surface roughness length ($z_0$) of each road was estimated
based on the surrounding building geometry and used to recalculate the localized meteorological parameters (e.g.
Monin-Obukhov length) within UCL according to the algorithm proposed by Benavides et al. (2019) ($z_0$ scheme).
The atmospheric turbulence intensity in urban areas around sunset in the afternoon was obviously enhanced
considering the influence of the urban heat island effect based on methods in the AERMOD model (Cimorelli et
al., 2005) (UHI scheme). The UHI scheme would affect the turbulent intensity based on the evaluation for the
upward surface heat flux and the urban boundary layer height due to convective effects, and then the mixing height,
convective velocity scale, surface friction velocity, and Monin-Obhukov length were all recalculated (details in the
Supplement Section S1). (3) Finally, the wind field within UCL was calculated according to different types of road
environments: open terrain and street canyon. The logarithmic wind profile based on Monin-Obhukov Similarity
Theory (MOST) (Foken, 2006) in the original RLINE model was still used when the grid receptor was located in
the open terrain (MOST scheme), while the MLSCF parameterization scheme was used for grid receptors within
the street canyon to quantitatively characterize the influence of the street canyon geometry and the external wind
environment at the top of the roof. The detailed introduction for street canyon geometry and the MLSCF scheme
was described in the following section.

The real-time vehicle emission inventory used in both regional and local air quality models was based on Street-
Level On-road Vehicle Emission (SLOVE) Model developed in our previous study (Lv et al., 2020), which was
based on the real-time traffic condition data from AMap. The daily averaged $NO_x$ emission from on-road vehicles
in Beijing in 2019 was estimated to be 136.0 Mg, of which emissions from heavy duty vehicles and heavy duty
trucks accounted for 31% and 34%, respectively. In our simulation, the concentrations of NO, $NO_2$, and $O_3$
excluding contributions from vehicle emissions were used as background concentrations at the roof level, avoiding
the double counting in the coupling process. These background concentrations were simulated by CMAQ-ISAM
model, in which the emissions were divided into local mobile and other four emission groups to trace their
contributions separately, so the influence of non-local vehicle emissions was considered, and details were presented
in our previous study (Lv et al., 2020). The spatial resolution of the innermost domain in both WRF and CMAQ
model was 1.33 km×1.33 km. In addition, the influence of atmospheric turbulence and building geometry on the
vertical mixing of background concentration was considered (vertical mixing scheme). The ratios of wind speed at
surface and roof levels were used as a proxy to calculate the contribution of background concentration over street
canyons to the near-ground level (Benavides et al., 2019). In this scheme, the surface wind was from MLSCF
scheme when the gird receptor is located within the street canyon, and otherwise the logarithmic wind profile was
used to calculate the wind speed at the specified height, and details were showed in the Supplement Section S2.
Finally, combined with the vehicle-induced primary $NO_x$ concentration calculated by the RLINE kernel, the high
spatial resolution $NO_2$ map could be simulated considering the photochemical process of $NO_x$. In this study, a
simplified two-reaction scheme, including the photolysis of $NO_2$ and the oxidation of NO, was incorporated into
the model to characterize the photochemical process of $NO_x$ (details in the Supplement Section S3), which has been
successfully applied in the SIRANE dispersion model (Soulhac et al., 2017).

**2.2 Development for MLSCF scheme**
**2.2.1 The database of street canyon geometry**
We first established a database of street canyon geometry for 15,398 roads in urban areas of Beijing based on the
three-dimensional building data obtained from our previous study (Lv et al., 2020) using Geographic Information
System (GIS). Three typical parameters to represent street canyon geometry were investigated, including height
ratio ($H_l/H_r$) ($H_l$ is the building height on the left side, while $H_r$ is the building height on the right side), aspect
ratio ($H/W$) ($H$ is set to be the average height, and W is the width of the street canyon), the canyon length to height
ratio ($L/H$) ($L$ is set to be the length of the street canyon). In this study, the extremely special geometry of canyons
was not considered, and the typical street canyons were selected as the following conditions: (1) The proportion of
actual street canyon length (the length of road where the buildings nearby) was greater than 0.5; (2) $H/W$ was
greater than 0.2; (3) $H_l/H_r$ was between 0.3 and 3.3. Finally, the total number of the typical street canyon was
1,889, with a total length of 787 km. The spatial distributions of canyon geometry are shown in Figure S1 in the
Supplement. In urban areas of Beijing, street canyon was generally wide with the averaged width of 50.3 m, and
buildings on both sides were relatively low with a mean of 23.6 m. Most street canyons were obviously located in
areas within the fourth ring road. The shallow ($H/W \leq 0.5$) canyons and long canyons ($L/H > 7$) were dominated,
accounting for 54% and 84% of the total number of street canyons.

**2.2.2 Description of CFD cases**
Here, to predict air flow in street canyons comprehensively, CFD simulations were conducted under combinations
of different values of controlling factors based on ANSYS FLUENT (v19.2). The controlling factors included the
aforementioned three typical parameters to represent canyon geometry, the background wind speed at the height of
$H$ ($V(H)$) and the angle between wind direction and street axis (α) to describe the external wind environment. The
selected values of each factor were listed in Table 1, and total 1600 (i.e., 5×4×4×5×4) simulations were
implemented.

In this study, the computational domain of three-dimensional (3D) full-scale CFD simulations is shown in Figure
2. The average building height $H$ of the street canyon was always set to 21 m in different simulations, which was
similar to the mean street canyon height in Beijing. Other actual size of street canyons (e.g., street canyon width
$W$) was calculated according to the ratio of each specific simulation. Distances between urban canopy layers (UCL)
boundaries and the domain top, domain inlet and domain outlet were set as $5H$, $5H$, and $20H$, respectively.

The turbulence closure schemes for CFD include the Reynolds-Averaged Navier-Stokes (RANS) and the Large-
Eddy Simulation (LES), and the choice of them depends on the computational cost, the accuracy required and the

purpose of application. The RANS resolves the mean time-averaged properties with all the turbulence motions to be modelled, while LES adopts a spatial filtering operation and consequently resolves large-scale eddies directly and parameterizes small-scale eddies (Zhong et al., 2016). Compared with the LES, the RANS is more easily established and computationally faster (Xie and Castro, 2006). However, the LES can provide a better prediction of air flow than that from the RANS when handling complex geometries (Dejoan et al., 2010; Santiago et al., 2010). In this study, considering the huge computational burden of a large number of simulations and the relatively simple geometry of street canyons in our modelling, the RANS was selected to characterize the air flow.

Following the CFD guideline (Tominaga et al., 2008; Franke et al., 2011), zero normal gradient conditions or pressure outlet conditions were applied at the domain outlet, and symmetry boundary conditions were adopted at the domain top and two lateral domain boundaries. For near-wall treatment, no-slip wall boundary conditions with standard wall functions were used (Fluent, 2006). All governing equations for the flow and turbulent quantities were discretized by the finite volume method with the second-order upwind scheme. The SIMPLE scheme was used for the pressure and velocity coupling. The residual for continuity equation, velocity components, turbulent kinetic energy, and its dissipation rate were all below $10^{-5}$. Meanwhile, the CFD simulation would also stop when the iteration steps exceeded 10,000, due to the large computing cost of so many simulations. In summary, the average iteration steps of total 1600 cases were 4,443. About 54.6% of cases met the convergence criteria, and the median residual values of continuity equation, velocity in X axis, velocity in Y axis, velocity in Z axis, $k$ and $\varepsilon$ were $1.0\times10^{-5}$, $8.5\times10^{-7}$, $8.5\times10^{-7}$, $4.1\times10^{-7}$, $3.4\times10^{-6}$ and $5.4\times10^{-6}$, respectively, indicating the overall model performance was acceptable. The selected turbulence model and grid arrangement are discussed in the following section.

At the domain inlet, the power-law velocity profile (Brown et al., 2001), vertical profiles of turbulent kinetic energy $k_{in}$ and its dissipation rate $\varepsilon_{in}$ at the domain inlet (Lien and Yee, 2004; Zhang et al., 2019a), were described below:

$$U_0(z) = U_{ref} \left( \frac{z}{H_{ref}} \right)^{\alpha}$$

$$k_{in}(z) = \left( I_{in} \times U_0(z) \right)^2$$

$$\varepsilon_{in}(z) = \frac{C_\mu^{3/4} k_{in}^{3/2}}{\kappa z}$$

Here, $U_0(z)$ stood for the stream-wise velocity at the height $z$. $U_{ref}$ represented the reference speed. The reference
height $H_{ref}$ was 21m. The power-law exponent of $\alpha$=0.22 denoted underlying surface roughness above medium-
dense urban area (Kikumoto et al., 2017). Turbulence intensity $I_{in}$ was 0.1, Von Karman constant $\kappa$ was 0.41 and
$C_\mu$ was 0.09.

**2.2.3 The CFD validation**
In this study, the stream-wise and vertical velocity predicted by CFD within street canyons was compared with
wind tunnel data in previous researches. For buildings of the cube arrays model, wind tunnel data from Brown et
al. (2001) was used to evaluate the reliability of CFD results by measuring vertical profiles of velocity. In this
experiment, street canyon was perpendicular to the wind direction at the roof level. For long-street models, we
predicted horizontal profiles of velocity along the street centerline at the height of z=0.11H or vertical profiles at
some points and then validated CFD simulations using wind tunnel data from Hang et al. (2010). In this validation
case, the wind direction at the roof level was parallel to the axis of street canyons. The description and validation
results are shown in Figure S2-S3, and Table S1 in the Supplement, respectively.

We identified the influence of different minimum sizes of hexahedral cells near wall surfaces (fine: 0.1m, medium:
0.2m, and coarse: 0.5m) and turbulence models (standard k-ε model and RNG k-ε model) on the predicted velocity,
to evaluate the grid independence and turbulence model accuracy (Figure S3 in the Supplement). The results
indicated that the predictions from the standard k-ε model could well match the variations of observed velocity
within the street canyon, of which performances were much better than that of the RNG model. In addition, different
grid resolutions used in simulations would not obviously affect the predicted results. We finally adopted the
standard k-ε model to characterize turbulence, and the minimum size of hexahedral cells near wall surfaces was
0.5 m with an expansion ratio of 1.1 was applied to save the computing cost, and the average mesh number in total
80 street canyon models is 1,367,965.

Moreover, the averaged wind speed from CFD in street canyons with different aspect ratios and external wind
direction was compared with predictions from other empirical methods used in SIRANE model (Soulhac et al.,
2012) and MUNICH model (Kim et al., 2018). Similar predictions using different methods also proved the
reliability of CFD simulation in this study (Figure S4 in the Supplement).

### 2.2.4 Machine learning

Data driven method, such as machine learning and deep learning, is now a successful operational geoscientific processing schemes and has co-evolved with data availability over the past decade (Reichstein et al., 2019). Specially, these models have been used as computationally efficient emulators of explicit mechanism models, to explore uncertainties (Aleksankina et al., 2019) and sensitivities or replace complex gas-phase chemistry schemes (Keller and Evans, 2019; Conibear et al., 2021). In addition, meta-models (Fang et al., 2005) such as neural networks and Gaussian process (Beddows et al., 2017) are also used to produce a quick to run model surrogate and show reliable performance. Random Forest (RF) model algorithm is an ensemble learning method that generates many decision trees and aggregates their results, which has been developed to solve the high variance errors typical of a single decision tree (Breiman, 2001). Multivariate Adaptive Regression Splines (MARS) is a nonparametric and nonlinear regression method, which can be regarded as an extension of the multivariate linear model (Friedman, 1991). RF and MARS are common machine learning methods which run efficiently on large data sets, and are relatively robust to outliers and noise. Furthermore, they never require the specification of underlying data model and the complex parameter tuning, and they can still provide efficient alternatives and generally show a high accuracy in applications for predict air pollutant concentrations (Hu et al., 2017; Chen et al., 2018; Kamińska, 2019; Geng et al., 2020).

Here, based on the database including 42,880 samples obtained from 1600 CFD simulations, the RF and MARS were both used to simulate the wind vector along X-axis ($V_x$) and Y-axis ($V_y$) at different heights within the street canyon respectively. The $V_x$ and $V_y$ were the average of all velocities along X or Y axis over the same horizontal profile at a specific height within the street canyons. The input predictor variables included $H/W$, $L/W$, $H_l/H_r$, the grid receptor relative height ($z/H$), the background wind vector at the height of H along X-axis ($Vbg_x = V(H) \times \sin\alpha$) and Y-axis ($Vbg_y = V(H) \times \cos\alpha$). We finally combined the advantages of these two machine learning models and developed the MLSCF scheme to predict wind environment in street canyons and incorporated into the hybrid model, which is discussed in the section 3.1.

In RF model, the number of predictors randomly sampled at each split node in the decision tree ($m_{try}$) and the number of trees to grow ($NumTrees$) are two important hyperparameters that determine the performance of the model. Similarly, in MARS model, the two important hyperparameters are the total number of terms ($nprune$) and the maximum number of interactions ($degree$). By comparing the mean squared error (MSE) for testing datasets

across models with candidate parameter combinations, we set $m_{try}$ and $NumTrees$ as 6 and 200 in RF, respectively, and $nprune$ and $degree$ as 23 and 3 in MARS, respectively. Additionally, the 10-fold cross-validation (CV) repeated ten times were considered to evaluate the prediction performance of our models. The total dataset was randomly divided into 10 subsets, where 9 subsets was used to train model and another was applied for validation. The fitted coefficients of MARS are shown in Table S2-S3 in the Supplement.

In order to identify the sensitivity and response relationship between prediction variables and results in RF model, we used the MSE for out-of-bag (OOB) to evaluate the relative importance of each feature to $V_x$ and $V_y$, by randomly replacing the value of a single prediction variable one by one (Liaw, 2002). Higher values of increase in MSE indicated that the predictor was more important. In addition, Partial Dependence Plots (PDPs) was applied to establish the response relationship between the change of a single predictive variable and the predicted results, considering the average influence of other variables (Greenwell, 2017).

**2.3 Configuration of CMAQ-RLINE_URBAN**

The near-ground $NO_2$ concentrations were simulated from August 1st to 31th in 2019 when the average of daily high temperatures was higher than 30 °C and sunlight duration was longer than 13 hours, leading to strong photochemical reactions. The simulation domain for the hybrid model covered the core urban areas within and surrounding the fifth ring road, shown in Figure 3. The receptors included both grid receptors and monitor receptors. The grid receptors were set at a spatial resolution of 50 m×50 m, and the height above the ground was 1.5 m, which was equivalent to the height of the human breathing. We used data from 10 observation stations (monitor receptors) located in the normal urban environment and 5 near-road monitoring sites for validation (Beijing Ecological Environment Monitoring Center, available at http://zx.bjmemc.com.cn/) (DSH, NSH, QM, XZM, and YDM) in the simulation domain (Figure 3), which were 10 meters and 3 meters above the ground respectively. The QM and XZM sites were located in shallow street canyons, and details for the morphometric of near-road measurement sites were shown in Table S4 in the Supplement.

In general, compared to the RLINE model, CMAQ-RLINE_URBAN has the following improvements:

(a) The gridded meteorological parameters provided by the WRF model were used.

(b) Gridded non-vehicle-related concentrations provided by CMAQ-ISAM model were used as background concentrations.

(c) A simple $NO_x$ photochemical scheme was incorporated to simulate $NO_2$ concentrations.
(d) Thermodynamic effects caused by the special underlying surface structures of the city were considered,

330       including UHI effects, the influence of local buildings on turbulence intensity and vertical mixing of

331       background concentrations.

(e) A newly developed MLSCF scheme was applied to predict wind environment in street canyons.

In our simulation, the model configurations in the base scenario CMAQ-RLINE_URBAN included all (a)-(e)
schemes, and the other two control scenarios were set to investigate the sensitivity of urban schemes on predictions,
where all input data was set to be the same. The scenario CMAQ-RLINE only including (a)-(c) schemes was set to
analyze the impacts of urban thermodynamic schemes, and the scenario CMAQ-RLINE_URBAN_nc including
(a)-(d) schemes was set to identify the impacts of the MLSCF scheme. Although the wind environment for each
road at the top of the canyon was provide by the WRF model in all scenarios, the calculation of wind profile within
the street canyon was different. It was estimated based on the MOST theory in the CMAQ-RLINE and CMAQ-
RLINE_URBAN_nc rather than that from the MLSCF in the CMAQ-RLINE_URBAN.

**3 Results**
**3.1 Fitting results of machine learning**
In this study, the 10-fold cross-validation (CV) repeated ten times was considered to evaluate the prediction
performances of RF and MARS models. As shown in Figure 4 and Figure S5, both models performed acceptable
robustness in CV, indicating that neither RF nor MARS model overfitted the data. In general, the performances of
both models in predicting $V_y$ was better than that in $V_x$ of which the absolute value was relatively small, especially
for MARS model. Since $V_x$ was responsible for the formation of the vortex within street canyons and affected by
multiple factors, it was more difficult to be simulated. The averages of mean absolute error (MAE), root mean
square error (RMSE), and correlation coefficient (R) in the CV of the RF model for $V_x$ and $V_y$ were 0.04 m/s and
0.05 m/s, 0.02 m/s and 0.03 m/s, and 0.99, respectively. Although the average of the relative error (RE) was a little
high (42.5% and 43%), particularly when the predicted wind speed was low, the median RE were relatively low
with 9.8% and 2.7%, respectively, indicating an acceptable performance. Compared with the advanced non-linear
RF algorithm, the MARS model performed not very well, especially when the absolute value of $V_x$ was greater than
1 m/s and $V_y$ was less than 3 m/s. However, when the predicted wind speed by machine learning methods was
compared with observations from wind tunnel experiments, we found that the performance of the MARS model
was obviously better than that of RF model in one of validation cases (see Figure 5). The decision tree model like
RF failed to respond to the parts beyond the range of prediction variables ($Vbg_y$=17 m/s >>5 m/s), while the more
reasonable predictions can be obtained by the MARS model which used piecewise linear function essentially.
Therefore, the MLSCF scheme was established based on a method to combine the advantages of each model. The
RF model was used when the input value was within the range of predictors shown in Table 1, otherwise the
predictions from the MARS model were used.

In addition, the importance of each predictor variable in the RF model was investigated to explain their impacts on
predictions. As shown in Figure 6, the background wind speeds on x and y axis played vital roles in predictions of
$V_x$ and $V_y$, respectively, followed by the relative height ($z/H$). Among the geometric parameters of the street
canyon, the impact of $L/W$ was least. Since $V_x$ was the main driving force for the formation of vortices in street
canyons, it was more affected by the geometry of street canyons especially $H_l/H_r$, comparing to $V_y$. This feature
importance ranking was basically consistent with the conclusion in a previous study (Fu et al., 2017). Figure S6 in
the Supplement shows the PDPs of each predictor variable in RF model for $V_x$ and $V_y$. As $z/H$ grew, $V_x$ and $V_y$
showed linear and logarithmic increase patterns, respectively. And the resistant effect of windward buildings on
wind speed enhanced with the increasing of $H_l/H_r$, resulting in a significant decrease in $V_x$ particularly when
$H_l/H_r$ was lower than 1.25. The relationship between predictors and results in the model was consistent with the
actual mechanism, indicating our model could provide an accurate description of the wind field in the street canyon.

**3.2 Impacts of MLSCF on simulations in street canyons**
We compared the differences between monthly mean wind profile in different street canyons including QM
(shallow canyon: $H/W = 0.22$), XZM (shallow canyon: $H/W = 0.35$), SZJ (standard canyon: $H/W = 1$) and
JTDL (deep canyon: $H/W = 1.93$), calculated by the default logarithmic function based on MOST in the original
RLINE model (Foken, 2006), and the MLSCF scheme developed in this study. As shown in Figure 7(a)-(d), the
wind profile estimated by MOST showed a logarithmic change at the height above displacement height ($d_h$) with
a decrease to 0 at $d_h$, and remained constant below $d_h$ (the $d_h$ is calculated by multiplying surface roughness length
($z_0$) times a factor which is recommended to be set as 5). Compared with the MOST, the simulated wind speeds
near the ground and at the top of canyons were generally lower based on the MLSCF scheme in shallow and
standard street canyons. In the deep street canyon, the significant reduction in ventilation volume led to the mean
wind speed simulated by the MLSCF scheme much lower than that of MOST at all heights. Although the aspect
ratios of the street canyon located in QM and XZM were similar, their orientations were quite different, resulting
in significant differences under prevailing external winds in different directions. Since the prevailing northerly and
southerly wind was observed in Beijing during the study period, the resistance effect of the buildings on both sides
of the east-west street canyon located in QM was more obvious.

We also investigated the impacts of the MLSCF on hourly wind direction at the bottom ($z = 3m$) of different street
canyons by comparing the roof-level predictions from WRF model (see Figure 7(e)-(f)). In the shallow street
canyon like QM, the simulated wind direction at the bottom was consistent with the background on the whole, with
the R reaching 0.8. When the background wind direction was less than 180°, the averaged wind direction at the
bottom simulated by MLSCF was 91.8°, which was basically consistent with the angle between the street and the
south direction (84.5°). When the background wind direction was greater than 180°, the average wind direction
predicted by MLSCF (257.4°) was similar to that in the opposite direction of the street (264.5°), which was in line
with the theory proposed by Soulhac et al. (2008) that the average wind direction in street canyons was assumed to
be consistent with the (opposite) orientation of the street. While in the deep street canyon of SZJ, when the external
wind perpendicularly blew to the street, the wind direction at the bottom was completely opposite to that at the top
due to the formation of vortex, with the R reaching -0.97. In conclusion, compared with the traditional MOST
method, the newly developed MLSCF scheme could well simulate the influence of the external wind environment
and geometry on the wind field inside the street canyon.

As shown in Figure 8, the impacts of the MLSCF scheme on simulated $NO_2$ concentration were identified by the
differences between CMAQ-RLINE_URBAN and CMAQ-RLINE_URBAN_nc scenario during a clean day
(August 24th). When the atmosphere was stable at night, in street canyons with a large aspect ratio, the wind
direction at the bottom changed to the opposite to that at the top, combined with the decreased wind speed affected
by the MLSCF scheme, the $NO_2$ concentrations at upwind grid receptors increased by up to 80 μg/m$^3$. Meanwhile,
the changes in wind direction would also decrease the concentrations at downwind grid receptors by up to 20 μg/m$^3$.
For example, in the SZJ standard canyon, the background wind direction over the street was 79° (easterly), and
the wind direction at the bottom changed to 291° affected by the MLSCF scheme (westerly). Therefore, the upwind
$NO_2$ concentrations increased, and the location of peak $NO_2$ concentration shifted to the windward. Since the
changes in $NO_2$ concentrations were also influenced by the local on-road emissions, the increase was only up to
2.1 μg/m$^3$ in SJZ street, where the traffic flow and vehicle emissions were small at night. However, a little influence
was observed during the day in the convective boundary layer. During this period, although the wind direction at
the bottom was not changed obviously due to the parallel background wind in SZJ street, the increased surface
wind speed was beneficial for the dispersion, resulting in the decreased concentration in grid receptors within both
sides of the street canyon. In summary, the MLSCF scheme enabled the characterization of the concentration
distribution in street canyons.

**3.3 Performance of near-road simulations from different models**

The performances in predicting $NO_2$ concentrations at all monitor receptors from different models were first
compared, including CMAQ-RLINE_URBAN, CMAQ-RLINE and CMAQ model. The mean bias (MB), RMSE,
normalized mean bias (NMB), normalized mean gross error (NMGE), the fraction of predictions within a factor of
two (FAC2), Index of agreement (IOA), and $R$ between simulations and observations were all selected as statistical
indicators for the evaluation (Table 2). In general, the performance of CMAQ-RLINE_URBAN was the best at all
urban sites. Compared to the CMAQ model, the averaged MB and NMB at urban sites in the hybrid model
decreased from 8 $\mu g/m^3$ to 1.3 $\mu g/m^3$ and 27% to 4%, respectively.

Diurnal variations of observed and predicted hourly averaged $NO_2$ concentrations at near-road sites from different
models were mainly compared and shown in Figure 9. The comparison of hourly and daily averaged concentrations
is shown in Figure 10. Overall, the CMAQ-RLINE_URBAN performed best with the smallest deviations. By
comparing the performances of the CMAQ and CMAQ-RLINE scenario, we found the direct coupling between the
CMAQ and RLINE models could reproduce the high $NO_2$ concentrations at near-road sites in daytime, and
significantly improve the underestimation of near-source concentrations due to grid dilution on emissions in
CMAQ model. The averaged MB and NMB at all sites changed from -10 $\mu g/m^3$ to 25.6 $\mu g/m^3$, and from -20% to
51%, respectively. However, a significant overestimation was found in the CMAQ-RLINE at night (0:00-6:00) and
around sunset in the afternoon (16:00-23:00), of which the peak could exceed the observed concentrations by more
than 1 times. This overestimation was reduced in the CMAQ-RLINE_URBAN, where the urban thermodynamic
schemes were implemented. The averaged MB and NMB decreased to 6.3 $\mu g/m^3$ and 12%, respectively, due to the
following reasons: (1) The increased surface roughness length slightly enhanced local turbulence intensity near
roads; (2) The UHI scheme enhanced the intensity of atmospheric turbulence in urban areas before and after sunset
in the afternoon; (3) The effect of turbulence intensity on the local vertical mixing of background concentrations
was considered, significantly reducing the mixing ratio of concentrations over UCL and near the ground at nights
in the stable boundary layer (Figure S7 in the Supplement), which was probably the main driving force of decreased
predictions in the hybrid model (Benavides et al., 2019). However, the CMAQ-RLINE_URBAN slightly
overestimated the nighttime $NO_2$ concentration of all observation stations except the DSH, which was probably
caused by overestimations of background concentrations from CMAQ-ISAM and vehicle emissions.

The accuracy of model performances at each traffic site showed a little difference affected by the variations in the
traffic flow and emissions of nearby roads, as well as the geometry of surrounding buildings and street canyons. At
DSH and NSH sites, which were adjacent to ring roads as the main urban freight corridors with a high traffic flow
including a large proportion of trucks, the high $NO_x$ emissions led to the highest roadside $NO_2$ observations among
all sites. The CMAQ model would significantly underestimate the high $NO_2$ concentration at sites nearby ring roads,
with MB and NMB lower than -15 $\mu g/m^3$ and -28% (Table S5 in the Supplement), respectively, which was
improved using CMAQ-RLINE_URBAN. However, the hybrid model performed a minor overestimation at the
NSH site, since the monitor was actually positioned in the road centerline but assumed to be located downwind in
the model, resulting in a relatively large systematically error (Snyder et al., 2013). In total, CMAQ-
RLINE_URBAN performed best among all models, especially improving the estimation of $NO_2$ concentrations
near roads by the original regional model.

Additionally, Figure S8 in the Supplement shows the comparison between simulated and observed roadside hourly
and daily maximum 8-hour average $O_3$ concentrations by different models, and their diurnal variations are shown
in Figure S9. Generally, the hybrid model significantly improved the overestimation of daytime $O_3$ concentrations
by the CMAQ model when considering the titration effect of high NO concentration near roads on $O_3$. In the hybrid
model, the peak time was delayed to about 15:00, which was closer to the observation, but still 1-2 hours earlier
than the actual time, which may be related to the uncertainty in $NO_2$ photolysis rate.

**3.4 Spatial distribution characteristics of simulated concentrations**
We investigated the differences between the spatial distribution of the monthly averaged $NO_2$ concentration
simulated by the CMAQ and CMAQ-RLINE_URBAN models, as shown in Figure 11. Since the urban
thermodynamic schemes were considered in the hybrid model, the overestimation of most urban environmental
grid receptors by CMAQ model was relieved. Within the fourth ring road and its surrounding areas, the mean
concentration of $NO_2$ from CMAQ-RLINE_URBAN was 30.1 $\mu g/m^3$, lower than that from the CMAQ model (39.5
$\mu g/m^3$). The overall spatial distribution characteristics of $NO_2$ predictions from both models showed that the
concentrations in south regions were high due to the pollution transport from Hebei province (An et al., 2019).
However, near-road hotspots for the $NO_2$ pollution were identified in the hybrid model where the spatial resolution
of results increased to 50 m×50 m. The $NO_2$ concentrations nearby ring roads with high traffic flow and emissions
were up to 120 μg/m$^3$, much higher than the maximum prediction from CMAQ model (52.4 μg/m$^3$). In addition,
the simulated near-road concentrations from the hybrid model during traffic peak hours (18:00-19:00) were
significantly higher than those at noon (12:00-13:00), while there were few changes in results from CMAQ model
(Figure S10 in the Supplement).

The $NO_2$ concentrations estimated by CMAQ-RLINE_URBAN at all grid receptors grids followed a two-mode
Gaussian distribution (Figure S11 in the Supplement), which was similar to Zhang's results (Zhang et al., 2021b).
The $NO_2$ concentrations as a result of vehicle emissions were further calculated by the differences between the total
and background concentrations. In general, the vehicle-induced $NO_2$ concentrations in urban areas was 11.8 μg/m$^3$,
accounting for 39% of the total concentrations, which was similar to the predicted contribution from the CMAQ-
ISAM model (42.5%).

Figure 12 shows the changes in $NO_2$ concentrations simulated by the hybrid model with distance from the grid
receptors to its nearest road centerline. The concentrations at grid receptors within 200 m from road were
significantly affected by vehicle emissions. Within 50 m around the road, as the distance from grid receptors to the
road centerline gradually increased, the $NO_2$ concentrations decreased exponentially. The total $NO_2$ concentrations
decreased from 53.1 μg/m$^3$ to 30 μg/m$^3$, and the vehicle-induced concentrations also dropped from 34.7 μg/m$^3$ to
12.6 μg/m$^3$. The concentrations near roads with different types were highly dependent on the emission intensity.
The $NO_2$ concentration was highest in the center of the urban freeway, which was 76 μg/m$^3$ and about 1.9 times
higher than that on local roads. The relative contribution of vehicle emissions to $NO_2$ concentration reached up to
75.3% on urban freeways, as well as 71.9% and 65.5% on artery roads and freeways, but only 51.1% on local roads.
It was worth noting that although the $NO_2$ concentrations at far grid receptors to the road on highways were slightly
higher than those on other road types, the contribution of vehicle emissions was the least. It was since the $NO_x$
emission intensity of freeways was as high as that on artery roads, but the density and height of buildings around
freeways were usually low, resulting in a high vertical flux of background concentrations from the top of UCL to
the ground. In conclusion, the results from the hybrid model accurately reflected not only the impacts of local on-
road emissions, but also the pollution characteristics affected by non-vehicle sources at the regional scale.

## 4 Conclusion and Discussions

In this study, we developed a hybrid model CMAQ-RLINE_URBAN to quantitatively analyse the effects of vehicle emissions on urban roadside $NO_2$ concentrations at a high spatial resolution of 50 m $\times$ 50 m. The main conclusions of this study are as follows:

The developed MLSCF scheme revealed that affected by the geometry of buildings on both sides of the road, the wind filed in the street canyon sometimes was quite different from that in the environmental background. In deep street canyons, the wind speed at the bottom decreased obviously due to the resistant effect of buildings, and the directions of horizontal flow at bottom and top of the canyon were completely opposite due to the formation of vortex. The application of MLSCF scheme in the hybrid model led to increase $NO_2$ concentrations at upwind grid receptors within deep street canyons due to changes in the wind environment. However, the influence of the turbulence induced by street canyon effects on the mixing of air pollution was not considered on which we will make effort in the future.

The comparison between observations and predictions showed that the hybrid model significantly improved the underestimation of near-source concentrations due to grid dilution on emissions in CMAQ model. The implementation of the urban thermodynamic schemes in the hybrid model also relieved the overestimation in night-time $NO_2$ concentrations from the CMAQ directly coupled with RLINE model. The predictions from CMAQ-RLINE_URBAN model could accurately reflect not only the impact of road local emissions, but also the pollution characteristics of non-vehicle sources at regional level. It revealed that in summer, the average contribution of vehicle emission to $NO_2$ concentrations in urban areas of Beijing was 11.8 $\mu g/m^3$, and the relative contribution accounted for approximately 39%. Moreover, the vehicle-induced $NO_2$ pollution increased significantly with the decreased distance to the road centerline, especially reaching 76 $\mu g/m^3$ (75%) on urban freeways.

On the basis of this study, the following perspectives are proposed for future research: (1) At present, the execution time during 1 h running CMAQ-RLINE_URBAN over the urban domain was about 3.9 hours in average, which reached 4.8 hours at night due to the difficulty of convergence in the condition of the high atmospheric stability. Therefore, considering the running cost, the grid resolution of area in Beijing 5th ring road and its surroundings can reach 50 m$\times$50 m. We will make efforts to develop a parallel computing method to reduce the computing time, in order to improve the grid resolution of a relatively large-scale simulation. (2) In our study, a simplified two-reaction scheme was incorporated into the model to characterize the photochemical process of $NO_x$, since it

performed similar predictions and less computational time compared with those of the complicated CB05 gas phase
chemical mechanism (Kim et al., 2018). However, another study pointed that the impact of nonlinear $O_3$-$NO_x$-VOC
chemistry on $NO_2$ concentrations in the deep canyon was nonnegligible (Zhong et al., 2017). The influence of
different chemistry schemes on near-road simulation will be investigated in the future. (3) The long-term site-
observation of wind environment and pollutant concentrations in various street canyons were suggested to be
compared with modelling results, especially in deep street canyons with large aspect ratio. The navigation
monitoring technology would be applied in the model verification, which can carry out large-scale observation of
concentration along streets. (4) Here, we considered the dynamic impact of idealized building structure on wind
environment in street canyons. However, there are many other influencing factors, such as building layout and
arrangement, roof shape, green vegetation, and thermodynamic effect, which are suggested to be considered in
future studies. (5) In this study, we mainly focused on the $NO_2$ concentrations. In fact, the concentration of
particulate matter, especially UFP, will also have an obvious peak near the road centerline. In the future, the process
of physical and chemical changes of particulate matter near the vehicle exhaust outlet should be further investigated.
(6) The high resolution $NO_2$ concentration map was benefit for the estimation of human health risks induced by the
air pollution at the street level in future researches.

**Data availability**
Data are available upon request from the corresponding author Huan Liu (liu_env@tsinghua.edu.cn).

**Code availability**
The RF and MARS model for MLSCF are both available on Github (https://github.com/claus0224/MLSCF-RF-
MARS), and other codes are available from the corresponding author on reasonable request.

**Acknowledgment**
We would like to acknowledgment professor Jian Hang from Sun Yat-sen University for supports for CFD
simulations and Dr. Jaime Benavides from Barcelona Supercomputing Center for the application of urban
thermodynamic schemes. This work is supported by the National Natural Science Foundation of China (grant nos.
41822505 and 42061130213 to H.L.), the Tsinghua−Toyota General Research Center. H.L. is supported by the
Royal Society of the United Kingdom through a Newton Advanced Fellowship (NAF\R1\201166).

**Author contributions**

Z. Lv and Z. Luo contributed equally. Z. Lv and Z. Luo designed the research and wrote the manuscript. H.L., Y.Z. and K.H. provided guidance on the research and revised the paper. Z. Lv, Z. Luo, and F.D. provided multiple analytical perspective on this research. X.W., J.Z., and L.X. helped collect and clean the data. T.H. helped on language modification.

**Competing interests**

The authors declare that they have no conflict of interest.

**Additional information**

The supplement is available for this paper at online resources.

Correspondence and requests for materials should be addressed to H.L.

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

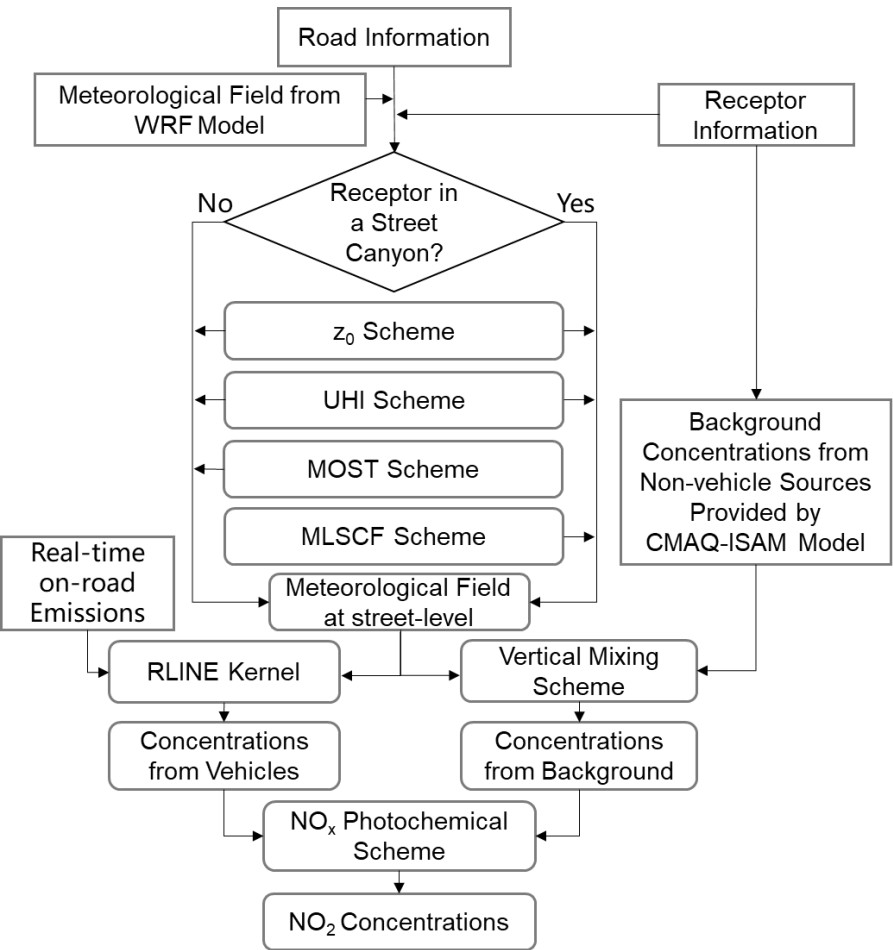


**Figure 1: The framework of multiscale hybrid model CMAQ-RLINE_URBAN.**


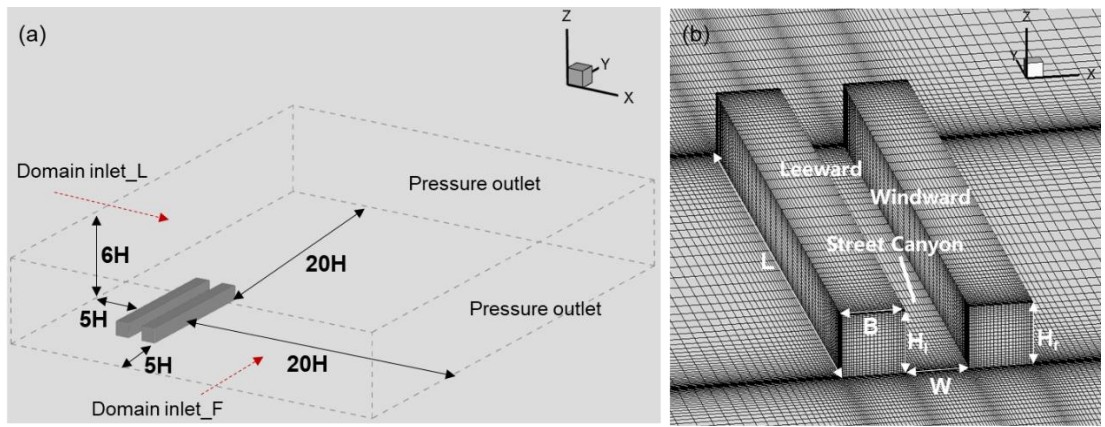

**Figure 2: Computational domain (a) and grid arrangement (b) in all CFD test case.**

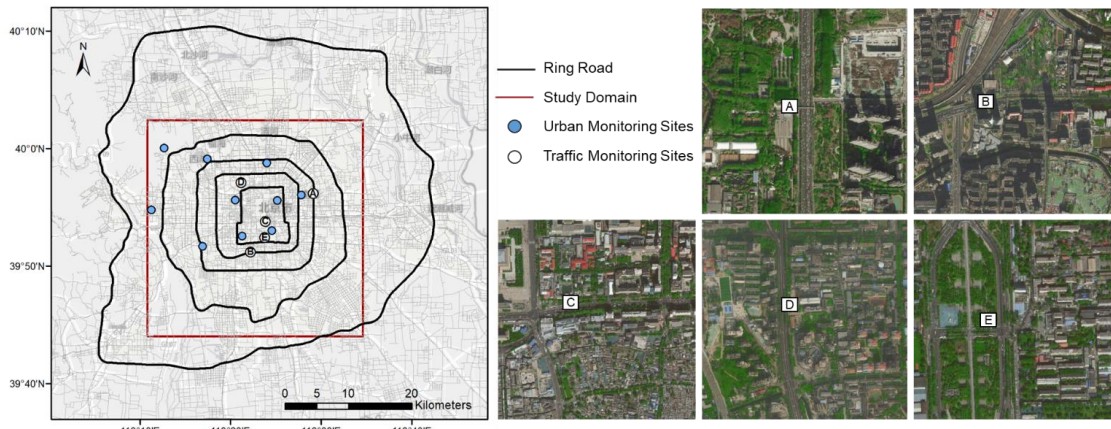

Figure 3: Study domain (© OpenStreetMap contributors 2020. Distributed under the Open Data Commons Open Database License (ODbL) v1.0) and location of monitoring sites (© Microsoft). A. DSH; B. NSH; C. QM; D. XZM; E. YDM.

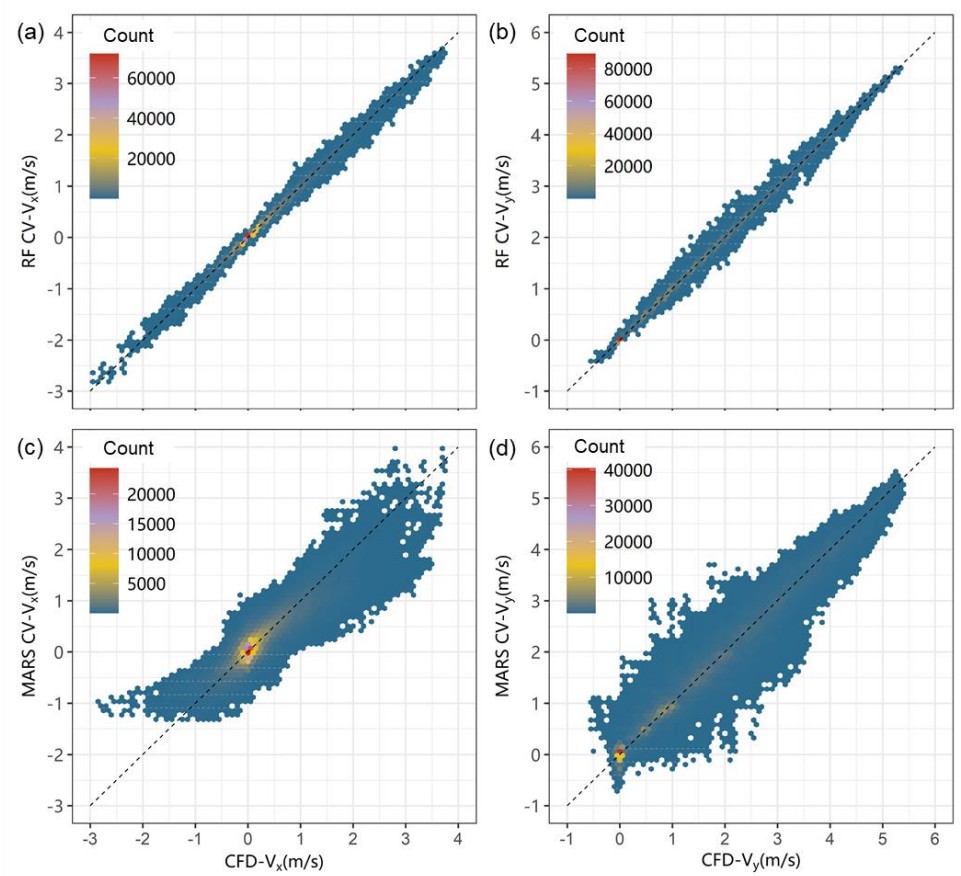

**Figure 4: Cross validations of machine learning models for Vx (a, c) and Vy (b, d): (a)-(b) RF model; (c)-(d)**
**MARS model.**

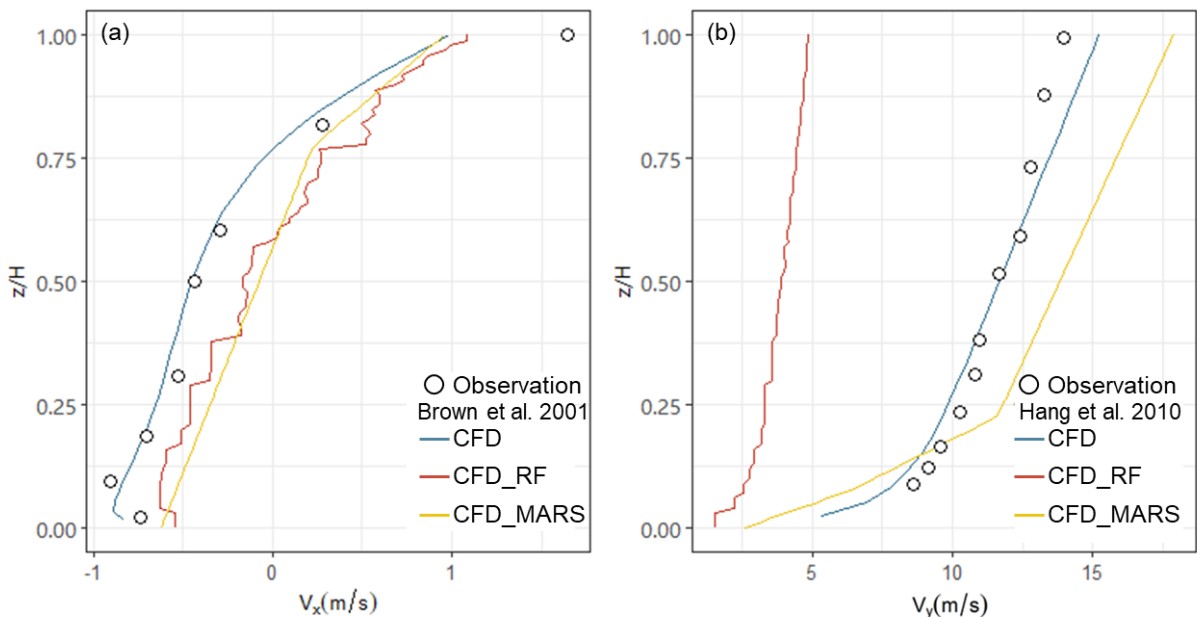

**Figure 5: Performances of machine learning on velocity profile in wind tunnel experiments. The street**
**canyon was perpendicular (a) or parallel (b) to the wind direction at the roof level in different experiments.**
**The detailed description of each experiment was introduced in Section 2.2.3.**


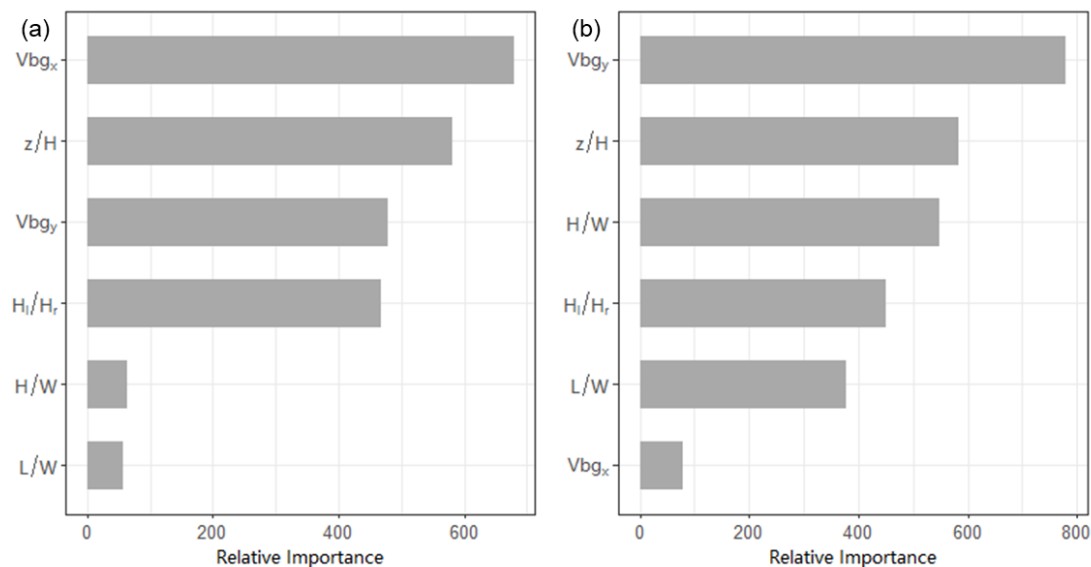

**Figure 6: Variable importance ranking in the RF model for (a) $V_x$ and (b) $V_y$.**

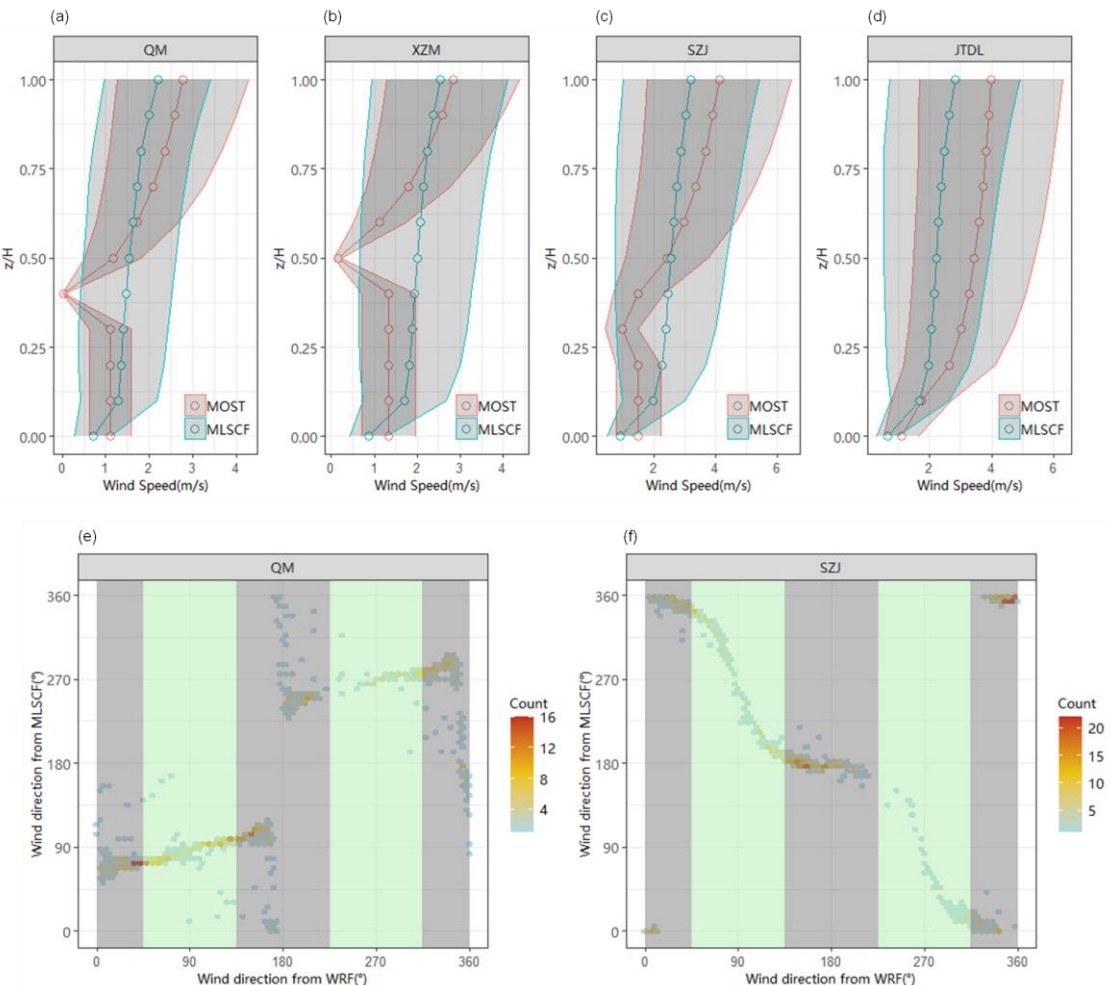

**Figure 7: Influence of MLSCF on wind filed in the street canyon. Monthly averaged vertical profile of wind speed from MOST and MLSCF method in different street canyons: (a) QM (H/W=0.22); (b) XZM (H/W=0.35); (c) SZJ (H/W=1); (b) JTDL (H/W=1.93). The gray shade represents the standard deviation in results of all hours. Hourly wind direction from WRF model (at roof level) and MLSCF method (at ground level) in different street canyons: (e) QM (H/W=0.22); (f) SZJ (H/W=1). As the gray and green shade shown, the background wind over the street canyon provided by WRF model was divided into four main directions: east, west, south and north.**

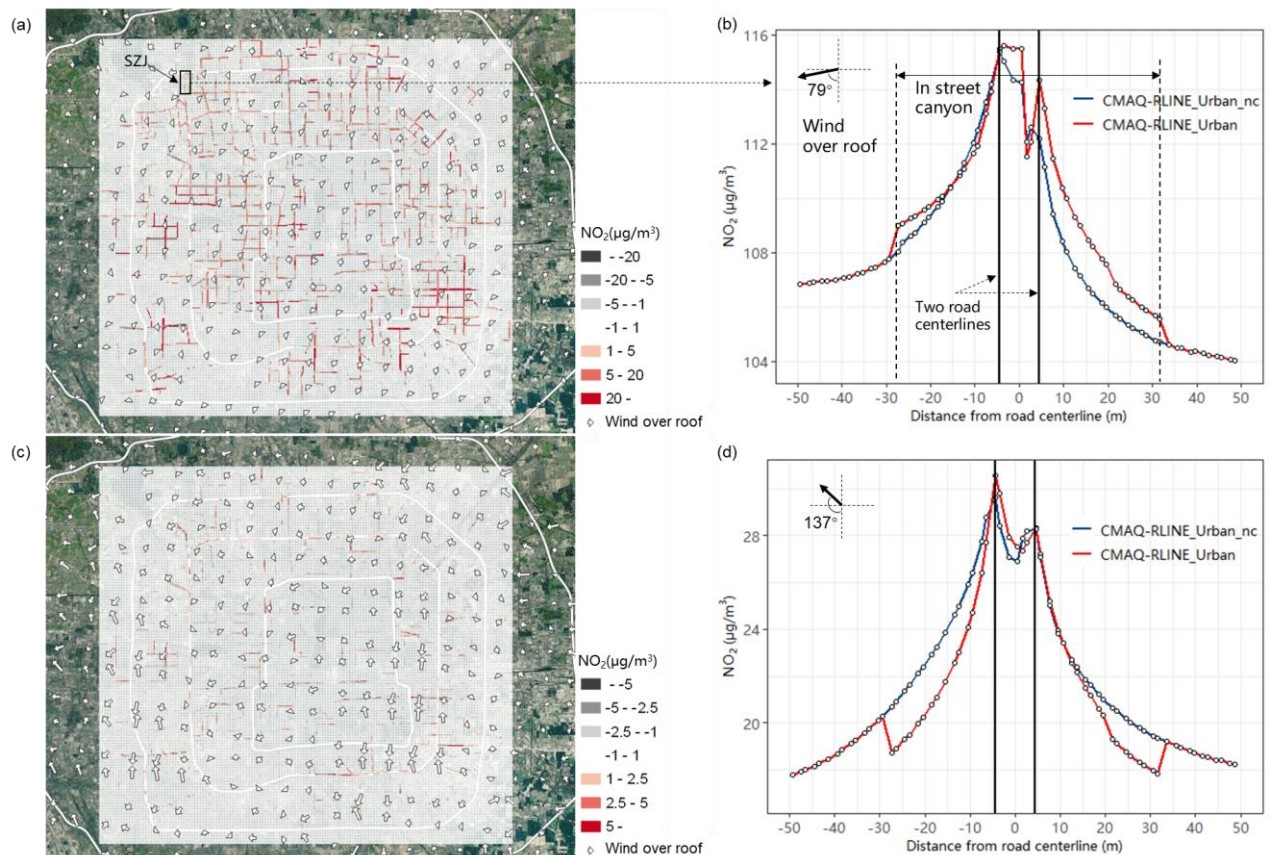


**Figure 8: Differences in NO₂ concentrations at the height of 1.5 m impacted by MLSCF scheme (a, c) over**

**the study domain (CMAQ-RLINE_URBAN - CMAQ-RLINE_URBAN_nc) (© Microsoft) and (b, d) near**

**SZJ in 2019-08-24 at 0:00-1:00 (a, b) and 10:00-11:00 (c, d).**


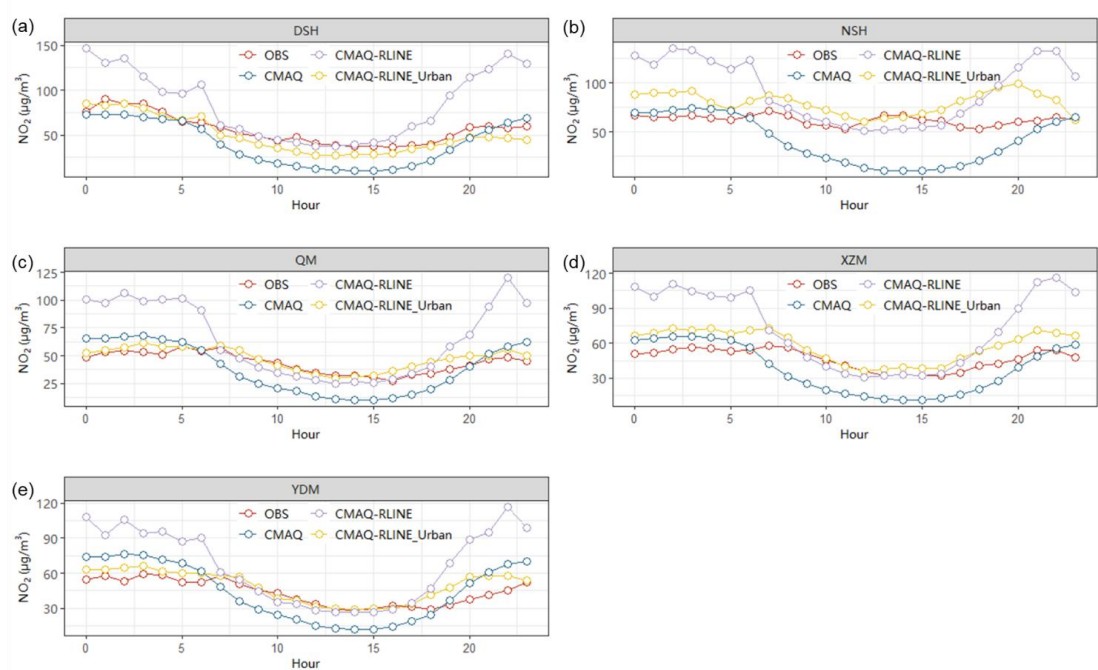


**Figure 9: Diurnal variations of observed and predicted hourly averaged NO₂ concentrations from different**

**models at near-road monitoring sites: (a) DSH; (b) NSH; (c) QM; (d) XZM; (e) YDM.**


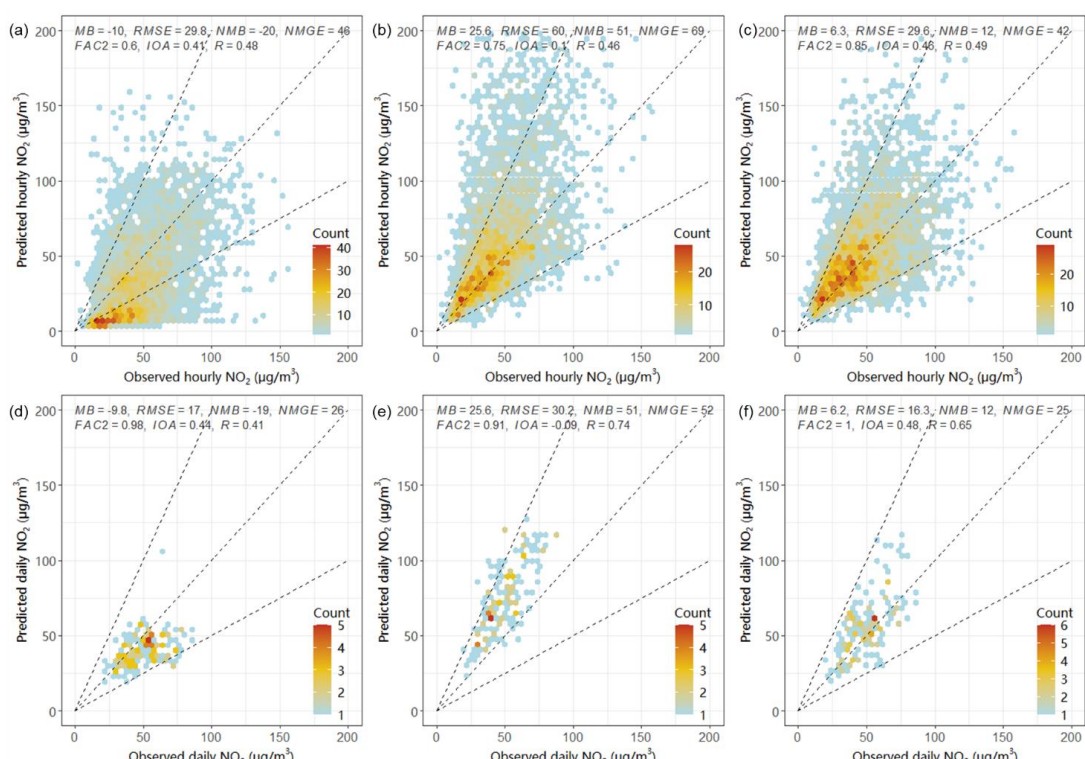


**Figure 10: Observed and predicted hourly (a-c) or daily averaged (d-f) NO2 concentrations from different**
**models at near-road sites: (a, d) CMAQ model; (b, e) CMAQ-RLINE model; (c, f) CMAQ-RLINE_URBAN**
**model.**

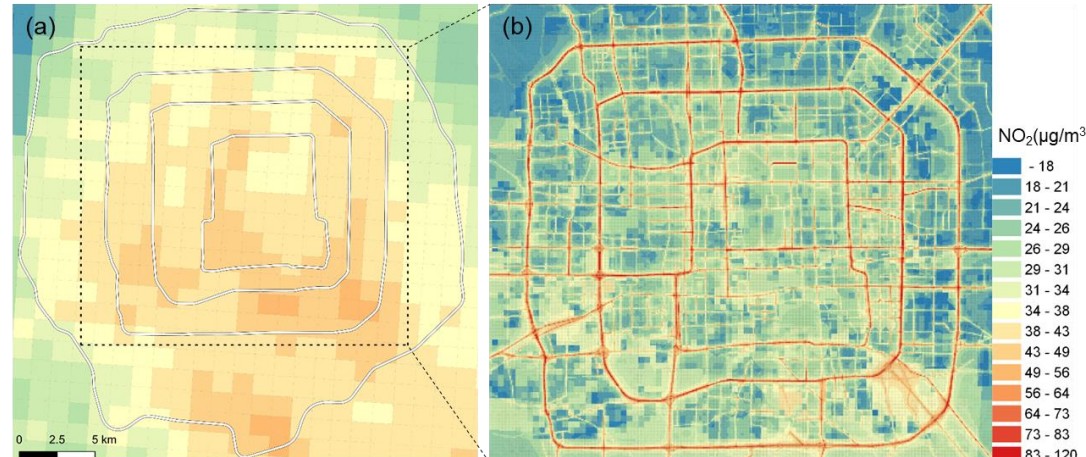

**Figure 11: Spatial distribution of monthly averaged NO₂ concentrations from (a) CMAQ model and (b) CMAQ-RLINE_URBAN model (© OpenStreetMap contributors 2020. Distributed under the Open Data Commons Open Database License (ODbL) v1.0).**

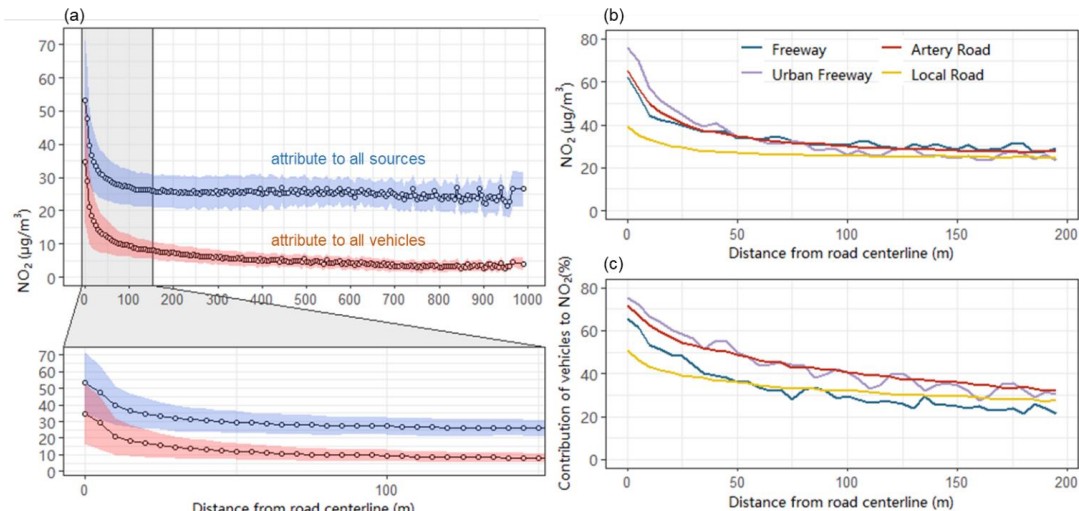

Figure 12: Monthly averaged $NO_2$ concentrations attributed to all emission sources or vehicles with distance from the receptor to its nearest road centerline. (a) $NO_2$ attributed to all emission sources near all roads; (b) $NO_2$ attributed to all emission sources near different road types; (c) Relative contribution of vehicles to $NO_2$ near different road types. The shade area in (a) represents the standard deviation in results of all receptors.

858 **Table 1: Values of controlling factors used in the simulations.**

| Controlling factor | Value | | | | |
|---|---|---|---|---|---|
| $H_l/H_r$ (unitless) | 0.50 | 0.75 | 1.00 | 1.33 | 2.00 |
| $H/W$ (unitless) | 0.25 | 0.50 | 1.00 | 2.00 | - |
| $L/H$ (unitless) | 3 | 5 | 10 | 20 | - |
| $V(H)$ (m/s) | 1 | 2 | 3 | 4 | 5 |
| $\alpha$ (°) | 0 | 30 | 60 | 90 | - |

859
860

**Table 2: Model performances under different scenarios**

| Sites | Scenario | MB | RMSE | NMB | NMGE | FAC2 | IOA | *R* |
|-------|----------|-----|------|-----|------|------|-----|-----|
|       | CMAQ | 3.1 | 25.6 | 9 | 53 | 0.65 | 0.45 | 0.52 |
| All   | CMAQ-RLINE | 18.5 | 46.6 | 53 | 77 | 0.67 | 0.19 | 0.55 |
|       | CMAQ-RLINE_URBAN | 4.6 | 25.8 | 13 | 49 | 0.75 | 0.49 | 0.57 |
|       | CMAQ | 8.0 | 24.3 | 27 | 58 | 0.68 | 0.40 | 0.59 |
| Urban | CMAQ-RLINE | 12.3 | 35.8 | 43 | 76 | 0.64 | 0.20 | 0.50 |
|       | CMAQ-RLINE_URBAN | 1.3 | 23.1 | 4 | 51 | 0.71 | 0.47 | 0.49 |

*MB: Mean bias; RSME: Root mean squared error; NMB: Normalized mean bias; NMGE: Normalized mean gross error; FAC2: Fraction of predictions within a factor of two; IOA: Index of agreement; R: correlation coefficient.