# Peer review of "Development and application of a multi-scale modelling framework for"

_Atmospheric Chemistry and Physics, 2022_

## Referee Comment (RC3)

**General comments:**

In this work a hybrid model has been developed and evaluated to analyse the effects of vehicle emissions on urban roadside concentrations of NO2 in Beijing. The article is well written and raises an important topic, the link between the simulations done using regional chemistry transport models and the simulations at urban level done using gaussian/dispersion models. However, there are few points that should be clarified in order to make clearer the evaluation of the model and the scenarios tested.

**Major comments:**

- The introduction clearly shows the differences between chemistry transport models and dispersion/gaussian models highlighting the difficulties of the former in predicting the roadside concentrations. However, there isn't a clear link between regional models and urban models. Few works have been published and few models have been already developed to couple regional and urban models and these should be mentioned in the introduction

- The methodology highlights only part of the process defined in Figure 1. The authors focus their discussion on the urban model but WRF and CMAQ configuration and outputs should also mentioned and discussed.

- The simulations are run for a period of high photochemical activity. This is surely dependent by weather conditions that are completely absent from the article.

- The NOx-O3 system include also VOCs. There is no mention of this in the methodology or in the results. Are VOCs included in the simulations? I t would be good to add the chemical mechanism somewhere in the supplementary material.

**Minor comments:**

*Line 32:* the reference (Cui et al., 2021), (Shah et al., 2020) should be (Cui et al., 2021; Shah et al., 2020)

*Line 33*: delete that: in is still much more severe than  in developed

*Line 40 – 42*: The comparison with the emission in Lyon is quite specific. I suggest to explain a bit more or in alternative to make a more general case of "other urban areas".

*Line 102:* the spatial resolution should be precise: please substitute < 100 m x 100 m with the real spatial resolution.

*Line 107 -108*: The choice of the midpoint height 22.5m suggests that the CMAQ has a first vertical layer at 45m of height. In first instance this would be in my opinion too high. Generally, CTMs has the first 9-10 vertical layers below the boundary layer but, to improve the prediction on the ground level, keep the 1st layer around 20m from the ground. Could the authors enforce their statements with one or more references that justify this choice?

*Line 277 – 279*: The authors describe the performance of the model in terms of "high" and "low" RE. It would be good to provide a more quantitative description or a reference value for this particular metric.

*Line 280 – 283*: I'm not sure that the MARS model performs better than RD in Figure 5. I suggest to clarify this paragraph better. In the a) figure the CFD is the closest to the observations, followed by RF

(red slope) and MARS (yellow slope). In figure b) again CFD is the closest to the observations, RF is completely underestimated and MARS overestimated from values of z/H > 0.25

*Line 378 – 383*: The authors mentions NOx emissions leading to high NO2 observations among all sites. They also say that CMAQ model underestimate the NO2 concentrations near ring roads (MB = -15μg/m3). The NOx emissions account for NO+NO2, if this variable is NOx before to be inserted in CMAQ, it has to be divided between NO and NO2. In roadside sites generally the NO emissions are high, could the underestimation in CMAQ related a not precise division of the original emissions of NOX in NO and NO2?

*Line 390 – 394*: The model actually improves the performance of O3 in comparison with CMAQ only model. This agrees with the underestimation in NO2 that the CMAQ only shows and previously described by the authors. Being the O3 chemistry dependent not only by NO, NO2 but also by VOCs I would spend some words to introduce these pollutant class and give more details on them.

---

## Author Response (AR1)

**Authors' Response:**

**Manuscript Title**: Development and application of a multi-scale modelling framework for urban high-resolution NO$_2$ pollution mapping
**Discussion Link:** https://acp.copernicus.org/preprints/acp-2022-371/#discussion

**Revision notes:**

> Reviewers' comments are in blue italic type.
>
> Authors' responses are in indent and in black normal font.
>
> Revisions in the manuscript are in indent red normal font.

**Content**

**Response to Reviewers #1's Comments**
*Summary*

The study of Lv et al. presents a multi-scale modelling framework for the simulation of urban scale NO2 and potentially other primary pollutants at high spatial resolution with a focus on traffic-related air pollution. The method combines several different types of models and approaches, a regional chemistry-transport-model (CMAQ), a dispersion model (RLINE), an urban heat island scheme, and machine-learning based simulation of street-canyon flows trained with a CFD model. The overall framework is referred to as CMAQ-RLINE_URBAN.

The overall approach is interesting, but the publication has major deficiencies, is difficult to follow, and leaves many questions unanswered. In my view it cannot be published in the present form but will need substantial improvements.

**Response:**

Thank you very much for spending time to give us many constructive comments, and they have great importance in improving our manuscript. We have revised our manuscript and we believe that all the concerns are now fully addressed in this revision. In general, as you suggested, a more detailed review on multi-scale air quality models was added in the Introduction, and more descriptions on various parameterization schemes were provided in the Method and Supplement Materials. In addition, more explanations and expectations in the future were added in the Results and Discussions, respectively.

*Major comments*
*Question 1*

The individual model components as well as their interplay are very poorly described. Examples:

The RLINE model is never explained. It remains unclear whether this is Gaussian dispersion or any other type of model. Providing only references without any further details is not sufficient given the fact that this model plays a central role in this study.

How are emissions released into the model? Is traffic a line source? How are the emissions transported forward and dispersed by the (RLINE) model?

**Response:**

Thanks for your advice. We apologized that the introduction of RLINE was missing in our

original manuscript so that the details of our model is not sufficient. In general, the RLINE is a Gaussian dispersion model specially for the line source simulation. In this revision, we have added the description of RLINE model including its mechanism and application in **Section 2.1.**

The traffic emission is treated as a line source in RLINE. The concentration from the traffic emission is found by approximating the line as a series of point sources and integrating the contributions of point sources using an efficient numerical integration scheme.

**Revisions in Manuscript:**

**(1) Materials and Methods, Line 119-128.**

RLINE is a Gaussian line source dispersion model developed by Snyder et al. (2013) to predict pollutant concentrations in near-road environments. In the RLINE model, the mobile source is considered as a finite line source, from which the concentration is found by approximating the line as a series of point sources and integrating the contributions of point sources using an efficient numerical integration scheme. The number of points needed for convergence to the proper solution is a function of distance from the source line to the receptor, and each point source is simulated using a Gaussian plume formulation. The RLINE model performs generally comparable results when evaluated with other line source models for on-road traffic emissions dispersion (Snyder et al., 2013; Heist et al., 2013; Chang et al., 2015), and has been successfully used in many studies to evaluate the impacts from traffic emissions on air quality (Zhai et al., 2016; Valencia et al., 2018; Benavides et al., 2019; Filigrana et al., 2020; Zhang et al., 2021a).

*Question 2*

An UHI scheme is implemented, which "increases atmospheric turbulence intensity around sunset in the afternoon", but it is never explained how this increased turbulence affects the simulation or what is meant by "afternoon". Is the UHI scheme only triggered around sunset? Does it affect the turbulent intensity in RLINE? If so, how exactly? Which localized meteorological parameters are recalculated and how? Is the UHI scheme of Cimorelli et al. (2005) different from the algorithm proposed by Benavides et al., or is the Benavides algorithm based on Cimorelli et al.? The text remains extremely vague despite

the fact that, again, this UHI scheme is an essential component of the final model system. Please note that WRF can be run with an urban canopy module (e.g. Barlage et al., 2016; doi:10.1002/2015JD024450), which would alleviate the necessity of implementing an UHI scheme in such a complicated (and unclear) way as done here. Why was this scheme not used to drive CMAQ and to compute the winds and stability above roof level?

**Response:**

Thanks for your questions. We apologize for misleading the reviewer about the UHI scheme, and the detailed description for UHI scheme were added in this revision. In this response, we will give a response to each question as follows:

- The impacts of UHI effect in the hybrid model is considered not only in the afternoon but also over the whole day. However, based on previous studies, the UHI effect is more significant in the afternoon around sunset.

- The UHI scheme will affect the turbulent intensity in RLINE. First, the upward surface heat flux and the urban boundary layer height due to convective effects was estimated. And then the mixing height $Z_{mix}$, convective velocity scale $w^*$, surface friction velocity $u^*$, and Monin-Obhukov length $L_{MO}$ were recalculated. The UHI scheme was never considered in Benavides's study (Benavides et al., 2019), and in this study, it was built based on the algorithm used in the AERMOD model (Cimorelli et al., 2005). We have also added a brief introduction of **UHI scheme** in the **Section S1. Urban heat island scheme** and details **in Supplement Materials**.

- The WRF model can actually be coupled with urban canopy models (UCMs) to quantify the changes in meteorological conditions caused by special underlying surface structures in cities, where the UHI effect is included. However, the WRF model was applied for mesoscale meteorological simulation (not smaller than an urban scale), so it is too large to consider the impact on each road. The hybrid model in this study mainly focuses on local meteorology at the street level. In addition, if WRF coupled with UCMs and the UHI scheme are both used in the hybrid model, it will cause a double-counting problem of UHI effects.

**Revisions in Manuscript:**

**(1) Materials and Methods. Line 140-145.**

The atmospheric turbulence intensity in urban areas around sunset in the afternoon was obviously enhanced considering the influence of the urban heat island effect based on methods in the AERMOD model (Cimorelli et al., 2005) (UHI scheme). The UHI scheme would affect the turbulent intensity based on the evaluation for the upward surface heat flux and the urban boundary layer height due to convective effects, and then the mixing height, convective velocity scale, surface friction velocity, and Monin-Obhukov length were all recalculated (details in the Supplement Section S1).

**(2) Supplement Materials. Section S1 Urban heat island scheme.**

The Urban Heat Island effect refers to a phenomenon that the temperature of urban atmosphere and surface is higher than that of nearby rural areas, of which intensity can be quantified by using the temperature difference. UHI is caused by the thermodynamic effect of the special underlying surface structure induced by urbanization and the influence of human activities. In past decades, the intensity of UHI in Beijing has been increasing at a rate of 1.35 °C every decade, and has gradually expanded from within the 2nd Ring Road to the 6th Ring Road and its surrounding areas (Ge et al., 2016). When the UHI intensity is high, the circulation between urban and suburban areas will enhance the boundary layer height and turbulence intensity in urban areas, and reduce the concentration of primary pollutants such as $NO_x$ which are easily affected by the local climate. After adding the UHI scheme to the model, the overestimation of the simulation can be reduced, and the simulation is more consistent with the observed concentration (Sarrat et al., 2006).

Here, based on the algorithm used in AERMOD (Cimorelli et al., 2005), we estimated the influence of UHI on turbulence in urban areas, especially in the afternoon (16:00-23:00), to reduce the over-predicted pollutant concentrations caused by the overestimation of atmospheric stability in this period. During this period, due to the large amount of anthropogenic heat generated by transportation, cooking and other human activities, as well as the gradual release of solar radiation stored by buildings in daytime, the UHI intensity in Beijing increase to the peak (Wang et al., 2017). In the calculation, we still regarded each road as a basic unit for the calculation, and first estimated the sensible heat flux $H_{u,UHI}$ (W/m$^2$) caused by UHI and the height of the mixing layer $Z_{mix,c}$ (m) formed by thermal turbulence. And then the mixing height $Z_{mix}$, convective velocity scale $w^*$, surface

friction velocity $u^*$, and Monin-Obhukov length $L_{MO}$ were recalculated, as follows,

$$\begin{cases} H_{u,UHI} = \alpha \rho c_p \Delta T_{u-r} u^* \\ Z_{mix,c} = Z_{mix,ref}(P/P_{ref})^{0.25} \end{cases}$$

where, $\alpha$ is the empirical coefficient, with a value of 0.03. $\rho$ is air density (kg/m³) and calculated by air pressure and temperature. $c_p$ is the specific heat capacity of air at constant pressure, with a value of 1004 J/kg·K. $\Delta T_{u-r}$ is the temperature difference between urban and suburban areas, which is set with the value of 3°C according to the observation of several meteorological ground observation stations and satellite remote sensing data (Wang et al., 2017). $Z_{mix,ref}$ and $P_{ref}$ are the reference boundary layer height and urban population, with values of 400 m and 2 million, respectively (Cimorelli et al., 2005). $P$ is the total population of urban areas in the research region, with the value of 9.2 million in our study domain for 2020 based on the WorldPop dataset (Bondarenko et al., 2020).

**Question 3**

- CFD simulations were performed to train a machine-learning based street-canyon flow model (MLSCF) in order to predict airflow in street canyons efficiently. This part of the publication is quite clear, but how the results of the MLSCF are finally applied to compute the dispersion of NO2 is never explained. It should also be noted that the MLSCF model only predicts wind speeds at different locations in the street canyon (in along-canyon and perpendicular direction), but not turbulence, which also varies depending on wind speeds and angle between wind and canyon. The publication mentions the importance of the buildup of a vortex in certain situations, but it remains unclear how the mixing of air pollution induced by this vortex affects the mixing in RLINE. If RLINE is a simple Gaussian dispersion model, how would it be able to represent such a vortex?

**Response:**

Thanks for questions. The MLSCF is developed to estimate the wind velocity and direction at different height in the street canyon. In other words, outputs from the MLSCF model are wind vectors. Therefore, the impact of turbulence induced by street canyon effect on wind environment is considered. As shown in the Figure 8, the impacts of the MLSCF scheme on simulated $NO_2$ concentration were identified by the differences between modeling scenarios with and without MLSCF. For example, in the SZJ standard canyon,

the application of MLSCF led to the wind direction at the bottom in street canyon opposite to that at the roof, increasing the upwind concentrations (Figure 8b and Section 3.2 in the manuscript).

However, as you mentioned that since the RLINE is a Gaussian model, we cannot directly calculate the impact of vortex in specific situations on the mixing of concentrations in street canyons. We will make effort on developing another empirical method to estimate effects of the vortex on the mixing of concentrations in the future research, and this is discussed in the **Conclusions** now.

**Revisions in Manuscript:**

**(1) Conclusion and Discussions, Line 505-507.**

However, the influence of the turbulence induced by street canyon effects on the mixing of air pollution was not considered on which we will make effort in the future.

*Question 4*

NOx is released by emission sources mainly in the form of NO and then converted to NO2 by reaction of O3. The paper mentions that a "two-reaction scheme" was incorporated into RLINE, but it is not explained which photochemical reactions exactly were considered, how this reaction scheme was implemented in RLINE, or in what form NOx was emitted. Figure 1 suggests that concentrations from vehicles and from background are combined within the "NOx photochemical scheme". Is the scheme applied separately to the two components? How exactly are they combined?

**Response:**

Thank you for questions. We apologized those unclear descriptions on the NOx photochemical reactions in the original manuscript. In general, we used the two-reaction method applied in other studies, such as the SIRANE model (Soulhac et al., 2017). The $NO_x$ photochemical scheme includes two main chemical reactions, namely the photolysis of $NO_2$ and the oxidation of NO as follows:

$$\begin{cases} NO_2 + h\nu \rightarrow NO + O_3 \\ NO + O_3 \rightarrow NO_2 \end{cases}$$

During simulation, the NOx ($NO+NO_2$) emitted from vehicles is first regarded as an inert gas and only the primary concentration after diffusion is simulated. Then, assuming a

photo-stationary equilibrium condition, the concentrations of NO, $NO_2$ and $O_3$ are calculated as follows:

$$\begin{cases} [NO_2] = (b - \sqrt{b^2 - 4c})/2 \\ [NO] = [NO]_b + [NO_2]_b + [NO_x]_d - [NO_2] \\ [O_3] = [O_3]_b + [NO_2]_b + \zeta[NO_x]_d - [NO_2] \\ b = k1/k2 + [O_3]_b + [NO]_b + 2[NO_2]_b + (1 + \zeta)[NO_x]_d \\ c = ([O_3]_b + [NO_2]_b + \zeta[NO_x]_d)([NO]_b + [NO_2]_b + [NO_x]_d) \end{cases}$$

where, $[NO_x]_d$ is the primary concentration of $NO_x$ directly simulated by RLINE model when taken as an inert gas. $[NO]_b$, $[NO_2]$, and $[O_3]_b$ are the background concentrations of NO, $NO_2$ and $O_3$ from non-vehicle sources, respectively, which are provided by CMAQ-ISAM model. $\zeta$ is the ratio of $NO_2$ to $NO_x$ in vehicle emissions, with a value of 0.2 (Benavides et al., 2019; Valencia et al., 2018).

We have added a brief introduction of the "two-reaction scheme" in **Materials and Methods** and details in **Supplement Materials (Section S3. NOx photochemical parameter scheme)**.

**Revisions in Manuscript:**

**(1) Materials and Methods. Line 167-170.**

In this study, a simplified two-reaction scheme, including the photolysis of $NO_2$ and the oxidation of NO, was incorporated into the model to characterize the photochemical process of $NO_x$ (details in SI. Section S2), which has been successfully applied to the SIRANE dispersion model (Soulhac et al., 2017).

**(2) Supplement Section S3. NOx photochemical parameter scheme.**

The $NO_x$ photochemical parameter scheme applied in this study includes two reactions:

$$\begin{cases} NO_2 + h\nu \rightarrow NO + O_3 \\ NO + O_3 \rightarrow NO_2 \end{cases}$$

Kim et al. compared two-reaction scheme with CB05 gas phase chemical mechanism by incorporated them into SinG model to estimate roadside $NO_2$ concentration, and found a similar results, while the computing time cost of two-reaction scheme was significantly less than that of the CB05 mechanism (Kim et al., 2018). Therefore, the simplified two-reaction scheme was incorporated into the model in this study to characterize the $NO_x$ photochemical process. During simulation, the NOx ($NO+NO_2$) emitted from vehicles is first regarded as an inert gas and only the primary concentration after diffusion is simulated.

Then, assuming a photo-stationary equilibrium condition, the concentrations of NO, $NO_2$

and $O_3$ are calculated using the two-reaction scheme, as follows:

$$\begin{cases} [NO_2] = (b - \sqrt{b^2 - 4c})/2 \\ [NO] = [NO]_b + [NO_2]_b + [NO_x]_d - [NO_2] \\ [O_3] = [O_3]_b + [NO_2]_b + \zeta[NO_x]_d - [NO_2] \\ b = k1/k2 + [O_3]_b + [NO]_b + 2[NO_2]_b + (1 + \zeta)[NO_x]_d \\ c = ([O_3]_b + [NO_2]_b + \zeta[NO_x]_d)([NO]_b + [NO_2]_b + [NO_x]_d) \end{cases}$$

where, $[NO_x]_d$ is the primary concentration of $NO_x$ directly simulated by RLINE model

when taken as an inert gas. $[NO]_b$, $[NO_2]$, and $[O_3]_b$ are the background concentrations of

NO, $NO_2$ and $O_3$ from non-vehicle sources, respectively, which are provided by CMAQ-

ISAM model. The unit of concentrations in these formulas is $mol/m^3$. $\zeta$ is the ratio of $NO_2$

to $NO_x$ in vehicle emissions, with a value of 0.2 (Benavides et al., 2019; Valencia et al.,

2018). The reaction rates of the photolysis of $NO_2$ and the oxidation of NO were set to be

$k1$ and $k2$ respectively, and calculated as follows (Hurley, 2005):

$$\begin{cases} k1 = 10^{-4} \times \delta \times \text{TSR} \\ k2 = 9.24 \times 10^5 \times \exp(-1450/T)/T \end{cases}$$

$$\delta = \begin{cases} 4.23 + 1.09/\cos Z, & 0 \le Z \le 47 \\ 5.82, & 47 < Z \le 64 \\ -0.997 + 12(1 - \cos Z), & 64 < Z \le 90 \end{cases}$$

where, all parameters were from the WRF model. TSR is the total solar radiation ($W/m^2$).

$Z$ is the solar zenith angle (°). $T$ is the ambient temperature (K).

**Question 5**

A "vertical mixing scheme" is mentioned on page 6, which accounts for the "influence of

atmospheric turbulence and building geometry on the vertical mixing" and seems to mix

background air from roof level into the street canyons. The scheme requires wind speeds

at the surface and at roof level, but it is not entirely clear which winds are used here. From

the MLSCF scheme? What is the motivation for using the ratio between wind speeds at

roof level and street level to compute the contribution of background air? I can only guess,

but decisions like this need to be motivated thoroughly.

**Response:**

Thanks for your suggestions. For motivation of this scheme, since the settings of vertical

pressure layer in the CMAQ and the WRF model are the same, the concentrations from

non-vehicle sources provided by the CMAQ-ISAM model are regarded as the background concentration at the top of the urban canopy layer. If the influence of turbulence changes on the mixing of background concentration is not taken into account, the pollutant concentrations near surface at night stable boundary layer is easy to be significantly overestimated (Benavides et al., 2019).

In this scheme, the wind speed at the roof level is from the WRF model. The surface wind is from MLSCF scheme when the gird receptor is located in the street canyon, and otherwise the logarithmic wind profile is used to calculate the wind speed at the specified height. We have added a brief introduction of the "vertical mixing scheme" in **Materials and Methods** and details in **Supplement Materials (Section S2. Vertical mixing scheme)**.

**Revisions in Manuscript:**

**(1) Materials and Methods. Line 163-165.**

In this scheme, the surface wind was from MLSCF scheme when the gird receptor is located within the street canyon, and otherwise the logarithmic wind profile was used to calculate the wind speed at the specified height, and details were showed in the Supplement Section S1.

**(2) Supplement Section S2. Vertical mixing scheme.**

Since the settings of vertical pressure layers in the CMAQ and the WRF model are same, the concentrations induced by non-vehicle sources provided by the CMAQ-ISAM model can be regarded as the background concentration at the top of the urban canopy layer (UCL). If the influence of turbulence changes on the mixing of background concentration is not taken into account, the pollutant concentration at night stable boundary layer is easy to be significantly overestimated (Benavides et al., 2019). Therefore, we assumed that the concentration relationship between the top of UCL and the near surface is affected by atmospheric stability, local street canyons and building morphology.

In this study, based on the method proposed by Benavides et al. (2019), the ratio of wind speed between the near surface of the road and the top of surrounding buildings was used as a proxy parameter in the model to characterize the turbulence intensity which affects the vertical concentration mixing between the top of UCL and near surface. However,

Benavides et al. assumed that the average wind speed in the street canyon was proportional to the angel between the top wind direction and the central axis of the road, and the logarithmic wind profile to was still used to represent the change of wind speed within UCL, resulting in the influence of the street canyon effect on vertical mixing of background concentration was not considered. In this study, when the grid receptor is located in the street canyon, the MLSCF scheme was used to describe the wind profile within UCL. Otherwise, the logarithmic wind profile was used to calculate the wind speed at the specified height. This parameter scheme mainly calculated the background concentration mixing ratio ($fac_{bg}$), which was multiplied by the background concentration provided by the CMAQ-ISAM model to estimate the background concentration at the specified height near the ground. Based on the estimated sensible heat flux ($H_u$, W/m$^2$) from the WRF model, convective boundary layer ($H_u > 0$) and stable boundary layer ($H_u < 0$) were distinguished, and the effect of building density around the receptor site on $fac_{bg}$ was also considered, as follows:

$$
fac_{bg} = \begin{cases}
1 - F + F \times \dfrac{WS_{sfc}}{WS_{bh}}, & bd > 0.1 \& H_u > 0 \\[2mm]
\dfrac{WS_{sfc}}{WS_{bh}}, & bd > 0.1 \& H_u \leq 0 \\[2mm]
1 - 5bd + 5bd \times \dfrac{WS_{sfc}}{WS_{bh}}, & bd \leq 0.1 \& H_u > 0 \\[2mm]
1 - 10bd + 10bd \times \dfrac{WS_{sfc}}{WS_{bh}}, & bd \leq 0.1 \& H_u \leq 0
\end{cases}
$$

where, $F = m + abs(0.25 - bd)$, where $m$ is an empirical parameter with value of 0.1.

**Question 6**

Why was a resolution of 50 m x 50 m chosen? Note that in Section 2.1 it is suggested that the resolution is only 100 m x 100 m. As mentioned on line 152, the average width of streets in Beijing is about 50 m. Thus, a resolution of 50 m is by far not sufficient to resolve gradients within street canyons.

**Response:**

Thanks for your reminding, and the original description was misleading. The grid spatial resolution of the hybrid model was 50 m x 50 m rather than 100 m x 100 m. This grid resolution over the whole urban area is limited due to the long computing time at present.

Since the addition parameterization schemes applied into the hybrid model, especially for the MLSCF, the meteorological field of each street needs to be calculated separately, leading to the large computational burden. However, when we focus on the distribution of concentration gradient near the street, the resolution of grid receptors will be improved to several meters level. For example, in Figure 8b and d, the grid resolution near SJZ street was improved to be 2 m. We will make efforts to develop a parallel computing method to reduce the computing time, in order to improve the grid resolution of a relatively large-scale simulation. We have modified the writing and added this discussion in **Conclusions**.

**Revisions in Manuscript:**

**(1) Materials and Methods. Line 116-117.**

In our model, a $NO_2$ pollution map with a high temporal (1 h) and spatial resolution (50 m×50 m) can finally be obtained.

**(2) Conclusion and Discussions. Line 519-522.**

At present, considering the running cost, the grid resolution of area in Beijing 5th ring road and its surroundings can reach 50 m×50 m. We will make efforts to develop a parallel computing method to reduce the computing time, in order to improve the grid resolution of a relatively large-scale simulation.

*Question 7*

The machine-learning model is rather simple and little convincing. Complex models are often replaced by artificial intelligence methods using neural networks or Gaussian process models, see for example Beddows et al. (2017, doi:10.1021/acs.est.6b05873). A good summary of methods applied in the context of air quality simulations is presented in Conibear et al. (2021, doi: doi.org/10.1029/2021GH000391). Here, a random forest (RF) regression and a MARS approach are used, but these choices are not motivated at all. The RF approach seems to generate quite noisy wind profiles (see Figure 5), but in most cases performs better than MARS The combination of RF and MARS is referred to as "ensemble learning", but according to page 11, there RF and MARS models have been trained completely independently and there is only a simple switch between the two methods depending on whether the input values are within the range of the predictors used in the

training or not. There is a long way from such a simple approach to "ensemble learning".

**Response:**

Thanks for your suggestions. As your suggested, the applications of machine learning models on air quality predictions were investigated. In general, Random Forest (RF) and Multivariate Adaptive Regression Splines (MARS) are common machine learning methods which run efficiently on large data sets, and are relatively robust to outliers and noise. Furthermore, compared with other models (e.g. ANN), RF and MARS never require the specification of underlying data model and the complex parameter tuning (Kühnlein et al., 2014), and they can still provide efficient alternatives and generally show a high accuracy in many applications of predicting air pollutant concentrations (Chen et al., 2018; Geng et al., 2020; Hu et al., 2017; Kamińska, 2019). Therefore, RF and MARS are selected in this study, and the validation results with a little deviation (R was 0.99 and median of RE was less than 10%) indicated a good performance of our model (details see Section 3.1). A brief review of the applications of machine learning models, and the advantages of RF and MARS models have been both discussed in the revised Materials and Methods.

As for another question about the "ensemble learning" used in the manuscript, we agreed that we just combined the results of two different models depends on whether the input value was within the range of predictors or not. We realized that it is not appropriate to name this kind of combination as "ensemble learning", so the description about our model has been modified in the manuscript.

**Revisions in Manuscript:**

**(1) Abstract. Line 16-17.**

A Machine Learning-based Street Canyon Flow (MLSCF) scheme was constructed based on Computational Fluid Dynamic and two machine learning methods.

**(2) Introduction. Line 101-103.**

We developed a Machine Learning-based Street Canyon Flow (MLSCF) parameterization scheme, which was based on two machine learning methods using wind data from 1,600 CFD simulations.

**(3) Materials and Methods. Line 258-272.**

Data driven method, such as machine learning and deep learning, is now a successful

operational geoscientific processing schemes and has co-evolved with data availability over the past decade (Reichstein et al., 2019). Specially, these models have been used as computationally efficient emulators of explicit mechanism models, to explore uncertainties (Aleksankina et al., 2019) and sensitivities or replace complex gas-phase chemistry schemes (Keller and Evans, 2019; Conibear et al., 2021). In addition, meta-models (Fang et al., 2005) such as neural networks and Gaussian process (Beddows et al., 2017) are also used to produce a quick to run model surrogate and show reliable performance. Random Forest (RF) model algorithm is an ensemble learning method that generates many decision trees and aggregates their results, which has been developed to solve the high variance errors typical of a single decision tree (Breiman, 2001). Multivariate Adaptive Regression Splines (MARS) is a nonparametric and nonlinear regression method, which can be regarded as an extension of the multivariate linear model (Friedman, 1991). RF and MARS are common machine learning methods which run efficiently on large data sets, and are relatively robust to outliers and noise. Furthermore, they never require the specification of underlying data model and the complex parameter tuning, and they can still provide efficient alternatives and generally show a high accuracy in applications for predict air pollutant concentrations (Hu et al., 2017; Chen et al., 2018; Kamińska, 2019; Geng et al., 2020).

**Question 8**

The introduction section does a fairly poor job in citing relevant literature. Quite many multi-scale air pollution models have been developed recently and also machine learning methods are increasingly used. It is important to place the present study in context and explain where it is different or better than other approaches.

**Response:**

Thanks for your comments. As mentioned in the above response, a brief review of the applications of machine learning models has been added in the Materials and Methods. We have also added a review on multi-scale model in the Introduction. Compared with previous studies, the innovation of our model lies in its comprehensiveness, which takes the influence of street canyons, the chemical process, and background schemes into

consideration. In addition, the MLSCF scheme built here is a machine learning based scheme which is suitable for a wide range of street canyon wind environment simulation without huge computational cost.

**Revisions in Manuscript:**

**(1) Introduction. Line 84-96.**

Considering the respective strengths and limitations of regional models and local models, several studies have been carried out on coupling of air quality models applicable to different scales (Ketzel et al., 2012; Stocker et al., 2012; Lefebvre et al., 2013; Jensen et al., 2017; Kim et al., 2018; Mallet et al., 2018; Hood et al., 2018; Benavides et al., 2019; Kamińska, 2019; Mu et al., 2022). Although these models performed accurately in near-road simulation, the influence of street canyons is still hard to be considered. In some hybrid models (Stocker et al., 2012; Jensen et al., 2017; Mallet et al., 2018), OSPM was still applied to calculate concentration levels within the street, where the application of logarithmic wind profile probably overestimated the bottom wind speed in a deep street canyon as abovementioned. Other models simply assumed that in street canyons, wind direction followed the street direction, and wind speed was uniform, which was not sufficient to resolve the concentration gradient within street canyons ( Kim et al., 2018; Benavides et al., 2019). Berchet et al. (2017) proposed a cost-effective method for simulating city-scale pollution taking advantage of high-resolution accurate CFD, while the primary $NO_x$ was predicted due to the lack of a chemical module. Therefore, it is essential to build an integrated model to predict long-term and near-road air pollution suitable for the urban complex underlying surface environment.

*Minor comments*
*Question 9*

I was confused by the usage of the term "receptor". It seems that a receptor can be a grid point but it can also be any other point in the domain, e.g. the location of a measurement station. This needs to be explained much more clearly and earlier in the manuscript. Note that receptor modelling has quite a distinct meaning in air quality modelling and is usually associated with source-apportionment modeling like chemical mass balance or positive

matrix factorization.

**Response:**

Thanks for your reminding. As you mentioned, the term "receptor" in our manuscript referred to the location where the concentration was predicted by the model. The receptors included both grid receptors and monitor receptors. The grid receptors were set at a spatial resolution of 50 m×50 m, and the monitor receptors were 10 observation stations located in the normal urban environment and 5 near-road monitoring sites. We have revised the writing in the manuscript to avoid misleading. Due to too many revisions, these revisions are not shown in this response. **Please see the word with red color in the manuscript**.

*Question 10*

The workflow illustrated in Figure 1 is not entirely clear to me: First of all, the arrow between the boxes "receptors in street canyon?" and "receptor information" likely points in the wrong direction. The most confusing thing is that there is a distinction between "Is a street canyon" and "Receptor in a street canyon". How is it possible that a point can be in a street canyon and at the same time not be inside? Why is there only "road information" needed as input to decide whether we are in a street canyon or not? Shouldn't there also be 3D building data? How the first decision "is a street canyon" is applied is not clear to me at all. Do you choose a road segment and then decide if it is inside a canyon or not? What about points between roads? How do you decide to which road a given point in the city belongs? What about other areas of the city without roads, e.g. parks?

**Response:**

Thank you very much for suggestion. We apologize for misleading the reviewer about the workflow figure. The box "Is a street canyon" was actually not correct in the workflow and removed in this revision. However, the arrow between the boxes "receptors in street canyon?" and "receptor information" points is in the right direction. In fact, the first criterion is "Receptor in a Street Canyon?", which is depended on the receptor information (e.g. coordinates of the receptor) and road information (e.g. coordinates and geometry parameters). The 3D building data was processed into the geometry parameters of each road segment as stated in the Section 2.2.1. The coordinates and road width were used to

decide whether the receptor is within a street canyon or not. If the receptor is not located within a street canyon, it will be regarded as located in the open terrain area, where the logarithmic wind profile will be used rather than MLSCF scheme. Now the corrected workflow is shown in Figure 1 in the revised manuscript.

**Revisions in Manuscript:**

**(1) Figures.**

[Figure]

Figure 1: The framework of multiscale hybrid model CMAQ-RLINE_URBAN.

*Question 11*

At many instances in the paper, references to figures, tables and other sections are made in past tense (".. was shown in Figure 1", ".. were discussed in the following section", etc.) but should be in present tense (" .. are shown in Figure 1", " .. are discussed in the following section", etc.)

**Response:**

Thanks for your advice. We have corrected the tense in **Results.** Due to too many revisions,

these revisions are not shown in this response. **Please see the word with red color in the manuscript**.

*Question 12*

Data and code availability: Both are only available upon request. Code is only available upon "reasonable request". What is reasonable? Why is the code not made accessible more easily? Advancements in science critically depend on open science and open data.

**Response:**

Thanks for your question. We agreed that advancements in science critically depend on open science and open data. Now the code of MLSCF scheme is open to the public. We have added the expressions and coefficients of MARS model in the **Supplement Materials (Table S2 and S3)**, and the code of both RF and MARS models in the R language are now available on the Github website. We are pleased to share our data and code for the purpose of a scientific research.

**Revisions in Manuscript:**

**(1) Code availability. Line 541-543.**

The RF and MARS model for MLSCF are both available on Github (https://github.com/claus0224/MLSCF-RF-MARS), and other codes are available from the corresponding author on reasonable request.

**(2) Supplement Materials. Tables**

**Table S2. Coefficients in $V_x$ fitting of Multivariate Adaptive Regression Splines**

| Terms | Expression | Coefficients |
|---|---|---|
| 1 | Intercept | 0.532 |
| 2 | $\max(0.5-Vbg_x, 0)$ | -0.623 |
| 3 | $\max(Vbg_x-0.5, 0)$ | 0.111 |
| 4 | $\max(2.5-Vbg_y, 0)$ | -0.131 |
| 5 | $\max(Vbg_y-2.5, 0)$ | -0.010 |
| 6 | $\max(0.5-H/W, 0)$ | 2.315 |
| 7 | $\max(H/W-0.5, 0)$ | -0.259 |
| 8 | $\max(0.774-z/H, 0)$ | -0.812 |
| 9 | $\max(z/H-0.774, 0)$ | 2.774 |
| 10 | $\max(2.5-Vbg_x, 0)\times\max(0.5-H/W, 0)$ | -1.103 |
| 11 | $\max(Vbg_x-2.5, 0)\times\max(0.5-H/W, 0)$ | 0.249 |
| 12 | $\max(0.87-Vbg_x, 0)\times\max(0.774-z/H, 0)$ | 0.481 |

| 13 | $\max(Vbg_x-0.87, 0)\times\max(0.774-z/H,0)$ | -0.444 |
| 14 | $\max(2.5-Vbg_x, 0)\times\max(z/H-0.774, 0)$ | -1.151 |
| 15 | $\max(Vbg_x-2.5, 0)\times\max(z/H-0.774, 0)$ | -1.139 |
| 16 | $\max(0.5-Vbg_y, 0)\times\max(0.5-H/W, 0)$ | -3.536 |
| 17 | $\max(Vbg_y-0.5, 0)\times\max(0.5-H/W, 0)$ | 0.028 |
| 18 | $\max(0.5-H/W, 0)\times\max(0.774-z/H, 0)$ | 0.897 |
| 19 | $\max(H/W-0.5, 0)\times\max(0.774-z/H, 0)$ | 0.664 |
| 20 | $\max(Vbg_x-2.5, 0)\times\max(H_l/H_r-1.33, 0)\times\max(z/H-0.774,0)$ | -2.054 |
| 21 | $\max(Vbg_x-2.5, 0)\times\max(1.33-H_l/H_r, 0)\times\max(z/H-0.774,0)$ | 6.242 |

**Table S3. Coefficients in $V_y$ fitting of Multivariate Adaptive Regression Splines**

| Terms | Expression | Coefficients |
| --- | --- | --- |
| 1 | Intercept | 2.117 |
| 2 | $\max(2.5-Vbg_y, 0)$ | -0.812 |
| 3 | $\max(Vbg_y-2.5, 0)$ | 0.624 |
| 4 | $\max(1-H/W, 0)$ | 0.455 |
| 5 | $\max(H/W-1, 0)$ | -0.335 |
| 6 | $\max(0.75-H_l/H_r, 0)$ | -0.081 |
| 7 | $\max(H_l/H_r-0.75, 0)$ | -0.690 |
| 8 | $\max(0.079-z/H, 0)$ | -14.220 |
| 9 | $\max(z/H-0.079, 0)$ | 0.200 |
| 10 | $\max(0.5-Vbg_x, 0)\times\max(H_l/H_r-0.75, 0)$ | 0.428 |
| 11 | $\max(Vbg_x-0.5, 0)\times\max(H_l/H_r-0.75, 0)$ | -0.036 |
| 12 | $\max(2.5-Vbg_y, 0)\times\max(H/W-1, 0)$ | 0.152 |
| 13 | $\max(2.5-Vbg_y, 0)\times\max(1-H/W, 0)$ | -0.265 |
| 14 | $\max(2.5-Vbg_y, 0)\times\max(H_l/H_r-0.75, 0)$ | 0.230 |
| 15 | $\max(Vbg_y-2.5, 0)\times\max(H_l/H_r-0.75, 0)$ | 0.109 |
| 16 | $\max(2.5-Vbg_y, 0)\times\max(z/H-0.079, 0)$ | -0.090 |
| 17 | $\max(2.5-Vbg_y, 0)\times\max(0.079-z/H, 0)$ | 5.602 |
| 18 | $\max(Vbg_y-2.5, 0)\times\max(z/H-0.226, 0)$ | 0.536 |
| 19 | $\max(Vbg_y-2.5, 0)\times\max(0.226-z/H, 0)$ | -2.361 |
| 20 | $\max(1-H/W, 0)\times\max(H_l/H_r-0.75, 0)$ | 0.480 |
| 21 | $\max(H/W, 0)\times\max(H_l/H_r-0.75, 0)$ | -0.052 |

*Question 13*

Parts of the code seem to be written in Fortran, other parts in R, but it is not clear which.

If only CMAQ and WRF are written in Fortran and all other parts in R, then it is not

justified to state that a multiscale hybrid model was developed based on Fortran (and R),

because there was no development but only application of Fortran code.

Whether the model was implemented on Linux (page 5, line 96) or another platform seems irrelevant to me.

**Response:**

Thanks for your question. The MLSCF scheme is written in R language. Other parameterization schemes, including surface roughness scheme, UHI scheme, vertical mixing scheme and NOx photochemical scheme, were all written in Fortran language and then added in to the original RLINE source code. We have revised the description about the development language in the manuscript to make it clearer. And as you suggested, the statement about Linux platform was removed.

**Revisions in Manuscript:**

**(1) Materials and Methods. Line 110-112.**

Here, we established the MLSCF scheme based on R language, and modified the code of RLINE model to add other parameterization schemes with FORTRAN language. Finally, a multiscale air quality hybrid model was developed to achieve a high-resolution $NO_2$ pollution mapping in urban areas.

*Question 14*

The wind profiles predicted by the MOST scheme presented in Figure 7 look very strange. Apparently, wind speeds reduce to zero at the displacement height, but then jump back to a non-zero value below. Why is this kink in the profile at lower altitude in Figure 7c than in Figures 7a and 7b (despite the higher aspect ratio H/W in case (c) than in (a) and (b)) and why is it not present at all in Figure 7d? Why are the winds at $z/H = 1$ different between the MOST and the MLSCF schemes? Shouldn't the wind at this level be constrained by the same WRF model output?

**Response:**

Thanks for your question. The differences in kink altitude among Figure 7a-d refer to the calculation of displacement height ($d_h$). In RLINE model, the $d_h$ is calculated by multiplying surface roughness length ($z_0$) times a factor which is recommended to be set as 5. Due to the great differences in $z_0$ (highly depends on the local geometry of buildings) of each street, $d_h$ is also different. Moreover, the height of each street is also different, and

the y axis of Figure 7a-d represented z/h, so the kink altitudes ($d_h$/h) in different streets of these figures are not comparable. And the Figure 7 mainly illustrates the different predictions of wind speed between MOST and MLSCF schemes. We added the calculation method of $d_h$ in the manuscript to make the statement clearer.

The differences in winds at z/H = 1 between the MOST and the MLSCF schemes are mainly because the influence of turbulence in the street canyon on wind at the roof level is considered in the MLSCF scheme. However, in the MOST and MLSCF schemes, the wind environment higher than the roof level (z/H>1) were both from WRF model and remained the same.

**Revisions in Manuscript:**

**(1) Results. Line 366-369.**

As shown in Figure 7(a)-(d), the wind profile estimated by MOST showed a logarithmic change at the height above displacement height ($d_h$) (the $d_h$ is calculated by multiplying surface roughness length ($z_0$) times a factor which is recommended to be set as 5) with a decrease to 0 at $d_h$, and remained constant below $d_h$.

*Question 15*

Figure 8 shows differences between simulations with and without the MLSCF scheme. Why are these differences limited to very narrow lines? It is very difficult to see details in this figure. It would be useful to see a zoom into a subregion.

**Response:**

Thanks for your question. It is because MLSCF scheme only affects the concentration of grid receptors inside the street canyon. The concentrations of grid receptors outside the street canyon are not affected, so the difference shown in Figure8 is a narrow strip visually. The detailed differences in the spatial distribution of concentrations within in a street canyon has already been described in Figure 8b and d, where the SJZ street was taken as an example.

*References*

Aleksankina K, Reis S, Vieno M, Heal MR. Advanced methods for uncertainty assessment and global sensitivity analysis of an Eulerian atmospheric chemistry transport model. Atmospheric

Chemistry and Physics 2019; 19: 2881-2898.

Beddows AV, Kitwiroon N, Williams ML, Beevers SD. Emulation and Sensitivity Analysis of the Community Multiscale Air Quality Model for a UK Ozone Pollution Episode. Environmental Science & Technology 2017; 51: 6229-6236.

Benavides J, Snyder M, Guevara M, Soret A, Pérez García-Pando C, Amato F, et al. CALIOPE-Urban v1.0: coupling R-LINE with a mesoscale air quality modelling system for urban air quality forecasts over Barcelona city (Spain). Geosci. Model Dev. 2019; 12: 2811-2835.

Bondarenko M, Kerr D, Sorichetta A, Tatem A. Census/projection-disaggregated gridded population datasets for 189 countries in 2020 using Built-Settlement Growth Model (BSGM) outputs.  2020.

Breiman L. Random Forests. Machine Learning 2001; 45: 5-32.

Chang SY, Vizuete W, Valencia A, Naess B, Isakov V, Palma T, et al. A modeling framework for characterizing near-road air pollutant concentration at community scales. Science of the Total Environment 2015; 538: 905-921.

Chen G, Li S, Knibbs LD, Hamm NA, Cao W, Li T, et al. A machine learning method to estimate PM2. 5 concentrations across China with remote sensing, meteorological and land use information. Science of the Total Environment 2018; 636: 52-60.

Cimorelli AJ, Perry SG, Venkatram A, Weil JC, Paine RJ, Wilson RB, et al. AERMOD: A dispersion model for industrial source applications. Part I: General model formulation and boundary layer characterization. Journal of applied meteorology 2005; 44: 682-693.

Conibear L, Reddington CL, Silver BJ, Chen Y, Knote C, Arnold SR, et al. Statistical Emulation of Winter Ambient Fine Particulate Matter Concentrations From Emission Changes in China. GeoHealth 2021; 5: e2021GH000391.

Fang K-T, Li R, Sudjianto A. Design and modeling for computer experiments: Chapman and Hall/CRC, 2005.

Filigrana P, Milando C, Batterman S, Levy JI, Mukherjee B, Adar SD. Spatiotemporal variations in traffic activity and their influence on air pollution levels in communities near highways. Atmospheric Environment 2020; 242: 117758.

Friedman JH. Multivariate adaptive regression splines. The annals of statistics 1991; 19: 1-67.

Geng G, Meng X, He K, Liu Y. Random forest models for PM2. 5 speciation concentrations using MISR fractional AODs. Environmental Research Letters 2020; 15: 034056.

Heist D, Isakov V, Perry S, Snyder M, Venkatram A, Hood C, et al. Estimating near-road pollutant dispersion: A model inter-comparison. Transportation Research Part D: Transport and Environment 2013; 25: 93-105.

Hood, C., MacKenzie, I., Stocker, J., Johnson, K., Carruthers, D., Vieno, M., and Doherty, R.: Air quality simulations for London using a coupled regional-to-local modelling system, Atmos. Chem. Phys., 18, 11221-11245, 10.5194/acp-18-11221-2018, 2018.

Hu X, Belle JH, Meng X, Wildani A, Waller LA, Strickland MJ, et al. Estimating PM2. 5 concentrations in the conterminous United States using the random forest approach. Environmental science & technology 2017; 51: 6936-6944.

Hurley P. The air pollution model (TAPM) version 3. Part 1. Technical description. Aspendale, Vic. CSIRO Atmospheric Research, 2005.

Kamińska JA. A random forest partition model for predicting NO2 concentrations from traffic flow and meteorological conditions. Science of the Total Environment 2019; 651: 475-483.

Keller CA, Evans MJ. Application of random forest regression to the calculation of gas-phase chemistry within the GEOS-Chem chemistry model v10. Geoscientific Model Development 2019; 12: 1209-1225.

Kim Y, Wu Y, Seigneur C, Roustan Y. Multi-scale modeling of urban air pollution: development and application of a Street-in-Grid model (v1. 0) by coupling MUNICH (v1. 0) and Polair3D (v1. 8.1). Geoscientific Model Development 2018; 11: 611-629.

Kühnlein M, Appelhans T, Thies B, Nauss T. Improving the accuracy of rainfall rates from optical satellite sensors with machine learning — A random forests-based approach applied to MSG SEVIRI. Remote Sensing of Environment 2014; 141: 129-143.

Reichstein M, Camps-Valls G, Stevens B, Jung M, Denzler J, Carvalhais N, et al. Deep learning and process understanding for data-driven Earth system science. Nature 2019; 566: 195-204.

Sarrat C, Lemonsu A, Masson V, Guédalia D. Impact of urban heat island on regional atmospheric pollution. Atmospheric environment 2006; 40: 1743-1758.

Snyder MG, Venkatram A, Heist DK, Perry SG, Petersen WB, Isakov V. RLINE: A line source dispersion model for near-surface releases. Atmospheric Environment 2013; 77: 748-756.

Soulhac L, Nguyen C, Volta P, Salizzoni P. The model SIRANE for atmospheric urban pollutant dispersion. PART III: Validation against NO 2 yearly concentration measurements in a large urban agglomeration. Atmospheric Environment 2017; 167.

Valencia A, Venkatram A, Heist D, Carruthers D, Arunachalam S. Development and evaluation of the R-LINE model algorithms to account for chemical transformation in the near-road environment. Transportation Research Part D: Transport and Environment 2018; 59: 464-477.

Wang K, Jiang S, Wang J, Zhou C, Wang X, Lee X. Comparing the diurnal and seasonal variabilities of atmospheric and surface urban heat islands based on the Beijing urban meteorological network. Journal of Geophysical Research: Atmospheres 2017; 122: 2131-2154.

Zhai X, Russell AG, Sampath P, Mulholland JA, Kim B-U, Kim Y, et al. Calibrating R-LINE model results with observational data to develop annual mobile source air pollutant fields at fine spatial resolution: Application in Atlanta. Atmospheric Environment 2016; 147: 446-457.

Zhang X, Just AC, Hsu H-HL, Kloog I, Woody M, Mi Z, et al. A hybrid approach to predict daily NO2 concentrations at city block scale. Science of The Total Environment 2021; 761: 143279.

**Response to Reviewers #2's Comments**

*Summary*

The regional to urban coupling allows the consideration of regional weather effects in local models and plays an important role in the improved prediction of local air pollution. The development of such multiscale modelling framework is interesting. This paper developed a hybrid CMAQ-RLINE_URBAN, which coupled the regional CMAQ Chemical Transport Model, RLINE local dispersion model and urban thermodynamic scheme. Intensive CFD street canyon simulations have been conducted for the application of Machine Learning. The hybrid model has been applied to one month simulation in Summer for Beijing as a case study. It performed better than the regional CMAQ model in predicting NO2 concentrations for roadside sites. However, there are still a number of major comments to be addressed.

**Response:**

Thank you very much to give us constructive comments. Upon learning through them, we greatly improved our manuscript. We believe all the concerns you mentioned at this time were addressed in this revision.

*Major comments*
*Question 1*

Literature review: Lack of discussion about the computational fluid dynamic (CFD). CFD can be classified into two categories: Reynolds-averaged Navier-Stokes (RANS) and Large-Eddy Simulation (LES), based on turbulence closure schemes (e.g. https://doi.org/10.1016/j.envpol.2016.04.052). Discussion about the comparison between RANS and LES is needed, and to justify the use of RANS in the present study (e.g. computationally faster than LES, but only resolve the mean time-averaged properties).

**Response:**

Thanks for your advice. Considering the topic of this paper is "coupled model", the description of CFD method and its turbulence closure schemes was not included in the Introduction. As you mentioned, we agreed that it was necessary to discussion about the comparison between RANS and LES. We have added the review of turbulence closure schemes in the CFD method section (Section 2.2.2). Furthermore, we clarified the reason for the use of the RANS, as follows:

a. A total of 1600 simulations are implemented in this study. Due to such a huge computation cost, we must choose the RANS which is much faster than the LES.

b. The geometry of street canyons in our modelling is uncomplicated, so the RANS can meet our experimental accuracy requirements.

**Revisions in Manuscript:**

**(1) Materials and Methods. Line 203-211.**

The turbulence closure schemes for CFD include the Reynolds-Averaged Navier-Stokes (RANS) and the Large-Eddy Simulation (LES), and the choice of them depends on the computational cost, the accuracy required and the purpose of application. The RANS resolves the mean time-averaged properties with all the turbulence motions to be modelled, while LES adopts a spatial filtering operation and consequently resolves large-scale eddies directly and parameterizes small-scale eddies (Zhong et al., 2016). Compared with the LES, the RANS is more easily established and computationally faster (Xie and Castro, 2006). However, the LES can provide a better prediction of air flow than that from the RANS when handling complex geometries (Dejoan et al., 2010; Santiago et al., 2010). In this study, considering the huge computational burden of a large number of simulations and the relatively simple geometry of street canyons in our modelling, the RANS was selected to characterize the air flow.

*Question 2*

Literature review: there is not enough information about local RLINE model, and also the coupled CMAQ-RLINE model. Then to justify the need of the further development of CMAQ-RLINE_URBAN in the present study. Also, the literature about the current status of regional-to-urban coupling is missing (e.g. https://doi.org/10.5194/acp-18-11221-2018, 2018).

**Response:**

Thanks for your advice. We apologized that the detailed introduction of RLINE was missing in our original manuscript. In general, the RLINE is a Gaussian dispersion model specially for the line source simulation. In this revision, we have added the description of RLINE model including its mechanism and application in **Method Section 2.1.**

Although RLINE has been successfully used in many studies to evaluate the impacts from traffic emissions on air quality, there are still large uncertainties in predictions from induced by the provided meteorological conditions and background concentrations, especially the application in urban areas. It is similar with other Gaussian dispersion models, where the natural logarithm function is still used to characterize the vertical profile of wind speed in both the inertial and rough sublayers, neglecting the influence of urban complex underlying surface compositions on the wind field. Thus, it is essential to develop a coupled model, such as CMAQ-RLINE_URBAN in our study. The detailed discussions about this weakness have been already presented in the **Introduction** section in the original manuscript.

In this revision, as you suggested, we have also added a review on about the current status of regional-to-urban coupling in the **Introduction** section to further describe the innovation of our model.

**Revisions in Manuscript:**

**(1) Introduction. Line 84-96.**

Considering the respective strengths and limitations of regional models and local models, several studies have been carried out on coupling of air quality models applicable to different scales (Ketzel et al., 2012; Stocker et al., 2012;   Lefebvre et al., 2013; Jensen et al., 2017; Kim et al., 2018; Mallet et al., 2018; Hood et al., 2018; Benavides et al., 2019; Kamińska, 2019; Mu et al., 2022). Although these models performed accurately in near-road simulation, the influence of street canyons is still hard to be considered. In some hybrid models (Stocker et al., 2012; Jensen et al., 2017; Mallet et al., 2018), OSPM was still applied to calculate concentration levels within the street, where the application of logarithmic wind profile probably overestimated the bottom wind speed in a deep street canyon as abovementioned. Other models simply assumed that in street canyons, wind direction followed the street direction, and wind speed was uniform, which was not sufficient to resolve the concentration gradient within street canyons ( Kim et al., 2018; Benavides et al., 2019). Berchet et al. (2017) proposed a cost-effective method for simulating city-scale pollution taking advantage of high-resolution accurate CFD, while the primary $NO_x$ was predicted due to the lack of a chemical module. Therefore, it is

essential to build an integrated model to predict long-term and near-road air pollution suitable for the urban complex underlying surface environment.

**(2) Materials and Methods. Line 119-128.**

RLINE is a Gaussian line source dispersion model developed by Snyder et al. (2013) to predict pollutant concentrations in near-road environments. In the RLINE model, the mobile source is considered as a finite line source, from which the concentration is found by approximating the line as a series of point sources and integrating the contributions of point sources using an efficient numerical integration scheme. The number of points needed for convergence to the proper solution is a function of distance from the source line to the receptor, and each point source is simulated using a Gaussian plume formulation. The RLINE model performs generally comparable results when evaluated with other line source models for on-road traffic emissions dispersion (Snyder et al., 2013; Heist et al., 2013; Chang et al., 2015), and has been successfully used in many studies to evaluate the impacts from traffic emissions on air quality (Zhai et al., 2016; Valencia et al., 2018; Benavides et al., 2019; Filigrana et al., 2020; Zhang et al., 2021a).

*Question 3*

Resolution for the hybrid model CMAQ-RLINE_URBAN. The resolution of 50 m x 50 m is still coarse to resolve the street scale dispersion of road sources. Could such resolution be flexible (i.e. further to higher resolutions) in the hybrid model? The justification of the use of 50 m x 50 m in the present study is needed

**Response:**

Thanks for your question. Actually, the grid resolution in our hybrid model is flexible. However, the grid resolution over the whole urban area is limited to 50 m×50 m due to the long computing time for such a large domain at present. Since the addition developed parameterization schemes were applied in the hybrid model, especially for the MLSCF, the meteorological field of each street needs to be calculated separately, leading to the large computational burden. However, when we focus on the distribution of concentration gradient near the street, the resolution of grid receptors will be improved to several meters level. For example, in Figure 8b and d, the grid resolution near SJZ street was improved

to be 2 m. In the future, we will make efforts to develop a parallel computing method with multi cores to reduce the computing time, in order to improve the grid resolution of a relatively large-scale simulation. We have modified the writing and added this discussion in the **Conclusions** section.

**Revisions in Manuscript:**

**(1) Materials and Methods. Line 116-117.**

In our model, a $NO_2$ pollution map with a high temporal (1 h) and spatial resolution (50 m×50 m) can finally be obtained.

**(2) Conclusion and Discussions. Line 519-522.**

At present, considering the running cost, the grid resolution of area in Beijing 5th ring road and its surroundings can reach 50 m×50 m. We will make efforts to develop a parallel computing method to reduce the computing time, in order to improve the grid resolution of a relatively large-scale simulation.

*Question 4*

Machine learning is for air flow (wind speed) only. How is it linked to the pollutant dispersion? Would it be better than a traditional street canyon model (e.g. https://doi.org/10.1080/10962247.2020.1803158)?

**Response:**

Thanks for questions. In this study, the MLSCF is developed to estimate the wind velocity and direction at different height in the street canyon. In other words, outputs from the MLSCF model are wind vectors. Therefore, the impact of turbulence induced by street canyon effect on wind environment is considered. When the receptor is located within a street canyon, the wind field within the UCL was simulated by MLSCF scheme and used to calculate the pollutant dispersion, which has been already shown in Figure 1 and discussed in section 2.1. As shown in the Figure 8, the impacts of the MLSCF scheme on simulated $NO_2$ concentration were identified by the differences between modeling scenarios with and without MLSCF. For example, in the SZJ standard canyon, the application of MLSCF led to the wind direction at the bottom in street canyon opposite to that at the roof, increasing the upwind concentrations (Figure 8b and Section 3.2 in the

manuscript).

Compared with the traditional street canyon model, such as the OSPM, the MLSCF in our model considered the influence of geometry of buildings on the air flow with high aspect ratio up to 2 (see Table 1), and the wind profile is affected by the geometry of buildings instead of a logarithmic wind profile. However, when compared with research of a "box model" simulation you mentioned (Hood et al., 2021), it is hard to compare which one is better due to different computation framework and mode. However, we are committed to comparing our model with other researches through real cases in the future.

*Question 5*

It is not clear how NOx photochemical scheme works? Does it explicitly resolve the simple NOx-O3 cycle? If VOCs chemistry is further considered, then it would likely make a substantial difference in predicting NO2 concentration (e.g. https://doi.org/10.1016/j.envpol.2017.01.076). It is suggested to add some discussion on this aspect.

**Response:**

Thank you for questions. We apologized those unclear descriptions on the NOx photochemical reactions in the original manuscript. In general, we used the two-reaction method applied in other studies, such as the SIRANE model (Soulhac et al., 2017). The $NO_x$ photochemical scheme includes two main chemical reactions, namely the photolysis of $NO_2$ and the oxidation of NO as follows:

$$\begin{cases} NO_2 + hv \rightarrow NO + O_3 \\ \quad\ NO + O_3 \rightarrow NO_2 \end{cases}$$

During simulation, the NOx ($NO+NO_2$) emitted from vehicles is first regarded as an inert gas and only the primary concentration after diffusion is simulated. Then, assuming a photo-stationary equilibrium condition, the concentrations of NO, $NO_2$ and $O_3$ are calculated.

We have added a brief introduction of the "two-reaction scheme" in **Materials and Methods** and details in **Supplement Materials (Section S3. NOx photochemical parameter scheme)**.

For the question about VOCs chemistry, Kim has already compared a simple mechanism

only involving NOx and $O_3$ (Leighton mechanism), with the CB05 gas phase chemical mechanism including VOCs chemistry by incorporated them into SinG model to estimate roadside $NO_2$ concentration respectively, and found a very similar predictions (Kim et al., 2018b). Therefore, the Leighton mechanism was selected in an operational version of SinG due to the halved computational time. However, Zhong et al. found the $NO_2$ and $O_x$ inside the canyon was enhanced by 30–40% via OH/HO2 chemistry in the canyon (Zhong et al., 2017). The difference of the influence of VOCs chemistry on concentrations in these studies mainly due to the differences in local meteorological conditions, emissions and other factors (e.g. geometry of canyons). Considering that the input emission data of two-reaction scheme are more accessible and the computational cost is lower compared with those in the $O_3$-NOx-VOC chemistry, we chose two-reaction scheme in this study. We will make effort on investigate the influence of chemistry scheme on simulation in the future, and this is added in the **Conclusions** now.

**Revisions in Manuscript:**

**(1) Materials and Methods. Line 167-170.**

In this study, a simplified two-reaction scheme, including the photolysis of $NO_2$ and the oxidation of NO, was incorporated into the model to characterize the photochemical process of $NO_x$ (details in the Supplement Section S3), which has been successfully applied in the SIRANE dispersion model (Soulhac et al., 2017).

**(2) Conclusion and Discussions. Line 522-527.**

In our study, a simplified two-reaction scheme was incorporated into the model to characterize the photochemical process of $NO_x$, since it performed similar predictions and less computational time compared with those of the complicated CB05 gas phase chemical mechanism (Kim et al., 2018). However, another study pointed that the impact of nonlinear $O_3$-$NO_x$-VOC chemistry on $NO_2$ concentrations in the deep canyon was nonnegligible (Zhong et al., 2017). The influence of different chemistry schemes on near-road simulation will be investigated in the future.

**(3) Supplement Materials Section S3. NOx photochemical parameter scheme.**

The $NO_x$ photochemical parameter scheme applied in this study includes two reactions:

$$\begin{cases} NO_2 + hv \rightarrow NO + O_3 \\ NO + O_3 \rightarrow NO_2 \end{cases}$$

Kim et al. compared two-reaction scheme with CB05 gas phase chemical mechanism by incorporated them into SinG model to estimate roadside $NO_2$ concentration, and found a similar results, while the computing time cost of two-reaction scheme was significantly less than that of the CB05 mechanism (Kim et al., 2018). Therefore, the simplified two-reaction scheme was incorporated into the model in this study to characterize the $NO_x$ photochemical process. During simulation, the NOx (NO+$NO_2$) emitted from vehicles is first regarded as an inert gas and only the primary concentration after diffusion is simulated. Then, assuming a photo-stationary equilibrium condition, the concentrations of NO, $NO_2$ and $O_3$ are calculated using the two-reaction scheme, as follows:

$$\begin{cases} [NO_2] = (b - \sqrt{b^2 - 4c})/2 \\ [NO] = [NO]_b + [NO_2]_b + [NO_x]_d - [NO_2] \\ [O_3] = [O_3]_b + [NO_2]_b + \zeta[NO_x]_d - [NO_2] \\ b = k1/k2 + [O_3]_b + [NO]_b + 2[NO_2]_b + (1 + \zeta)[NO_x]_d \\ c = ([O_3]_b + [NO_2]_b + \zeta[NO_x]_d)([NO]_b + [NO_2]_b + [NO_x]_d) \end{cases}$$

where, $[NO_x]_d$ is the primary concentration of $NO_x$ directly simulated by RLINE model when taken as an inert gas. $[NO]_b$, $[NO_2]$, and $[O_3]_b$ are the background concentrations of NO, $NO_2$ and $O_3$ from non-vehicle sources, respectively, which are provided by CMAQ-ISAM model. The unit of concentrations in these formulas is mol/m$^3$. $\zeta$ is the ratio of $NO_2$ to $NO_x$ in vehicle emissions, with a value of 0.2 (Benavides et al., 2019; Valencia et al., 2018). The reaction rates of the photolysis of $NO_2$ and the oxidation of NO were set to be $k1$ and $k2$ respectively, and calculated as follows (Hurley, 2005):

$$\begin{cases} k1 = 10^{-4} \times \delta \times TSR \\ k2 = 9.24 \times 10^5 \times \exp(-1450/T)/T \end{cases}$$

$$\delta = \begin{cases} 4.23 + 1.09/\cos Z, & 0 \leq Z \leq 47 \\ 5.82, & 47 < Z \leq 64 \\ -0.997 + 12(1 - \cos Z), & 64 < Z \leq 90 \end{cases}$$

where, all parameters were from the WRF model. TSR is the total solar radiation (W/m$^2$).

$Z$ is the solar zenith angle (°). $T$ is the ambient temperature (K).

***Minor comments***

***Question 6***

Line 39: Which pollutant does these measures of industrial and domestic sources aim to tackle?

Is it for PM2.5, rather than NO2?

**Response:**

Thanks for your reminding. This research (Zhang et al., 2019) is aim to track $PM_{2.5}$. However, the main air pollution control measures on industrial, domestic and mobile sources mentioned in this study will also relieve the $NO_2$ pollution, such as the strengthen industrial emissions standards, upgrades and phase out on industrial capacities. We have revised the statement now for better understanding.

**Revisions in Manuscript:**

**(1) Introduction. Line 38-40.**

The improvement of $PM_{2.5}$ in China was mainly due to the emission reduction and control measures of industrial and domestic sources (Zhang et al., 2019), which also relieved the $NO_2$ pollution, but the reduction potential of these sources has been gradually declining.

*Question 7*

Lines 42-43: The poor dispersion caused by buildings along the street would also play a key role in it. High pollutant concentrations in street canyon environment are caused by combined effects of poor dispersion, increased traffic emissions and chemistry processes.

**Response:**

Thanks very much for your advice. We agreed that this statement should be more precise. We have revised the original description.

**Revisions in Manuscript:**

**(1) Introduction. Line 41-43.**

Due to the low release height of vehicle emissions, combined with the negative dispersion condition caused by nearby buildings, air pollutants will be significantly accumulated near the street.

*Question 8*

Line 58: "has" should be "have".

**Response:**

Thanks for your comments. We have revised it now.

**Revisions in Manuscript:**

**(1) Introduction. Line 55-59.**

Regional-scaled air quality models, represented by Chemical Transport Models (CTMs) including Community Multi-scale Air Quality (CMAQ) model (Byun and Schere, 2006), Comprehensive Air quality Model with extensions (CAMx), and Weather Research and Forecasting/Chemistry model (WRF-Chem) (Grell et al., 2005), have been used extensively in assessment on the impacts of vehicle emissions on the regional atmospheric environment.

*Question 9*

Line 96: "Based on FORTRAN and R languages", it is not clear. Which part is based on R? Is it for a post-processing tool?

**Response:**

Thanks for your reminding. The MLSCF scheme (including RF and MARS) was established based on R language, and we further modified the code of RLINE model to add other parameterization schemes with FORTRAN language. We added more description to make it clearer in the manuscript.

**Revisions in Manuscript:**

**(1) Materials and Methods. Line 110-111.**

Here, we established the MLSCF scheme based on R language, and modified the code of RLINE model to add other parameterization schemes with FORTRAN language.

*References*

Benavides J, Snyder M, Guevara M, Soret A, Pérez García-Pando C, Amato F, et al. CALIOPE-Urban v1.0: coupling R-LINE with a mesoscale air quality modelling system for urban air quality forecasts over Barcelona city (Spain). Geosci. Model Dev. 2019; 12: 2811-2835.

Berchet A, Zink K, Muller C, Oettl D, Brunner J, Emmenegger L, et al. A cost-effective method for simulating city-wide air flow and pollutant dispersion at building resolving scale. Atmospheric Environment 2017; 158: 181-196.

Byun D, Schere KL. Review of the Governing Equations, Computational Algorithms, and Other Components of the Models-3 Community Multiscale Air Quality (CMAQ) Modeling System. Applied Mechanics Reviews 2006; 59: 51-77.

Chang SY, Vizuete W, Valencia A, Naess B, Isakov V, Palma T, et al. A modeling framework for characterizing near-road air pollutant concentration at community scales. Science of the Total Environment 2015; 538: 905-921.

Dejoan A, Santiago J, Martilli A, Martin F, Pinelli A. Comparison between large-eddy simulation and Reynolds-averaged Navier–Stokes computations for the MUST field experiment. Part II: effects of incident wind angle deviation on the mean flow and plume dispersion. Boundary-layer meteorology 2010; 135: 133-150.

Filigrana P, Milando C, Batterman S, Levy JI, Mukherjee B, Adar SD. Spatiotemporal variations in traffic activity and their influence on air pollution levels in communities near highways. Atmospheric Environment 2020; 242: 117758.

Grell GA, Peckham SE, Schmitz R, McKeen SA, Frost G, Skamarock WC, et al. Fully coupled "online" chemistry within the WRF model. Atmospheric Environment 2005; 39: 6957-6975.

Heist D, Isakov V, Perry S, Snyder M, Venkatram A, Hood C, et al. Estimating near-road pollutant dispersion: A model inter-comparison. Transportation Research Part D: Transport and Environment 2013; 25: 93-105.

Hood C, MacKenzie I, Stocker J, Johnson K, Carruthers D, Vieno M, et al. Air quality simulations for London using a coupled regional-to-local modelling system. Atmos. Chem. Phys. 2018; 18: 11221-11245.

Hood C, Stocker J, Seaton M, Johnson K, O'Neill J, Thorne L, et al. Comprehensive evaluation of an advanced street canyon air pollution model. Journal of the Air & Waste Management Association 2021; 71: 247-267.

Hurley P. The air pollution model (TAPM) version 3. Part 1. Technical description. Aspendale, Vic. CSIRO Atmospheric Research, 2005.

Jensen SS, Ketzel M, Becker T, Christensen J, Brandt J, Plejdrup M, et al. High resolution multi-scale air quality modelling for all streets in Denmark. Transportation Research Part D: Transport and Environment 2017; 52: 322-339.

Kamińska JA. A random forest partition model for predicting $NO_2$ concentrations from traffic flow and meteorological conditions. Science of the Total Environment 2019; 651: 475-483.

Ketzel M, Jensen S, Brandt J, Ellermann T, Berkowicz R, Hertel O. Evaluation of the street pollution model OSPM for measurement at 12 street stations using using newly developed and freely available evaluation tool. J. Civil. Environ. Eng. 2012.

Kim Y, Wu Y, Seigneur C, Roustan Y. Multi-scale modeling of urban air pollution: development and application of a Street-in-Grid model (v1.0) by coupling MUNICH (v1.0) and Polair3D (v1.8.1). Geosci. Model Dev. 2018; 11: 611-629.

Lefebvre W, Van Poppel M, Maiheu B, Janssen S, Dons E. Evaluation of the RIO-IFDM-street canyon model chain. Atmospheric Environment 2013; 77: 325-337.

Mallet V, Tilloy A, Poulet D, Girard S, Brocheton F. Meta-modeling of ADMS-Urban by dimension reduction and emulation. Atmospheric Environment 2018; 184: 37-46.

Mu Q, Denby BR, Wærsted EG, Fagerli H. Downscaling of air pollutants in Europe using uEMEP_v6. Geosci. Model Dev. 2022; 15: 449-465.

Santiago J, Dejoan A, Martilli A, Martin F, Pinelli A. Comparison between large-eddy simulation and Reynolds-averaged Navier–Stokes computations for the MUST field experiment. Part I: study of the flow for an incident wind directed perpendicularly to the front array of containers. Boundary-Layer Meteorology 2010; 135: 109-132.

Snyder MG, Venkatram A, Heist DK, Perry SG, Petersen WB, Isakov V. RLINE: A line source dispersion model for near-surface releases. Atmospheric Environment 2013; 77: 748-756.

Soulhac L, Nguyen C, Volta P, Salizzoni P. The model SIRANE for atmospheric urban pollutant

dispersion. PART III: Validation against NO 2 yearly concentration measurements in a large urban agglomeration. Atmospheric Environment 2017; 167.

Stocker J, Hood C, Carruthers D, McHugh C. ADMS-Urban: developments in modelling dispersion from the city scale to the local scale. International Journal of Environment and Pollution 2012; 50: 308-316.

Valencia A, Venkatram A, Heist D, Carruthers D, Arunachalam S. Development and evaluation of the R-LINE model algorithms to account for chemical transformation in the near-road environment. Transportation Research Part D: Transport and Environment 2018; 59: 464-477.

Xie Z, Castro IP. LES and RANS for turbulent flow over arrays of wall-mounted obstacles. Flow, Turbulence and Combustion 2006; 76: 291-312.

Zhai X, Russell AG, Sampath P, Mulholland JA, Kim B-U, Kim Y, et al. Calibrating R-LINE model results with observational data to develop annual mobile source air pollutant fields at fine spatial resolution: Application in Atlanta. Atmospheric Environment 2016; 147: 446-457.

Zhang Q, Zheng Y, Tong D, Shao M, Wang S, Zhang Y, et al. Drivers of improved $PM_{2.5}$ air quality in China from 2013 to 2017. Proceedings of the National Academy of Sciences 2019; 116: 24463-24469.

Zhang X, Just AC, Hsu H-HL, Kloog I, Woody M, Mi Z, et al. A hybrid approach to predict daily NO2 concentrations at city block scale. Science of The Total Environment 2021; 761: 143279.

Zhong J, Cai X-M, Bloss WJ. Coupling dynamics and chemistry in the air pollution modelling of street canyons: A review. Environmental Pollution 2016; 214: 690-704.

Zhong J, Cai X-M, Bloss WJ. Large eddy simulation of reactive pollutants in a deep urban street canyon: Coupling dynamics with O3-NOx-VOC chemistry. Environmental Pollution 2017; 224: 171-184.

**Response to Reviewers #3's Comments**

*Summary*

In this work, a hybrid model has been developed and evaluated to analyse the effects of vehicle emissions on urban roadside concentrations of NO2 in Beijing. The article is well written and raises an important topic, the link between the simulations done using regional chemistry transport models and the simulations at the urban level done using gaussian/dispersion models. However, there are a few points that should be clarified in order to make clearer the evaluation of the model and the scenarios tested.

**Response:**

Thank you very much for spending time to give us so many constructive comments. Upon learning through them, we improved our manuscript. We try our best to address all the concerns in this revision.

*Major comments*
*Question 1*

The introduction clearly shows the differences between chemistry transport models and dispersion/gaussian models highlighting the difficulties of the former in predicting the roadside concentrations. However, there isn't a clear link between regional models and urban models. Few works have been published and few models have been already developed to couple regional and urban models and these should be mentioned in the introduction.

**Response:**

Thanks for your advice. In recent years, considering the respective strengths and limitations of chemistry transport models and dispersion/gaussian models, several studies have been carried out on coupling of air quality models applicable to different scales. And we apologized that the introduction of coupled model was missing in our original manuscript. Now we have added a review about the current status of coupled model in the Introduction to further describe the innovation of our model.

**Revisions in Manuscript:**

**(1) Introduction. Line 84-96.**

Considering the respective strengths and limitations of regional models and local models,

several studies have been carried out on coupling of air quality models applicable to different scales (Ketzel et al., 2012; Stocker et al., 2012; Lefebvre et al., 2013; Jensen et al., 2017; Kim et al., 2018; Mallet et al., 2018; Hood et al., 2018; Benavides et al., 2019; Kamińska, 2019; Mu et al., 2022). Although these models performed accurately in near-road simulation, the influence of street canyons is still hard to be considered. In some hybrid models (Stocker et al., 2012; Jensen et al., 2017; Mallet et al., 2018), OSPM was still applied to calculate concentration levels within the street, where the application of logarithmic wind profile probably overestimated the bottom wind speed in a deep street canyon as abovementioned. Other models simply assumed that in street canyons, wind direction followed the street direction, and wind speed was uniform, which was not sufficient to resolve the concentration gradient within street canyons ( Kim et al., 2018; Benavides et al., 2019). Berchet et al. (2017) proposed a cost-effective method for simulating city-scale pollution taking advantage of high-resolution accurate CFD, while the primary NOx was predicted due to the lack of a chemical module. Therefore, it is essential to build an integrated model to predict long-term and near-road air pollution suitable for the urban complex underlying surface environment.

*Question 2*

The methodology highlights only part of the process defined in Figure 1. The authors focus their discussion on the urban model but WRF and CMAQ configuration and outputs should also be mentioned and discussed.

**Response:**

Thanks for your advice. The configuration of WRF and CMAQ model was introduced in detail in our previous study (Lv et al., 2020), where each input data and parameterization schemes are discussed. This information has been already provided in the Method section. The prediction of CMAQ model has also already shown and discussed in Figure 9-11 and Table 2, including the outputs of CMAQ, the comparison of different models and the validation of CMAQ. In this revision, we added the validation of predictions from WRF model compared with observations in **Table S6 in in Supplement Materials.**

**Revisions in Manuscript:**

**(1) Supplement Materials. Table S6.**

**Table S6. The performance of WRF model compared with observations.**

| Variables | Sample size | Observed Average | Simulated Average | MB | NMB | RMSE | *R* |
|---|---|---|---|---|---|---|---|
| WS10 (m/s) | 732 | 2.5 | 3.7 | 1.2 | 46 | 1.9 | 0.6 |
| WD10 (°) | 456 | 190.4 | 169.0 | -8.0 | -4 | 49.5 | 0.4 |
| T2 (℃) | 742 | 25.8 | 29.0 | 3.2 | 12 | 3.5 | 0.9 |
| RH (%) | 741 | 64.3 | 50.4 | -13.9 | -22 | 17.4 | 0.9 |

*WS10: wind speed at the height of 10 m; WD10: wind direction at the height of 10 m; T2: Temperature at the height of 2 m; RH: Relative humidity; MB: Mean bias; RSME: Root mean squared error; NMB: Normalized mean bias; R: correlation coefficient.

*Question 3*

The simulations are run for a period of high photochemical activity. This is surely dependent on weather conditions that are completely absent from the article?

**Response:**

Thanks for your advice. We apologized that the description about weather conditions in simulation period is missing. In this study, we choose summer (August 1st to 31th in 2019) as simulation period since the strong photochemical reactions, induced by the high temperature (the average of daily high temperatures higher than 30 ℃) and strong solar radiation conditions (sunlight hours longer than 13 hours). Now we revised this statement to make it clearer.

**Revisions in Manuscript:**

**(1) Materials and Methods. Line 300-302.**

The near-ground $NO_2$ concentrations were simulated from August 1st to 31th in 2019 when the average of daily high temperatures was higher than 30 ℃ and sunlight duration was longer than 13 hours, leading to strong photochemical reactions.

*Question 4*

The NOx-O3 system include also VOCs. There is no mention of this in the methodology or in the results. Are VOCs included in the simulations? It would be good to add the chemical mechanism somewhere in the supplementary material?

**Response:**

Thanks for your advice. We apologized those unclear descriptions on the NOx photochemical reactions in the original manuscript. In general, the VOCs is not included in our study, where a simple mechanism only involving NOx and $O_3$ was used. We used the two-reaction method applied in other studies, such as the SIRANE model (Soulhac et al., 2017). The $NO_x$ photochemical scheme includes two main chemical reactions, namely the photolysis of $NO_2$ and the oxidation of NO as follows:

$$\begin{cases} NO_2 + hv \rightarrow NO + O_3 \\ \quad NO + O_3 \rightarrow NO_2 \end{cases}$$

During simulation, the NOx (NO+$NO_2$) emitted from vehicles is first regarded as an inert gas and only the primary concentration after diffusion is simulated. Then, assuming a photo-stationary equilibrium condition, the concentrations of NO, $NO_2$ and $O_3$ are calculated.

Following your suggestions, we have added a brief introduction of the "two-reaction scheme" in **Materials and Methods** and details in **Supplement Materials (Section S3. NOx photochemical parameter scheme)**.

For comments about VOCs chemistry, Kim has already compared a simple mechanism only involving NOx and $O_3$ (Leighton mechanism), with the CB05 gas phase chemical mechanism including VOCs chemistry by incorporated them into SinG model to estimate roadside $NO_2$ concentration respectively, and found a very similar predictions (Kim et al., 2018). Therefore, the Leighton mechanism was selected in an operational version of SinG due to the halved computational time. However, Zhong et al. found the $NO_2$ and $O_x$ inside the canyon was enhanced by 30–40% via OH/HO2 chemistry in the canyon (Zhong et al., 2017). The difference of the influence of VOCs chemistry on concentrations in these studies mainly due to the differences in local meteorological conditions, emissions and other factors (e.g. geometry of canyons). Considering that the input emission data of two-reaction scheme are more accessible and the computational cost is lower compared with those in the $O_3$-NOx-VOC chemistry, we chose two-reaction scheme in this study. We will make effort on investigate the influence of chemistry scheme on simulation in the future, and this is added in the **Conclusions** now.

**Revisions in Manuscript:**

**(1) Materials and Methods. Line 167-170.**

In this study, a simplified two-reaction scheme, including the photolysis of $NO_2$ and the oxidation of NO, was incorporated into the model to characterize the photochemical process of $NO_x$ (details in the Supplement Section S3), which has been successfully applied in the SIRANE dispersion model (Soulhac et al., 2017).

**(2) Conclusion and Discussions. Line 522-527.**

In our study, a simplified two-reaction scheme was incorporated into the model to characterize the photochemical process of $NO_x$, since it performed similar predictions and less computational time compared with those of the complicated CB05 gas phase chemical mechanism (Kim et al., 2018). However, another study pointed that the impact of nonlinear $O_3$-$NO_x$-VOC chemistry on $NO_2$ concentrations in the deep canyon was nonnegligible (Zhong et al., 2017). The influence of different chemistry schemes on near-road simulation will be investigated in the future.

**(3) Supplement Materials Section S3. NOx photochemical parameter scheme.**

The $NO_x$ photochemical parameter scheme applied in this study includes two reactions:

$$\begin{cases} NO_2 + hv \rightarrow NO + O_3 \\ \quad NO + O_3 \rightarrow NO_2 \end{cases}$$

Kim et al. compared two-reaction scheme with CB05 gas phase chemical mechanism by incorporated them into SinG model to estimate roadside $NO_2$ concentration, and found a similar results, while the computing time cost of two-reaction scheme was significantly less than that of the CB05 mechanism (Kim et al., 2018). Therefore, the simplified two-reaction scheme was incorporated into the model in this study to characterize the $NO_x$ photochemical process. During simulation, the NOx ($NO+NO_2$) emitted from vehicles is first regarded as an inert gas and only the primary concentration after diffusion is simulated. Then, assuming a photo-stationary equilibrium condition, the concentrations of NO, $NO_2$ and $O_3$ are calculated using the two-reaction scheme, as follows:

$$\begin{cases} [NO_2] = (b - \sqrt{b^2 - 4c})/2 \\ [NO] = [NO]_b + [NO_2]_b + [NO_x]_d - [NO_2] \\ [O_3] = [O_3]_b + [NO_2]_b + \zeta[NO_x]_d - [NO_2] \\ b = k1/k2 + [O_3]_b + [NO]_b + 2[NO_2]_b + (1 + \zeta)[NO_x]_d \\ c = ([O_3]_b + [NO_2]_b + \zeta[NO_x]_d)([NO]_b + [NO_2]_b + [NO_x]_d) \end{cases}$$

where, $[NO_x]_d$ is the primary concentration of $NO_x$ directly simulated by RLINE model when taken as an inert gas. $[NO]_b$, $[NO_2]$, and $[O_3]_b$ are the background concentrations of NO, $NO_2$ and $O_3$ from non-vehicle sources, respectively, which are provided by CMAQ-ISAM model. The unit of concentrations in these formulas is mol/m$^3$. $\zeta$ is the ratio of $NO_2$ to $NO_x$ in vehicle emissions, with a value of 0.2 (Benavides et al., 2019; Valencia et al., 2018). The reaction rates of the photolysis of $NO_2$ and the oxidation of NO were set to be *k1* and *k2* respectively, and calculated as follows (Hurley, 2005):

$$\begin{cases} k1 = 10^{-4} \times \delta \times \text{TSR} \\ k2 = 9.24 \times 10^5 \times \exp(-1450/T) /T \end{cases}$$

$$\delta = \begin{cases} 4.23 + 1.09/\cos Z, & 0 \leq Z \leq 47 \\ 5.82, & 47 < Z \leq 64 \\ -0.997 + 12(1 - \cos Z), & 64 < Z \leq 90 \end{cases}$$

where, all parameters were from the WRF model. TSR is the total solar radiation (W/m$^2$). $Z$ is the solar zenith angle (°). $T$ is the ambient temperature (K).

*Minor comments*
*Question 5*

Line 32: the reference (Cui et al., 2021), (Shah et al., 2020) should be (Cui et al., 2021; Shah et al., 2020).

**Response:**

Thanks for your reminding. We apologized that the citation format is wrong and corrected it now in this revision.

**Revisions in Manuscript:**

**(1) Introduction. Line 30-33.**

During the last decade, benefiting from the implementations of several air pollution control strategies by the Chinese government, the air quality has improved (Jin et al., 2016; Zheng et al., 2018), and the vertical column densities of $NO_2$ displayed a decreasing trend after 2013 (Shah et al., 2020; Cui et al., 2021).

*Question 6*

Line 33: delete that: it is still much more severe than that in developed.

**Response:**

Thanks for your advice. We apologized that this statement is not appropriate and deleted it now in this revision.

**Question 7**

Line 40 – 42: The comparison with the emission in Lyon is quite specific. I suggest explaining a bit more or in an alternative to make a more general case of "other urban areas.

**Response:**

Thanks for your suggestion. We agreed that the case in Lyon introduced here was too specific and deleted it now in this revision.

**Revisions in Manuscript:**

**(1) Introduction. Line 40-41.**

Meanwhile, as the population of vehicles is growing rapidly, vehicle emissions have become a major source of $NO_2$ pollution, especially in urban areas (Nguyen et al., 2018).

**Question 8**

Line 102: the spatial resolution should be precise: please substitute < 100 m x 100 m with the real spatial resolution.

**Response:**

Thanks for your reminding, and the original description was misleading. The grid spatial resolution of the hybrid model over the urban area was 50 m x 50 m. We revised the statement to be precise.

**Revisions in Manuscript:**

**(1) Materials and Methods. Line 116-117.**

In our model, a $NO_2$ pollution map with a high temporal (1 h) and spatial resolution (50 m×50 m) can finally be obtained.

**Question 9**

Line 107 -108: The choice of the midpoint height of 22.5m suggests that the CMAQ has a first vertical layer at 45m of height. In the first instance, this would be in my opinion too high. Generally, CTMs have the first 9-10 vertical layers below the boundary layer but, to improve the prediction on the ground level, keep the 1st layer around 20m from the ground.

Could the authors enforce their statements with one or more references that justify this choice.

**Response:**

Thanks for your question. We agreed that in traditional CTMs, it is useful to improve the prediction when the first layer was set to be lower. However, in our hybrid model, we planned to get the wind environment and the background concentrations at the top of the canyon, so the midpoint height of the first layer in both WRF and CMAQ model must be similar with the height of street canyon. In Beijing, the average height of street canyon is 23.6 m, so we set 22.5 m as the midpoint height of the first layer. This setting is similar with that in Benavides's study (Benavides et al., 2019), where the bottom layer in the model was set to be 40.6 m, of which midpoint height was similar to the average building height. Now we added the reference in the **Materials and Methods** section to enforce our statements.

**Revisions in Manuscript:**

**(1) Materials and Methods. Line 132-134.**

The height of midpoint in the bottom layer to the ground was set as 22.5 m, which is close to the average height of buildings near street canyons, similar to the settings in the previous study (Benavides et al., 2019).

*Question 10*

Line 277 – 279: The authors describe the performance of the model in terms of "high" and "low" RE. It would be good to provide a more quantitative description or a reference value for this particular metric.

**Response:**

Thanks for your question. We have already provided a quantitative description about the "high" and "low" RE. The "high" RE referred to the value of 42.5% and 43% in previous sentence. And the "low" RE referred to the value of 9.8% and 2.7% in this sentence. There is not a quantitative criterion to judge whether the RE is high or low, which generally depends on the specific requirement of the experiment. Now we revised our statement and put the quantitative description in this one sentence.

**Revisions in Manuscript:**

**(1) Results. Line 337-339.**

Although the average of the relative error (RE) were a little high (42.5% and 43%), particularly when the predicted wind speed was low, the median RE were relatively low with 9.8% and 2.7%, respectively, indicating an acceptable performance.

*Question 11*

Line 280 – 283: I'm not sure that the MARS model performs better than RD in Figure 5. I suggest clarifying this paragraph better. In the a) figure the CFD is the closest to the observations, followed by RF (red slope) and MARS (yellow slope). In figure b) again CFD is the closest to the observations, RF is completely underestimated and MARS is overestimated from values of z/H > 0.25.

**Response:**

Thanks for your question. Figure 5 was aimed to compare the performance of RF and MARS in two different cases, so we should find between MARS and RF which one is closer to the CFD and observations. In the first case (Figure 5a), the MARS model performed not very well when compared with the RF. However, in another uncommon case when $Vbg_y$=17 m/s >>5 m/s (Figure 5b), RF failed to respond to the parts beyond the range of prediction variables, and the predictions from MARS is closer to the CFD and observations. Therefore, by comparing the performance of two models in Figure 5, the MLSCF scheme was established based on a method to combine the advantages of each model. The RF model was used when the input value was within the range of predictors shown in Table 1, otherwise the predictions from the MARS model were used. We have corrected the description of x-lable in Figure 5b and revised the text under this figure for better understanding.

**Revisions in Manuscript:**

**(1) Manuscript. Figure 5.**

[Figure]

Figure 5: Performances of machine learning on velocity profile in wind tunnel experiments. The street canyon was perpendicular (a) or parallel (b) to the wind direction at the roof level in different experiments. The detailed description of each experiment was introduced in Section 2.2.3.

*Question 12*

Line 378 – 383: The authors mention NOx emissions leading to high NO2 observations among all sites. They also say that the CMAQ model underestimates the NO2 concentrations near ring roads (MB = -15μg/m3). The NOx emissions account for NO+NO2, if this variable is NOx before being inserted in CMAQ, it has to be divided between NO and NO2. In roadside sites generally, the NO emissions are high, could the underestimation in CMAQ be related to a not precise division of the original emissions of NOX in NO and NO2?

**Response:**

Thanks for your question. The setting of the division of the original NOx emissions in NO and $NO_2$ depends on what emission source it is, rather than what model we use. For example, divisions of NOx emission from industry and vehicle are different, but the divisions of NOx emission from vehicles in both CMAQ or our hybrid model remained the same. In our study, the ratio of $NO_2$ to NOx in vehicle emission was set as 0.2 according to previous studies (Benavides et al., 2019; Valencia et al., 2018), which was introduced in the **Supplement Materials Section S3. NOx photochemical parameter scheme** now.

**Revisions in Manuscript:**

**(1) Supplement Materials Section S3. NOx photochemical parameter scheme.**

$\zeta$ is the ratio of $NO_2$ to $NO_x$ in vehicle emissions, with a value of 0.2 (Benavides et al., 2019; Valencia et al., 2018).

*Question 13*

Line 390 – 394: The model actually improves the performance of O3 in comparison with CMAQ only model. This agrees with the underestimation in NO2 that the CMAQ only shows and was previously described by the authors. Being the O3 chemistry dependent not only on NO, and NO2 but also on VOCs I would spend some words introducing these pollutant classes and giving more details on them.

**Response:**

Thanks for your advice. In this study we focused on near-source process, where the $O_3$ was largely affected by the titration of NOx. As mentioned in the previous study (Biggart et al., 2020), roads with higher NOx emissions led to lower $NO_2/NO_x$ concentration ratios within distances of 100m, indicating greater $O_3$ loss through its titration by NO. This is one of the reasons why a simple mechanism only involving NOx and O3 was used in this study.

The influence of VOCs concentrations on pollutant concentrations have been discussed in several studies (Kim et al., 2018; Zhong et al., 2017), and in this study we didn't take it into consideration. Details about this has been described in the *Question 4*.

**Revisions in Manuscript:**

**(1) Conclusion and Discussions. Line 522-527.**

In our study, a simplified two-reaction scheme was incorporated into the model to characterize the photochemical process of $NO_x$, since it performed similar predictions and less computational time compared with those of the complicated CB05 gas phase chemical mechanism (Kim et al., 2018). However, another study pointed that the impact of nonlinear $O_3$-$NO_x$-VOC chemistry on $NO_2$ concentrations in the deep canyon was nonnegligible (Zhong et al., 2017). The influence of different chemistry schemes on near-road simulation will be investigated in the future.

*References*

Benavides J, Snyder M, Guevara M, Soret A, Pérez García-Pando C, Amato F, et al. CALIOPE-Urban v1.0: coupling R-LINE with a mesoscale air quality modelling system for urban air quality forecasts over Barcelona city (Spain). Geosci. Model Dev. 2019a; 12: 2811-2835.

Berchet A, Zink K, Muller C, Oettl D, Brunner J, Emmenegger L, et al. A cost-effective method for simulating city-wide air flow and pollutant dispersion at building resolving scale. Atmospheric Environment 2017; 158: 181-196.

Biggart M, Stocker J, Doherty RM, Wild O, Hollaway M, Carruthers D, et al. Street-scale air quality modelling for Beijing during a winter 2016 measurement campaign. Atmospheric Chemistry and Physics 2020; 20: 2755-2780.

Cui Y, Wang L, Jiang L, Liu M, Wang J, Shi K, et al. Dynamic spatial analysis of NO2 pollution over China: Satellite observations and spatial convergence models. Atmospheric Pollution Research 2021; 12: 89-99.

Hood C, MacKenzie I, Stocker J, Johnson K, Carruthers D, Vieno M, et al. Air quality simulations for London using a coupled regional-to-local modelling system. Atmos. Chem. Phys. 2018; 18: 11221-11245.

Hurley P. The air pollution model (TAPM) version 3. Part 1. Technical description. Aspendale, Vic. CSIRO Atmospheric Research, 2005.

Jensen SS, Ketzel M, Becker T, Christensen J, Brandt J, Plejdrup M, et al. High resolution multi-scale air quality modelling for all streets in Denmark. Transportation Research Part D: Transport and Environment 2017; 52: 322-339.

Jin Y, Andersson H, Zhang S. Air Pollution Control Policies in China: A Retrospective and Prospects. International Journal of Environmental Research and Public Health 2016; 13: 1219.

Kamińska JA. A random forest partition model for predicting NO2 concentrations from traffic flow and meteorological conditions. Science of the Total Environment 2019; 651: 475-483.

Ketzel M, Jensen S, Brandt J, Ellermann T, Berkowicz R, Hertel O. Evaluation of the street pollution model OSPM for measurement at 12 street stations using using newly developed and freely available evaluation tool. J. Civil. Environ. Eng. 2012.

Kim Y, Wu Y, Seigneur C, Roustan Y. Multi-scale modeling of urban air pollution: development and application of a Street-in-Grid model (v1.0) by coupling MUNICH (v1.0) and Polair3D (v1.8.1). Geosci. Model Dev. 2018a; 11: 611-629.

Lefebvre W, Van Poppel M, Maiheu B, Janssen S, Dons E. Evaluation of the RIO-IFDM-street canyon model chain. Atmospheric Environment 2013; 77: 325-337.

Lv Z, Wang X, Deng F, Ying Q, Archibald AT, Jones RL, et al. Source–Receptor Relationship Revealed by the Halted Traffic and Aggravated Haze in Beijing during the COVID-19 Lockdown. Environmental Science & Technology 2020; 54: 15660-15670.

Mallet V, Tilloy A, Poulet D, Girard S, Brocheton F. Meta-modeling of ADMS-Urban by dimension reduction and emulation. Atmospheric Environment 2018; 184: 37-46.

Mu Q, Denby BR, Wærsted EG, Fagerli H. Downscaling of air pollutants in Europe using uEMEP_v6. Geosci. Model Dev. 2022; 15: 449-465.

Shah V, Jacob DJ, Li K, Silvern RF, Zhai S, Liu M, et al. Effect of changing NOx lifetime on the seasonality and long-term trends of satellite-observed tropospheric NO2 columns over China. Atmos. Chem. Phys. 2020; 20: 1483-1495.

Soulhac L, Nguyen C, Volta P, Salizzoni P. The model SIRANE for atmospheric urban pollutant

dispersion. PART III: Validation against NO 2 yearly concentration measurements in a large urban agglomeration. Atmospheric Environment 2017; 167.

Stocker J, Hood C, Carruthers D, McHugh C. ADMS-Urban: developments in modelling dispersion from the city scale to the local scale. International Journal of Environment and Pollution 2012; 50: 308-316.

Valencia A, Venkatram A, Heist D, Carruthers D, Arunachalam S. Development and evaluation of the R-LINE model algorithms to account for chemical transformation in the near-road environment. Transportation Research Part D: Transport and Environment 2018; 59: 464-477.

Zheng B, Tong D, Li M, Liu F, Hong C, Geng G, et al. Trends in China's anthropogenic emissions since 2010 as the consequence of clean air actions. Atmos. Chem. Phys. 2018; 18: 14095-14111.

Zhong J, Cai X-M, Bloss WJ. Large eddy simulation of reactive pollutants in a deep urban street canyon: Coupling dynamics with O3-NOx-VOC chemistry. Environmental Pollution 2017; 224: 171-184.

---

## Referee Report (RR1)

This work presents a Multi-scale system to predict NO2 at high-spatial resolution. The system combines mesoscale results (CMAQ) with a Gaussian dispersion solver (RLINE). The study deals with the coupling between these two scales. The main innovation is the approach to derive local wind velocity within street canyons. The idea of using CFD simulations to train machine learning models to improve classic Gaussian Dispersion models is interesting and timely. However, some points of the current methodology need to be clarified before publication.

1) As stated in the manuscript, ML methods estimate "the wind vector along X-axis and Y-axis at different heights within the street canyon respectively". However, where is this vertical profile located within the street canyon? Are the edges of the canyon affecting the profile? Are 3D CFD simulations justified?

2) In RLINE, a single wind speed value is associated with each line source. Could you please clarify in the manuscript how this wind speed has been derived for the different presented approaches (CMAQ-RLINE, CMAQ-RLINE-Urban-nc, CMAQ-RLINE-Urban)?

3) How intersections of two different street-canyons are treated?

4) The heat flux condition of the vertical mixing parametrization is different from the one presented in Benavides et al. Also, Benavides's parametrization was adjusted to work with MOST estimation of surface wind speed. How is this parametrization still valid when using a different surface wind speed approach?
Also, to compute the ratio WSsfc/WSbh for the vertical mixing parametrization, how is WSsfc computed along the canyon, given that the ML surrogate provides a single vertical profile?

5) To fairly assess the ML performance, data from a velocity profile for a given set of predictors (Vbgx, Vbgy, z/H, Hl/Hr, H/W,L/W) should not be split between training and test sets. Then for the testing, the ML models will predict the entire velocity profile without seeing any other data point from this profile. This more restrictive data splitting will estimate better the performance in unseen street canyons.

6) How does this approach account for all non-local vehicle emissions?

7) Is the building geometry constraining the Gaussian dispersion?

8) A maximum of 1e4 iterations in the CFD solver are considered to avoid extra CPU time. From the full 1600 CFD simulations, what percentage did not reach the convergence conditions described in the paper?

9) What are the CFD mesh refinements criteria? The CFD mesh should be refined close to walls to capture strong velocity gradients, and it does not seem to be the case.

10) The described boundary conditions are not in accordance with Fig. 2. Also, please report the number of CFD mesh points and simulation time.

11) What is the spatial resolution of CMAQ /WRF?

12) Add ref. for the observation data in Fig. 5. Figure captions should generally be improved to facilitate the reading.

13) Clarify the innovation aspect in the introduction.

14) Section S1 is incomplete. How are $Z_{mix}$, convective velocity scale $w^*$, surface friction velocity $u^*$, and Monin-Obhukov length recalculated?

15) What are the specific input data used in RLINE?

16) How the facbg is further used in the photochemical scheme? Is $[NO2]_b = facbg*[NO2]_{cmaq}$? If yes, are O3 and NO also scaled with facbg?

---

## Author Response (AR2)

**Authors' Response:**

**Manuscript Title**: Development and application of a multi-scale modelling framework for urban high-resolution NO₂ pollution mapping

**Discussion Link:** https://acp.copernicus.org/preprints/acp-2022-371/#discussion

**Revision notes:**

| |
|---|
| Reviewers' comments are in blue italic type.

    Authors' responses are in indent and in black normal font.

    Revisions in the manuscript are in indent red normal font. |

**Content**

**Response to Editor's Comments**

For the next revision, please add the name of the corresponding author into the line 9 (title page of the *.pdf manuscript file). 2. Regarding the figures #11: with the next revision, please check if a copyright statement/image credit is required and add it to the figure caption, if applicable. If you are the originator, you can just inform us via email.

**Response:**

Thanks for reminding. We have added the name of the corresponding author and a copyright statement/image credit in the manuscript.

**Revisions in Manuscript:**

**(1) Line 9.**

*Phone and fax: 86-10-62771679; e-mail: liu_env@tsinghua.edu.cn (Huan Liu)

**(2) Figure 11.**

Figure 11: Spatial distribution of monthly averaged NO2 concentrations from (a) CMAQ model and (b) CMAQ-RLINE_URBAN model (© OpenStreetMap contributors 2020. Distributed under the Open Data Commons Open Database License (ODbL) v1.0).

**Response to Reviewers #2's Comments**

*Summary*

The revised version has been much improved. All my concerns on the previous version have been addressed properly and quite clearly. I only have a few further minor comments to be considered.

**Response:**

Thank you very much to give us comments. We believed all the concerns you mentioned at this time were addressed in this revision.

*Question 1*

Line 153-155. Apart from the general description of the emission inventory, some quantities (e.g. total emission or emission factors) about the emissions for the simulation period would be helpful.

**Response:**

Thanks for your advice. We totally agreed, and the description about the emission amount and the main contribution of vehicle types has been added in the **Materials and Methods**.

**Revisions in Manuscript:**

**(1) Materials and Methods, Line 158-160.**

The daily averaged $NO_x$ emission from on-road vehicles in Beijing in 2019 was estimated to be 136.0 Mg, of which emissions from heavy duty vehicles and heavy duty trucks accounted for 31% and 34%, respectively.

*Question 2*

It is suggested to add the running cost for the hybrid model CMAQ-RLINE_URBAN for the 50 m x 50 m resolution run, which may guide the future development towards higher resolutions at the street scale.

**Response:**

Thanks for your advice. In the base scenario CMAQ-RLINE_URBAN, in the hybrid model, the local meteorological field should be calculated separately according to the location of road and receptor points, thus the average simulation duration per unit hour is about 3.9 hours, and can reach 4.8 hours at night when the atmospheric stability is high

and the results were difficult to convergence. The running cost has been added in **Conclusion and Discussions**.

**Revisions in Manuscript:**

**(1) Conclusion and Discussions. Line 534-536.**

At present, the execution time during 1 h running CMAQ-RLINE_URBAN over the urban domain was about 3.9 hours in average, which reached 4.8 hours at night due to the difficulty of convergence in the condition of the high atmospheric stability.

*Question 3*

Health impact of NO2 pollution (to be linked with population data and health data) could be added into the future research.

**Response:**

Thanks for your advice. This has been added in **Conclusion and Discussions**.

**Revisions in Manuscript:**

**(1) Conclusion and Discussions. Line 554-555.**

The high resolution $NO_2$ concentration map was benefit for the estimation of human health risks induced by the air pollution at the street level in future researches.

**Response to Reviewers #4's Comments**

*Summary*

This work presents a Multi-scale system to predict NO2 at high-spatial resolution. The system combines mesoscale results (CMAQ) with a Gaussian dispersion solver (RLINE). The study deals with the coupling between these two scales. The main innovation is the approach to derive local wind velocity within street canyons. The idea of using CFD simulations to train machine learning models to improve classic Gaussian Dispersion models is interesting and timely. However, some points of the current methodology need to be clarified before publication.

**Response:**

Thank you very much for spending time to give us so many constructive comments. Upon learning through them, we improved our manuscript. We try our best to address all the concerns in this revision, where more details about the methodology was introduced and clarified.

*Question 1*

As stated in the manuscript, ML methods estimate "the wind vector along X-axis and Y-axis at different heights within the street canyon respectively". However, where is this vertical profile located within the street canyon? Are the edges of the canyon affecting the profile? Are 3D CFD simulations justified?

**Response:**

Thanks for your question. We apologized that the description of the wind vector used in the ML method was unclear. Here, the wind vector used in the ML method is the average velocity of all horizontal CFD grids at same the height within the street canyon. For example, if there are 100 CFD grids at the height of 1.5 m within the street canyon, the wind vector was the average of predictions of these all grids. Therefore, the influence of edges of the canyon on the wind profile is included in predictions from ML method. We have revised the description of the wind vector along X-axis and Y-axis in the manuscript for better understating.

For 3D CFD simulations, the configurations refer to the CFD guideline (Tominaga et al., 2008; Franke et al., 2011) and previous studies (Hang et al., 2012; Hang et al., 2010), and

the performance of the simulation is validated by wind tunnel data (Brown et al., 2001; Hang et al., 2010) (Figure S3), thus we believe the 3D CFD simulations are justified.

**Revisions in Manuscript:**

**(1) Materials and Methods. Line 287-288.**

The $V_x$ and $V_y$ were the average of all velocities along X or Y axis over the same horizontal profile at a specific height within the street canyons.

*Question 2*

In RLINE, a single wind speed value is associated with each line source. Could you please clarify in the manuscript how this wind speed has been derived for the different presented approaches (CMAQ-RLINE, CMAQ-RLINE-Urban-nc, CMAQ-RLINE-Urban).

**Response:**

Thanks for your advice. As we have introduced in the manuscript (Line 137-140), the wind at the top of street canyon was provided by the WRF for all scenarios (CMAQ-RLINE, CMAQ-RLINE-Urban_nc, and CMAQ-RLINE-Urban), and the wind environment of each road were obtained separately from WRF model according to the road location. The difference between different scenarios is the wind speed within the street canyon. For CMAQ-RLINE and CMAQ-RLINE-Urban_nc, the wind profiles within the street canyon are calculated by the MOST theory. For CMAQ-RLINE_URBAN, the wind profile within the street canyon is calculated by MLSCF. We have added this description in the manuscript for better understanding.

**Revisions in Manuscript:**

**(1) Materials and Methods. Line 338-341.**

Although the wind environment for each road at the top of the canyon was provide by the WRF model in all scenarios, the calculation of wind profile within the street canyon was different. It was estimated based on the MOST theory in the CMAQ-RLINE and CMAQ-RLINE_URBAN_nc rather than that from the MLSCF in the CMAQ-RLINE_URBAN.

*Question 3*

How intersections of two different street-canyons are treated.

**Response:**

Thanks for your question. The main simulation is based on the RLINE model, which is a Gaussian line source dispersion model rather than a street box model such as the SIRANE and MUNICH model. The description of the RLINE model has been introduced in **Line 122-127** in the manuscript. Therefore, the calculation of intersection between different road is not involved and each street canyon is treated as a single source in simulation.

*Question 4*

The heat flux condition of the vertical mixing parametrization is different from the one presented in Benavides et al. Also, Benavides's parametrization was adjusted to work with MOST estimation of surface wind speed. How is this parametrization still valid when using a different surface wind speed approach?

Also, to compute the ratio WSsfc/WSbh for the vertical mixing parametrization, how is WSsfc computed along the canyon, given that the ML surrogate provides a single vertical profile?

**Response:**

Thanks for questions. Firstly, when the background concentration mixing ratio ($fac_{bg}$) was calculated, actually we changed the threshold of heat flux from 0.3 to 0 compared with the Benavides's study. The transition value of 0.3 in Benavides's study is set to consider the impact of Urban Heat Island (Kheirbek et al.) effect, which is already considered in the UHI scheme of our hybrid model. Therefore, this change was aim to avoid the double-counting of impacts from the UHI effect, which was added into the description of the vertical mixing parametrization in **Supplement Section S2**.

Whether in our hybrid model or Benavides's study, we both assumed that the ratio of wind speed near the surface and the top of street can be used as a proxy parameter to characterize the turbulence intensity which influenced the vertical mixing of background concentrations in canyons. Although we use a different wind profile approach in street canyons compared with that in the Benavides's study, the basic theory of the vertical mixing of background concentrations in canyons was unchanged. Therefore, we believed that this parametrization still valid in our hybrid model.

We apologized again for the missing understanding of the output from MLSCF scheme, which is explained in the answer of **Question 1**. So WSsfc was the same at a specific height and canyon rather than different along the canyon,

**Revisions in Manuscript:**

**(1) Supplement Section S2. Vertical mixing scheme**

The transition value of $H_u$ changed from 0.3 to 0 in this research compared with the Benavides's study in order to avoid the double-counting of impacts from the UHI effect.

*Question 5*

To fairly assess the ML performance, data from a velocity profile for a given set of predictors (Vbgx, Vbgy, z/H, Hl/Hr, H/W,L/W) should not be split between training and test sets. Then for the testing, the ML models will predict the entire velocity profile without seeing any other data point from this profile. This more restrictive data splitting will estimate better the performance in unseen street canyons.

**Response:**

Thank you for advice. According to the train strategy you proposed, we think you are concerned about the generalization ability of the machine learning model. There is no doubt that the training strategy you proposed will make the performance of the model acceptable in unseen street canyon. Although we split training and test sets randomly in this study, the sample in the test set with specific height in the specific canyon was never included in the training set. Therefore, the generalization ability of our model is believable, and has been validated in **Figure 5**. This figure showed the comparison between observations and predictions from MLSCF in two CFD validation cases, where the geometry of canyons and background wind were never included in the training sets of MLSCF, and the acceptable model performance were shown.

*Question 6*

How does this approach account for all non-local vehicle emissions?

**Response:**

Thanks for your question. As the statement in our manuscript (Line 162 to 165), the impact

of all non-local vehicle emissions has been involved in the background concentrations simulated by CMAQ-ISAM model, in which the emissions were divided into mobile and other four emission groups to trace their contributions separately. The detailed configuration of CMAQ-ISAM model has been already introduced in our previous study (Lv et al., 2020).

**Revisions in Manuscript:**

**(1) Materials and Methods. Line 162-164.**

These background concentrations were simulated by CMAQ-ISAM model, in which the emissions were divided into local mobile and other four emission groups to trace their contributions separately, so the influence if non-local emission could be considered.

*Question 7*

Is the building geometry constraining the Gaussian dispersion?

**Response:**

Thanks for your question. In our hybrid model, the building geometry influenced the wind in street canyons using the MLSCF scheme, and also the surface roughness length as well as the atmospheric turbulence intensity using the $z_0$ scheme. Therefore, the building geometry influenced the Gaussian dispersion indirectly.

*Question 8*

A maximum of 1e4 iterations in the CFD solver are considered to avoid extra CPU time. From the full 1600 CFD simulations, what percentage did not reach the convergence conditions described in the paper?

**Response:**

Thanks for your question. As shown in the Figure below, most simulation cases (54.6%) met the convergence criteria ($10^{-5}$). The median residual error of continuity equation, velocity in X direction, velocity in Y direction, velocity in Z direction, k and $\varepsilon$ are $1.0\times10^{-5}$, $8.5\times10^{-7}$, $8.5\times10^{-7}$, $4.1\times10^{-7}$, $3.4\times10^{-6}$ and $5.4\times10^{-6}$, respectively, indicating that the overall model performance is acceptable. However, the residual error of continuity equation was relatively large, the number of cases with residuals greater than $10^{-5}$ and $10^{-4}$ accounts for 45.4% and 1.5% respectively. Particularly, when the external wind was

parallel to the street and L/H was large, more iteration steps are required for the velocity in street canyons become a steady state, which should be improved in the future. And now we have added a brief description of the convergence conditions in the manuscript.

[Figure]

Figure. Residual distributions of each parameter in 1600 CFD simulation scenarios

**Revisions in Manuscript:**

**(1) Materials and Methods. Line 226-230.**

About 54.6% of cases met the convergence criteria, and the median residual values of continuity equation, velocity in X axis, velocity in Y axis, velocity in Z axis, $k$ and $\varepsilon$ were $1.0\times10^{-5}$, $8.5\times10^{-7}$, $8.5\times10^{-7}$, $4.1\times10^{-7}$, $3.4\times10^{-6}$ and $5.4\times10^{-6}$, respectively, indicating the overall model performance was acceptable.

*Question 9*

What are the CFD mesh refinements criteria? The CFD mesh should be refined close to walls to capture strong velocity gradients, and it does not seem to be the case.

**Response:**

Thanks for your question. In the CFD validation (Section 2.2.3), we identify the influence of different minimum sizes of hexahedral cells near wall surfaces (fine: 0.1m, medium: 0.2m, and coarse: 0.5m) with an expansion ratio of 1.1 on the predicted velocity to an expansion ratio of 1.1, and we find different grid resolutions used in simulations would not obviously affect the predicted results (Figure S3). Therefore, the minimum size of

hexahedral cells near wall surfaces was 0.5 m with an expansion ratio of 1.1 is applied to save the computing cost (**Line 259-260**).

*Question 10*

The described boundary conditions are not in accordance with Fig. 2. Also, please report the number of CFD mesh points and simulation time.

**Response:**

Thanks for your advice. We apologized that the distance between UCL and the domain top was wrong in the manuscript. We have revised its description in the manuscript.

The mesh number of 80 (5×4×4) CFD models is shown in the Table below. The average mesh number of 80 street canyon models is 136,7965.

Table. The mesh number of 80 CFD models

| ID | Hl/Hr | H/W | L/H | Hl | Hr | W | L | Cells |
|----|-------|------|-----|----|----|----|-----|---------|
| 1  | 0.50  | 0.25 | 3   | 14 | 28 | 84 | 63  | 1433054 |
| 2  | 0.50  | 0.25 | 5   | 14 | 28 | 84 | 105 | 1540924 |
| 3  | 0.50  | 0.25 | 10  | 14 | 28 | 84 | 210 | 1681155 |
| 4  | 0.50  | 0.25 | 20  | 14 | 28 | 84 | 420 | 1832173 |
| 5  | 0.50  | 0.50 | 3   | 14 | 28 | 42 | 63  | 1330574 |
| 6  | 0.50  | 0.50 | 5   | 14 | 28 | 42 | 105 | 1430044 |
| 7  | 0.50  | 0.50 | 10  | 14 | 28 | 42 | 210 | 1559355 |
| 8  | 0.50  | 0.50 | 20  | 14 | 28 | 42 | 420 | 1698613 |
| 9  | 0.50  | 1.00 | 3   | 14 | 28 | 21 | 63  | 1236634 |
| 10 | 0.50  | 1.00 | 5   | 14 | 28 | 21 | 105 | 1328404 |
| 11 | 0.50  | 1.00 | 10  | 14 | 28 | 21 | 210 | 1447705 |
| 12 | 0.50  | 1.00 | 20  | 14 | 28 | 21 | 420 | 1576183 |
| 13 | 0.50  | 2.00 | 3   | 14 | 28 | 11 | 63  | 1168314 |
| 14 | 0.50  | 2.00 | 5   | 14 | 28 | 11 | 105 | 1254484 |
| 15 | 0.50  | 2.00 | 10  | 14 | 28 | 11 | 210 | 1366505 |
| 16 | 0.50  | 2.00 | 20  | 14 | 28 | 11 | 420 | 1487143 |
| 17 | 0.75  | 0.25 | 3   | 18 | 24 | 84 | 63  | 1332478 |
| 18 | 0.75  | 0.25 | 5   | 18 | 24 | 84 | 105 | 1433068 |
| 19 | 0.75  | 0.25 | 10  | 18 | 24 | 84 | 210 | 1563835 |
| 20 | 0.75  | 0.25 | 20  | 18 | 24 | 84 | 420 | 1704661 |
| 21 | 0.75  | 0.50 | 3   | 18 | 24 | 42 | 63  | 1257204 |
| 22 | 0.75  | 0.50 | 5   | 18 | 24 | 42 | 105 | 1330108 |
| 23 | 0.75  | 0.50 | 10  | 18 | 24 | 42 | 210 | 1450735 |
| 24 | 0.75  | 0.50 | 20  | 18 | 24 | 42 | 420 | 1580641 |

| 25 | 0.75 | 1.00 | 3 | 18 | 24 | 21 | 63 | 1150088 |
|----|------|------|----|----|----|----|-----|---------|
| 26 | 0.75 | 1.00 | 5 | 18 | 24 | 21 | 105 | 1235728 |
| 27 | 0.75 | 1.00 | 10 | 18 | 24 | 21 | 210 | 1347060 |
| 28 | 0.75 | 1.00 | 20 | 18 | 24 | 21 | 420 | 1466956 |
| 29 | 0.75 | 2.00 | 3 | 18 | 24 | 11 | 63 | 1118508 |
| 30 | 0.75 | 2.00 | 5 | 18 | 24 | 11 | 105 | 1167088 |
| 31 | 0.75 | 2.00 | 10 | 18 | 24 | 11 | 210 | 1271660 |
| 32 | 0.75 | 2.00 | 20 | 18 | 24 | 11 | 420 | 1403380 |
| 33 | 1.00 | 0.25 | 3 | 21 | 21 | 84 | 63 | 1168566 |
| 34 | 1.00 | 0.25 | 5 | 21 | 21 | 84 | 105 | 1256796 |
| 35 | 1.00 | 0.25 | 10 | 21 | 21 | 84 | 210 | 1371495 |
| 36 | 1.00 | 0.25 | 20 | 21 | 21 | 84 | 420 | 1495017 |
| 37 | 1.00 | 0.50 | 3 | 21 | 21 | 42 | 63 | 1085118 |
| 38 | 1.00 | 0.50 | 5 | 21 | 21 | 42 | 105 | 1166508 |
| 39 | 1.00 | 0.50 | 10 | 21 | 21 | 42 | 210 | 1272315 |
| 40 | 1.00 | 0.50 | 20 | 21 | 21 | 42 | 420 | 1386261 |
| 41 | 1.00 | 1.00 | 3 | 21 | 21 | 21 | 63 | 1008624 |
| 42 | 1.00 | 1.00 | 5 | 21 | 21 | 21 | 105 | 1083744 |
| 43 | 1.00 | 1.00 | 10 | 21 | 21 | 21 | 210 | 1181400 |
| 44 | 1.00 | 1.00 | 20 | 21 | 21 | 21 | 420 | 1286568 |
| 45 | 1.00 | 2.00 | 3 | 21 | 21 | 11 | 63 | 952992 |
| 46 | 1.00 | 2.00 | 5 | 21 | 21 | 11 | 105 | 1023552 |
| 47 | 1.00 | 2.00 | 10 | 21 | 21 | 11 | 210 | 1115280 |
| 48 | 1.00 | 2.00 | 20 | 21 | 21 | 11 | 420 | 1214064 |
| 49 | 1.33 | 0.25 | 3 | 24 | 18 | 84 | 63 | 1332478 |
| 50 | 1.33 | 0.25 | 5 | 24 | 18 | 84 | 105 | 1433068 |
| 51 | 1.33 | 0.25 | 10 | 24 | 18 | 84 | 210 | 1563835 |
| 52 | 1.33 | 0.25 | 20 | 24 | 18 | 84 | 420 | 1704661 |
| 53 | 1.33 | 0.50 | 3 | 24 | 18 | 42 | 63 | 1257204 |
| 54 | 1.33 | 0.50 | 5 | 24 | 18 | 42 | 105 | 1330108 |
| 55 | 1.33 | 0.50 | 10 | 24 | 18 | 42 | 210 | 1450735 |
| 56 | 1.33 | 0.50 | 20 | 24 | 18 | 42 | 420 | 1580641 |
| 57 | 1.33 | 1.00 | 3 | 24 | 18 | 21 | 63 | 1150088 |
| 58 | 1.33 | 1.00 | 5 | 24 | 18 | 21 | 105 | 1235728 |
| 59 | 1.33 | 1.00 | 10 | 24 | 18 | 21 | 210 | 1347060 |
| 60 | 1.33 | 1.00 | 20 | 24 | 18 | 21 | 420 | 1466956 |
| 61 | 1.33 | 2.00 | 3 | 24 | 18 | 11 | 63 | 1118508 |
| 62 | 1.33 | 2.00 | 5 | 24 | 18 | 11 | 105 | 1167088 |
| 63 | 1.33 | 2.00 | 10 | 24 | 18 | 11 | 210 | 1271660 |
| 64 | 1.33 | 2.00 | 20 | 24 | 18 | 11 | 420 | 1403380 |
| 65 | 2.00 | 0.25 | 3 | 28 | 14 | 84 | 63 | 1433054 |
| 66 | 2.00 | 0.25 | 5 | 28 | 14 | 84 | 105 | 1540924 |
| 67 | 2.00 | 0.25 | 10 | 28 | 14 | 84 | 210 | 1681155 |

| 68 | 2.00 | 0.25 | 20 | 28 | 14 | 84 | 420 | 1832173 |
|----|------|------|----|----|----|----|-----|---------|
| 69 | 2.00 | 0.50 | 3 | 28 | 14 | 42 | 63 | 1330574 |
| 70 | 2.00 | 0.50 | 5 | 28 | 14 | 42 | 105 | 1430044 |
| 71 | 2.00 | 0.50 | 10 | 28 | 14 | 42 | 210 | 1559355 |
| 72 | 2.00 | 0.50 | 20 | 28 | 14 | 42 | 420 | 1698613 |
| 73 | 2.00 | 1.00 | 3 | 28 | 14 | 21 | 63 | 1236634 |
| 74 | 2.00 | 1.00 | 5 | 28 | 14 | 21 | 105 | 1328404 |
| 75 | 2.00 | 1.00 | 10 | 28 | 14 | 21 | 210 | 1447705 |
| 76 | 2.00 | 1.00 | 20 | 28 | 14 | 21 | 420 | 1576183 |
| 77 | 2.00 | 2.00 | 3 | 28 | 14 | 11 | 63 | 1168314 |
| 78 | 2.00 | 2.00 | 5 | 28 | 14 | 11 | 105 | 1254484 |
| 79 | 2.00 | 2.00 | 10 | 28 | 14 | 11 | 210 | 1366505 |
| 80 | 2.00 | 2.00 | 20 | 28 | 14 | 11 | 420 | 1487143 |

We apologize for the fact that the simulation time of CFD is not recorded. However, the iteration number of each simulation is recorded. The distribution of iteration number of 1600 simulations is shown in the Figure below. The average of that is 4443.

[Figure]

Figure. The distribution of iteration number of 1600 simulations.

**Revisions in Manuscript:**

**(1) Materials and Methods. Line 226-227.**

In summary, the average iteration steps of total 1600 cases were 4,443.

**(2) Materials and Methods. Line 260-261.**

The average mesh number in total 80 street canyon models is 1,367,965.

**(3) Materials and Methods. Line 206-207.**

Distances between urban canopy layers (UCL) boundaries and the domain top, domain inlet and domain outlet were set as 5$H$, 5$H$, and 20$H$, respectively.

**Response:**

Thanks for your question. The WRF-CMAQ system draws on a 4-nested run with a horizontal resolution at 1.33 km of the innermost domain. This introduction was added into the manuscript.

**Revisions in Manuscript:**

**(1) Materials and Methods. Line 165-166.**

The spatial resolution of the innermost domain in both WRF and CMAQ model was 1.33 km×1.33 km.

*Question 12*

Add ref. for the observation data in Fig. 5. Figure captions should generally be improved to facilitate the reading?

**Response:**

Thanks for your advice. We have added the references in Figure 5 to facilitate the reading.

**Revisions in Manuscript:**

**(1) Figures.**

[Figure]

Figure 5: Performances of machine learning on velocity profile in wind tunnel experiments. The street canyon was perpendicular (a) or parallel (b) to the wind direction at the roof level in different experiments. The detailed description of each experiment was introduced in

*Question 13*

Clarify the innovation aspect in the introduction.

**Response:**

Thanks for your advice. The innovation of this study is that we developed a hybrid model to combine the advantages of the dispersion model and CTMs, where a cost-effective way using ML was proposed to simulate the street-level wind environment. We have already introduced advantages and limitation of different models, including CFD, the dispersion model and CTMs. In addition, the review of multi-scale air quality model was also introduced. In the revised introduction section, we have clarified the aim and innovation aspect more explicitly in the Introduction.

**Revisions in Manuscript:**

**(1) Introduction. Line 98-99.**

The objective of the present work is to investigate the street-level $NO_2$ concentrations and quantify the contribution of vehicle emissions considering the influence of the refined wind flow in complex urban environment.

**(2) Introduction. Line 102-104.**

We developed a Machine Learning-based Street Canyon Flow (MLSCF) parameterization scheme to estimate the wind filed in a cost-effective way, which was based on integrating two machine learning methods using big wind profile data from 1600 CFD simulations.

*Question 14*

Section S1 is incomplete. How are Z mix , convective velocity scale w * , surface friction velocity u * , and Monin-Obhukov length recalculated?

**Response:**

Thanks for your question. The mixing height $Z_{mix}$, convective velocity scale $w^*$, surface friction velocity $u^*$, and Monin-Obhukov length $L_{MO}$ were recalculated based on AERMOD method (Cimorelli et al., 2005; Epa, 2019).

**Revisions in Supplement:**

**(1) Section S1. Urban heat island scheme.**

And then the mixing height $Z_{mix}$, convective velocity scale $w^*$, surface friction velocity $u^*$, and Monin-Obhukov length $L_{MO}$ were recalculated based on AERMOD method (Cimorelli et al., 2005; Epa, 2019).

*Question 15*

What are the specific input data used in RLINE?

**Response:**

Thanks for your question. The input data of RLINE including road information, receptor information, meteorological parameters from WRF model, background concentrations from non-vehicle sources provided by CMAQ-ISAM model, and real-time on-road emission, which are contents of the square box in Figure 1.

*Question 16*

How the facbg is further used in the photochemical scheme? Is [NO2]b = facbg*[NO2]cmaq ? If yes, are O3 and NO also scaled with facbg?

**Response:**

Thanks for your question. In the scenario CMAQ-RLINE, the vertical mixing scheme is not included, so the $fac_{bg}$ is not calculated. In the scenario CMAQ-RLINE_URBAN and CMAQ-RLINE_URBAN_nc, the $fac_{bg}$ is calculated in the vertical mixing scheme and further used in the photochemical scheme, so the $NO_2$, $NO$, and $O_3$ are scaled with $fac_{bg}$ to derive $[NO_2]_b$, $[NO]_b$, and $[O_3]_b$. We added this

**(1) Section S3. NOx photochemical parameter scheme.**

If the vertical mixing scheme is used in the hybrid model, $[NO]_b$, $[NO_2]$, and $[O_3]_b$ are derived by multiplying background concentrations from CMAQ-ISAM model with $fac_{bg}$.

**References**

Brown, M., Lawson, R., DeCroix, D., and Lee, R.: COMPARISON OF CENTERLINE VELOCITY MEASUREMENTS OBTAINED AROUND 2D AND 3D BUILDING ARRAYS IN A WIND TUNNEL, 2001.

Cimorelli, A. J., Perry, S. G., Venkatram, A., Weil, J. C., Paine, R. J., Wilson, R. B., Lee, R. F.,

Peters, W. D., and Brode, R. W.: AERMOD: A Dispersion Model for Industrial Source Applications. Part I: General Model Formulation and Boundary Layer Characterization, Journal of Applied Meteorology, 44, 682-693, 10.1175/JAM2227.1, 2005.

EPA: AERMOD model formulation and evaluation[R]. EPA-454/R-19-014. US Environmental Protection Agency, Research Triangle Park, NC., 2019.

Franke, J., Hellsten, A., Schlunzen, K. H., and Carissimo, B.: The COST 732 Best Practice Guideline for CFD simulation of flows in the urban environment: a summary, International Journal of Environment and Pollution, 44, 419-427, 10.1504/IJEP.2011.038443, 2011.

Hang, J., Sandberg, M., Li, Y., and Claesson, L.: Flow mechanisms and flow capacity in idealized long-street city models, Building and Environment, 45, 1042-1053, https://doi.org/10.1016/j.buildenv.2009.10.014, 2010.

Hang, J., Li, Y., Sandberg, M., Buccolieri, R., and Di Sabatino, S.: The influence of building height variability on pollutant dispersion and pedestrian ventilation in idealized high-rise urban areas, Building and Environment, 56, 346-360, https://doi.org/10.1016/j.buildenv.2012.03.023, 2012.

Kheirbek, I., Haney, J., Douglas, S., Ito, K., and Matte, T.: The contribution of motor vehicle emissions to ambient fine particulate matter public health impacts in New York City: a health burden assessment, Environmental Health, 15, 89, 10.1186/s12940-016-0172-6, 2016.

Lv, Z., Wang, X., Deng, F., Ying, Q., Archibald, A. T., Jones, R. L., Ding, Y., Cheng, Y., Fu, M., Liu, Y., Man, H., Xue, Z., He, K., Hao, J., and Liu, H.: Source–Receptor Relationship Revealed by the Halted Traffic and Aggravated Haze in Beijing during the COVID-19 Lockdown, Environmental Science & Technology, 54, 15660-15670, 10.1021/acs.est.0c04941, 2020.

Tominaga, Y., Mochida, A., Yoshie, R., Kataoka, H., Nozu, T., Yoshikawa, M., and Shirasawa, T.: AIJ guidelines for practical applications of CFD to pedestrian wind environment around buildings, Journal of Wind Engineering and Industrial Aerodynamics, 96, 1749-1761, https://doi.org/10.1016/j.jweia.2008.02.058, 2008.